# Targeting advanced prostate cancer with STEAP1 chimeric antigen receptor T cell and tumor-localized IL-12 immunotherapy

Vipul Bhatia[1,10], Nikhil V. Kamat[2,10], Tiffany E. Pariva[1,10], Li-Ting Wu[1], Annabelle Tsao [1], Koichi Sasaki [3], Huiyun Sun[1], Gerardo Javier[1], Sam Nutt[1], Ilsa Coleman[1], Lauren Hitchcock[1], Ailin Zhang[1], Dmytro Rudoy[1], Roman Gulati [4], Radhika A. Patel[1], Martine P. Roudier[5], Lawrence D. True[6], Shivani Srivastava [1], Colm M. Morrissey[5], Michael C. Haffner[1,6,7], Peter S. Nelson [1,2,4,5,6,7], Saul J. Priceman [8,9], Jun Ishihara [3] ✉ & John K. Lee [1,2,6,7] ✉

Six transmembrane epithelial antigen of the prostate 1 (STEAP1) is a cell surface antigen for therapeutic targeting in prostate cancer. Here, we report broad expression of STEAP1 relative to prostate-specific membrane antigen (PSMA) in lethal metastatic prostate cancers and the development of a STEAP1-directed chimeric antigen receptor (CAR) T cell therapy. STEAP1 CAR T cells demonstrate reactivity in low antigen density, antitumor activity across metastatic prostate cancer models, and safety in a human STEAP1 knock-in mouse model. STEAP1 antigen escape is a recurrent mechanism of treatment resistance and is associated with diminished tumor antigen processing and presentation. The application of tumor-localized interleukin-12 (IL-12) therapy in the form of a collagen binding domain (CBD)-IL-12 fusion protein combined with STEAP1 CAR T cell therapy enhances antitumor efficacy by remodeling the immunologically cold tumor microenvironment of prostate cancer and combating STEAP1 antigen escape through the engagement of host immunity and epitope spreading.

Metastatic prostate cancer represents an incurable disease responsible for over 33,000 deaths per year in the United States[1]. Prostate cancer is critically reliant on androgen receptor (AR) signaling and thus the suppression of gonadal androgen production through surgical or chemical castration (androgen deprivation therapy) has been a mainstay of treatment for advanced disease. However, metastatic prostate cancer inevitably develops resistance to androgen deprivation therapy

and enters a stage called metastatic castration-resistant prostate cancer (mCRPC). mCRPC is currently incurable and is considered the end-stage of the disease and is associated with a median overall survival of three years[2]. In the past decade, multiple therapies including an inhibitor of extragonadal androgen synthesis (abiraterone acetate)[3], second-generation AR antagonists (enzalutamide)[4], radioactive isotope (radium-223)[5], and a prostate-specific membrane antigen

[1]Human Biology Division, Fred Hutchinson Cancer Center, 1100 Fairview Ave N, Seattle, WA 98109, USA. [2]Division of Medical Oncology, University of Washington, 1959 NE Pacific Street, Seattle, WA 98195, USA. [3]Department of Bioengineering, Imperial College London, 86 Wood Lane, London W12 0BZ, UK. [4]Public Health Sciences Division, Fred Hutchinson Cancer Center, 1100 Fairview Ave N, Seattle, WA 98109, USA. [5]Department of Urology, University of Washington, 1959 NE Pacific Street, Seattle, WA 98195, USA. [6]Department of Pathology and Laboratory Medicine, University of Washington, 1959 NE Pacific Street, Seattle, WA 98195, USA. [7]Clinical Research Division, Fred Hutchinson Cancer Center, 1100 Fairview Ave N, Seattle, WA 98109, USA. [8]Department of Hematology and Hematopoietic Cell Transplantation, City of Hope, 1500 East Duarte Road, Duarte, CA 91010, USA. [9]Department of Immuno-Oncology, Beckman Research Institute of City of Hope, 1500 East Duarte Road, Duarte, CA 91010, USA. [10]These authors contributed equally: Vipul Bhatia, Nikhil V. Kamat and Tiffany E. Pariva. ✉e-mail: j.ishihara@imperial.ac.uk; jklee5@fredhutch.org

(PSMA)-specific radioligand therapy (lutetium Lu 177 vipivotide tetraxetan)[6] have been approved for mCRPC. Each of these agents extends survival on average by several months but long-term remissions are rare.

Strategies to reprogram the immune system to combat prostate cancer first gained traction with the clinical approval of the dendritic cell vaccine sipuleucel-T for asymptomatic mCRPC[7]. More recently, several types of immunotherapies including immune checkpoint inhibitors, a DNA cancer vaccine, antibody-drug conjugates (ADC), T cell engaging bispecific antibodies (T-BsAb), and chimeric antigen receptor (CAR) T cell therapies have been under active clinical investigation[8,9]. CARs are synthetic receptors that leverage the potency, expansion, and memory of T cells and can be engineered against virtually any tumor-associated cell surface antigen. The adoptive transfer of CAR T cells has rapidly become an established treatment for hematologic malignancies with exceptional response rates leading to six clinical approvals in the last five years[10]. In contrast, CAR T cell therapies targeting solid tumors have lagged due to additional challenges related to the lack of bona fide tumor-specific antigens, inhospitable tumor microenvironments, and poor trafficking, persistence, and expansion of CAR T cells[11].

Despite the challenges observed in driving effective immune responses toward solid tumors, recent early phase clinical trials investigating CAR T cell therapies targeting PSMA in mCRPC have reported safety and evidence of significant biochemical and radiographic responses[12,13]. These preliminary results serve to embolden efforts to develop and optimize new CAR T cell therapies for prostate cancer. While PSMA is the preeminent target for therapeutic and diagnostic development in prostate cancer, recent work indicates that PSMA expression may be quite heterogeneous in mCRPC[14]. Tumor antigen heterogeneity, especially in the context of single antigen-targeted CAR T cell therapies for solid tumors like prostate cancer, is an important barrier to therapeutic efficacy[15]. Thus, identifying cell surface antigens with broad and relatively homogeneous expression in prostate cancer is imperative. In addition, very few if any tumor-associated antigens demonstrate tumor-restricted expression—most also exhibit low level expression in normal tissues that could represent liabilities for CAR T cell therapies due to on-target off-tumor toxicities which can lead to devastating consequences including death[16].

We previously performed integrated transcriptomic and cell surface proteomic profiling of human prostate adenocarcinoma cell lines and identified six transmembrane epithelial antigen of the prostate 1 (STEAP1) as one of the most highly enriched cell surface antigens[17]. STEAP1 was first described over two decades ago[18] and was recognized as being highly expressed in prostate cancer. STEAP1 is strongly expressed in >80% of mCRPC with bone or lymph node involvement[19], 62% of Ewing sarcoma[20], and multiple other cancer types[21]. STEAP1 belongs to the STEAP family of metalloreductases that can form homotrimers or heterotrimers with other STEAP proteins[22]. STEAP1 has an established functional role in promoting cancer cell proliferation, invasion, and epithelial-to-mesenchymal transition[18,23–26]. Furthermore, STEAP1 demonstrates limited expression in normal tissue[27] which makes it a highly compelling target for cancer therapy.

Multiple immunotherapeutic agents have been developed to target STEAP1 yet none are clinically approved. The ADC vandortuzumab vedotin (DSTP3086S) consisting of a humanized anti-STEAP1 IgG1 antibody linked to monomethyl auristatin E was found to have an acceptable safety profile in a phase I clinical trial in mCRPC but few objective tumor responses were observed[28]. A T-BsAb incorporating two anti-STEAP1 fragment-antigen binding (Fab) domains, an anti-CD3 single chain variable fragment (scFv), and a fragment crystallizable (Fc) domain engineered to lack effector function called AMG 509 is currently being evaluated in a phase I clinical trial (NCT04221542) in mCRPC[29]. A symmetric dual bivalent T-BsAb called BC261 was also recently reported to demonstrate potent antitumor activity across

multiple preclinical models of prostate cancer and Ewing sarcoma[30]. In addition, a human leukocyte antigen (HLA) class I-restricted T cell receptor (TCR) specific for a STEAP1 peptide has been shown to inhibit local and metastatic Ewing sarcoma growth in a preclinical xenograft model after adoptive transfer of transgenic T cells[31].

In this study, we perform comparative analysis of the relative expression of STEAP1 and PSMA in lethal mCRPC to investigate the utility of targeting STEAP1 in the current era of PSMA theranostics. We engineer and screen second-generation STEAP1 CARs for antigen-specific T cell activation and target cell cytolysis to yield a lead candidate for further characterization. We determine the functional epitope specificity of STEAP1 CAR T cells and profile the expansion and immunophenotype of STEAP1 CAR T cell products from multiple donors. We then establish the potency and preliminary safety of STEAP1 CAR T cell therapy in relevant preclinical models of prostate cancer but observe recurrent loss of STEAP1 antigen expression as a mechanism of treatment resistance. To overcome this issue, we evaluate the concomitant administration of CBD-IL-12 which remodels the immunosuppressive tumor microenvironment of prostate cancer and engages endogenous immunity to broaden antitumor responses. Collectively, these studies provide strong rationale for the clinical translation of STEAP1 CAR T cell therapy to men with mCRPC and guide strategies to overcome potential mechanisms of therapeutic resistance.

## Results

### STEAP1 is broadly expressed in treatment-refractory mCRPC tissues

We first set out to determine the pattern and extent of STEAP1 expression relative to PSMA in advanced metastatic prostate cancer. We performed immunohistochemical (IHC) staining on a duplicate set of tissue microarrays consisting of 121 metastatic tumors (each with up to three cores represented) collected from 44 men with lethal mCRPC patients collected by rapid autopsy between the years 2010 and 2017 through the University of Washington Tumor Acquisition Necropsy Program[32] (Fig. 1a). Plasma membrane staining for STEAP1 and PSMA in each tissue was scored by a research pathologist and semiquantitative H-scores were determined based on the staining intensity (Supplementary Fig. 1a) multiplied by the percentage of cancer cells staining at each intensity (Supplementary Fig. 1b). By implementing a minimal staining threshold with an H-score cut-off of 30, we found that 87.7% of evaluable matched mCRPC tissues (100 of 114) demonstrated staining for STEAP1 compared to only 60.5% (69 of 114) for PSMA (Fig. 1b). In addition, 28.1% of mCRPC tissues (32 of 114) showed STEAP1 but not PSMA staining (Fig. 1b, c) whereas only 0.9% (one of 114) exhibited PSMA but not STEAP1 staining. Based on these results, we used a linear mixed statistical model to determine that the odds of non-zero (H-score >0) staining was 22-fold (95% CI 6-173) higher for STEAP1 than for PSMA and the odds of an H-score ≥30 are 84-fold (95% CI 30-317) higher for STEAP1 than for PSMA. The mean STEAP1 H-score in bone (193; 95% CI 171 to 215) was significantly higher than in lymph node metastases (difference −48; 95% CI −21 to −76; $p < 0.001$) and significantly higher than in visceral metastases (difference −59; 95% CI −42 to −77; $p < 0.001$). There was no significant difference between the mean STEAP1 H-score in lymph node metastasis compared to visceral metastases (difference 11; 95% CI −16 to 39; $p = 0.4$) (Supplementary Fig. 1c). We also observed several cases with heterogeneous expression of PSMA within cores (Fig. 1d) which is consistent with a recent report of intratumoral PSMA heterogeneity in mCRPC biopsies[14].

Patient-level analysis using a mean H-score threshold of ≥30 and McNemar's test revealed that 95% of evaluable patients (42 of 44) had tumors with STEAP1 expression while 68% (30 of 44) were positive for PSMA (Supplementary Fig. 1d). To study the patterns of inter-patient and intra-patient heterogeneity associated with STEAP1 and PSMA expression, we used STEAP1 and PSMA H-scores to evaluate the hypergeometric, Simpson, and Shannon diversity scores. We observed

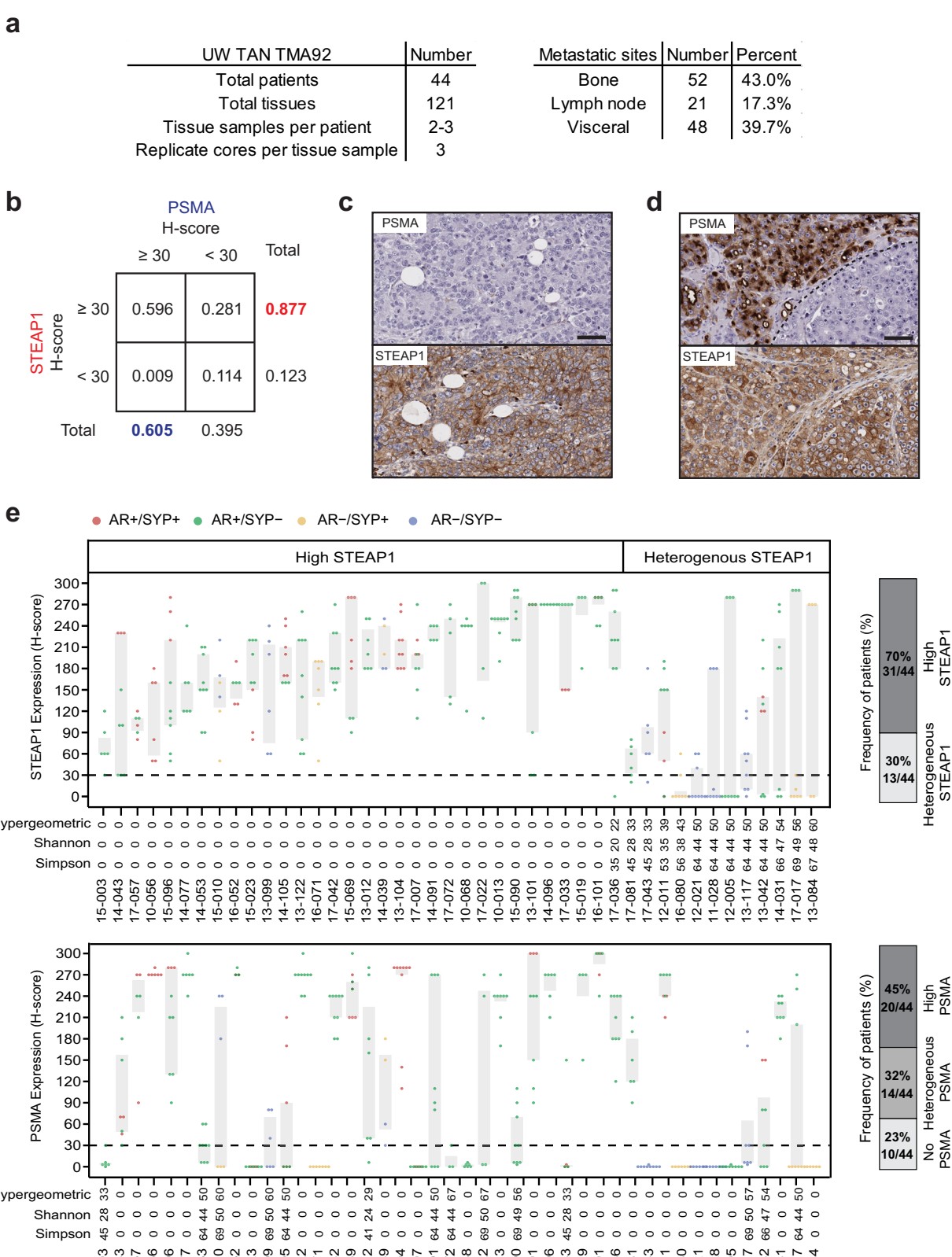

**a**

| UW TAN TMA92 | Number | | Metastatic sites | Number | Percent |
|---|---|---|---|---|---|
| Total patients | 44 | | Bone | 52 | 43.0% |
| Total tissues | 121 | | Lymph node | 21 | 17.3% |
| Tissue samples per patient | 2-3 | | Visceral | 48 | 39.7% |
| Replicate cores per tissue sample | 3 | | | | |

two patterns of STEAP1 expression (Fig. 1e) with 68% (30/44) of patients showing STEAP1 expression across all metastatic sites (high STEAP1) and 32% (14/44) patients showing metastatic sites with and without STEAP1 expression (heterogenous STEAP1). No patients were identified in which all metastatic tissues lacked STEAP1 expression. A similar analysis for PSMA expression in the same cohort revealed 45%

(20/44) patients with high PSMA expression, 32% (14/44) with heterogeneous PSMA expression, and 23% (10/44) with no PSMA expression. Based on molecular subclassification of mCRPC tissues using AR and the neuroendocrine marker synaptophysin (SYP) expression assessed by IHC, most patients with high or heterogeneous STEAP1 and PSMA expression had AR-positive prostate cancer (AR+/SYP− or

**Fig. 1 | Comparative analysis of STEAP1 and PSMA in lethal, metastatic castration-resistant prostate cancer (mCRPC). a** Characteristics of the mCRPC tissues represented on University of Washington Tissue Acquisition Necropsy Tissue Microarray 92 (UW TAN TMA92). **b** Contingency table showing the frequency of mCRPC tissues with STEAP1 or PSMA IHC staining above or below an H-score threshold of 30. Micrographs of select mCRPC tissues after STEAP1 and PSMA IHC staining to highlight the (**c**) absence of PSMA but presence of STEAP1 expression and (**d**) intratumoral heterogeneity of PSMA expression but not STEAP1. Scale bars = 50 μm. For panels (**c**, **d**) *n* = 332 mCRPC cores were immunostained for STEAP1 and PSMA. **e** Dot and box plot showing the distribution of STEAP1 (top) and PSMA (bottom) H-scores in 44 patients from the UW TAN TMA92 cohort. Each dot represents a tumor specimen/core (*n* = 319 cores for PSMA and 333 cores for STEAP1) and the color indicates the molecular subtype: AR+/SYP+ (red), AR+/SYP− (green), AR−/SYP+ (yellow) and AR−/SYP− (purple). Gray rectangles show interquartile ranges spanning the 25th to the 75th percentiles of PSMA H-scores from each patient. Bar plots (on the right) summarize the frequencies of patients classified based on STEAP1 and PSMA expression as no expression (all cores with H-score ≤30, light grey), heterogeneous expression (at least one core with H-score ≤30 and H-score >30, mid grey) and high expression (all cores with H-scores >30, dark grey). Source data are provided in the Source Data file.

AR+/SYP+) while those with no PSMA expression had AR-null prostate cancer (AR-/SYP+ or AR-/SYP-). We identified a positive correlation between the expression of STEAP1 and AR ($p < 0.001$) by a fitted linear mixed model with random effect in cases represented on the tissue microarray (Supplementary Fig. 2a, b) which was expected given that *STEAP1* is an androgen-regulated gene[33,34]. In contrast, a negative trend was appreciated between the expression of STEAP1 and SYP (Supplementary Fig. 2c). These findings suggest that, like PSMA[35], STEAP1 expression may be lost with neuroendocrine transdifferentiation of prostate cancer.

## Development of a potent, antigen-specific STEAP1 CAR
Given the widespread expression of STEAP1 in late-stage mCRPC and its reported functional role in cancer progression[27,36,37], we next started to engineer a lentiviral STEAP1-specific second-generation CAR. We used the pCCL-c-MNDU3-X lentiviral backbone[38] which has been widely used for hematopoietic stem cell gene therapy[39] and CAR expression in T cells driven by the internal MNDU3 promoter has been shown to be higher than that achieved with an EFS promoter[40]. A 4-1BB costimulatory domain was favored due to its association with T cell memory formation and prolonged persistence[41] and a CD28 transmembrane domain was introduced as this has been shown to reduce the antigen threshold for second-generation 4-1BB CAR T cell activation[42]. We incorporated the fully humanized scFv derived from vandortuzumab vedotin, an ADC targeting STEAP1 whose development was discontinued after a phase I clinical trial[28]. This scFv is a humanized variant of the murine monoclonal antibody (mAb 120.545) originally developed by Agensys, Inc. that demonstrates 1 nM affinity in cell-based binding assays[43]. To potentially tune CAR activity, we implemented three different hinge/spacer lengths including short (IgG4 hinge), medium (IgG4 hinge-CH3), and long (IgG4 hinge-CH2-CH3). The long spacer was engineered with previously described 4/2-NQ mutations[44] in the CH2 domain to prevent Fc-gamma receptor binding and activation-induced cell death that occurs with the adoptive transfer of long spacer CAR T cells into immunodeficient mice. The three candidate CARs were cloned into the lentiviral vector (Fig. 2a) that also co-expresses truncated epidermal growth factor receptor (EGFRt) as a transduction marker. Lentiviruses were generated and used to transduce human CD4 and CD8 T cells enriched from human donor peripheral blood mononuclear cells (PBMCs) collected from pheresis. Expanded CD4 and CD8 CAR T cells were immunophenotyped (Supplementary Fig. 3a) and reconstituted into cell products of a defined composition with a normal CD4/CD8 ratio to evaluate their functional activities.

To control for STEAP1 expression in an isogenic manner, we focused on the 22Rv1 human prostate cancer cell line that demonstrates native STEAP1 expression and performed STEAP1 knockout (ko) by CRISPR/Cas9 genome editing. We then generated a STEAP1 rescue line from the 22Rv1 STEAP1 ko by transduction with a STEAP1 expressing lentivirus (Fig. 2b). These lines were then used to screen the three short, medium, and long spacer STEAP1 CAR T cells in co-culture assays with a readout of interferon-gamma (IFN-γ) release as an indicator of T cell activation. Only the long spacer STEAP1 CAR T cells (hereafter called STEAP1-BBζ CAR T cells) demonstrated the anticipated antigen-specific pattern of IFN-γ release, while the short and medium spacer STEAP1 CAR T cells did not (Fig. 2c, Supplementary Fig. 3b, c). Further, STEAP1-BBζ CAR T cells showed substantial dose-dependent cytolysis of 22Rv1 cells compared to untransduced T cells (Fig. 2d) and demonstrated relative sparing of 22Rv1 STEAP1 ko cells (Fig. 2e). Similar studies were then performed in the DU145 human prostate cancer cell line that lacks native STEAP1 expression but was engineered to express STEAP1 (DU145 STEAP1) by lentiviral transduction (Supplementary Fig. 4a). In this setting, STEAP1-BBζ CAR T cell activation was only observed in co-cultures with DU145 STEAP1 cells and not the parental DU145 cells (Supplementary Fig. 4b). Cytolytic activity was only appreciated with STEAP1-BBζ CAR T cells and not untransduced T cells in co-cultures with DU145 STEAP1 cells (Supplementary Fig. 4c).

We subsequently analyzed a larger panel of human prostate cancer cell lines to characterize their native STEAP1 expression by immunoblot analysis. The cell lines with known AR expression/activity (LNCaP, 22Rv1, VCaP, and LNCaP95) showed varying levels of STEAP1 expression while the AR-null cell lines (PC3, DU145, MSKCC EF1, and NCI-H660) did not appear to express detectable levels of STEAP1 (Fig. 2f). We proceeded to perform co-cultures of STEAP1-BBζ CAR T with these lines to further validate their antigen-specific activation based on IFN-γ release (Fig. 2g). However, we observed a discordant finding in that the PC3 line, which showed no apparent STEAP1 protein expression (Fig. 2f), induced substantial activation of STEAP1-BBζ CAR T cells. Prior literature suggested that STEAP1 is expressed in the PC3 cell line at low levels[45]. Indeed, prolonged immunoblot exposure revealed a band suggesting the presence of very low expression of STEAP1 (Fig. 2h). To confirm whether the STEAP1-BBζ CAR T cell activation was due to this minor STEAP1 expression in PC3 cells, we generated three PC3 STEAP1 ko sublines (Fig. 2h) and again performed co-cultures with STEAP1-BBζ CAR T cells. STEAP1 ko in the PC3 line led to the abrogation of STEAP1-BBζ CAR T cell activation (Fig. 2i), further validating specificity and providing evidence of the sensitivity of STEAP1-BBζ CAR T cells to low antigen density conditions.

## Lack of cross-reactivity of STEAP1-BBζ CAR with mouse Steap1 and human STEAP1B
Consistent with the anti-human specificity of vandortuzumab vedotin, STEAP1-BBζ CAR T cells did not demonstrate cross reactivity with mouse Steap1 (Supplementary Fig. 4a, d, e). However, we used this as an opportunity to individually reconstitute the three human STEAP1 extracellular domains (ECDs) onto mouse Steap1 (Supplementary Fig. 4f) to determine which ECDs are critical for epitope recognition by STEAP1-BBζ CAR T cells. Co-culture experiments were performed with STEAP1-BBζ CAR T cells and DU145 cells engineered to express mouse Steap1 with individual replacement of mouse ECDs with human ECDs. We found that human STEAP1 ECD2 but not ECD1 or ECD3 was associated with STEAP1-BBζ CAR T cell activation (Supplementary Fig. 4g). Interestingly, the human STEAP1 and mouse Steap1 ECD2 demonstrate 93.9% (31/33 amino acids) homology (Supplementary Fig. 4h), indicating that Q198 and/or I209 of human STEAP1 are critical to productive recognition by STEAP1-BBζ CAR T cells. Q198 has been shown

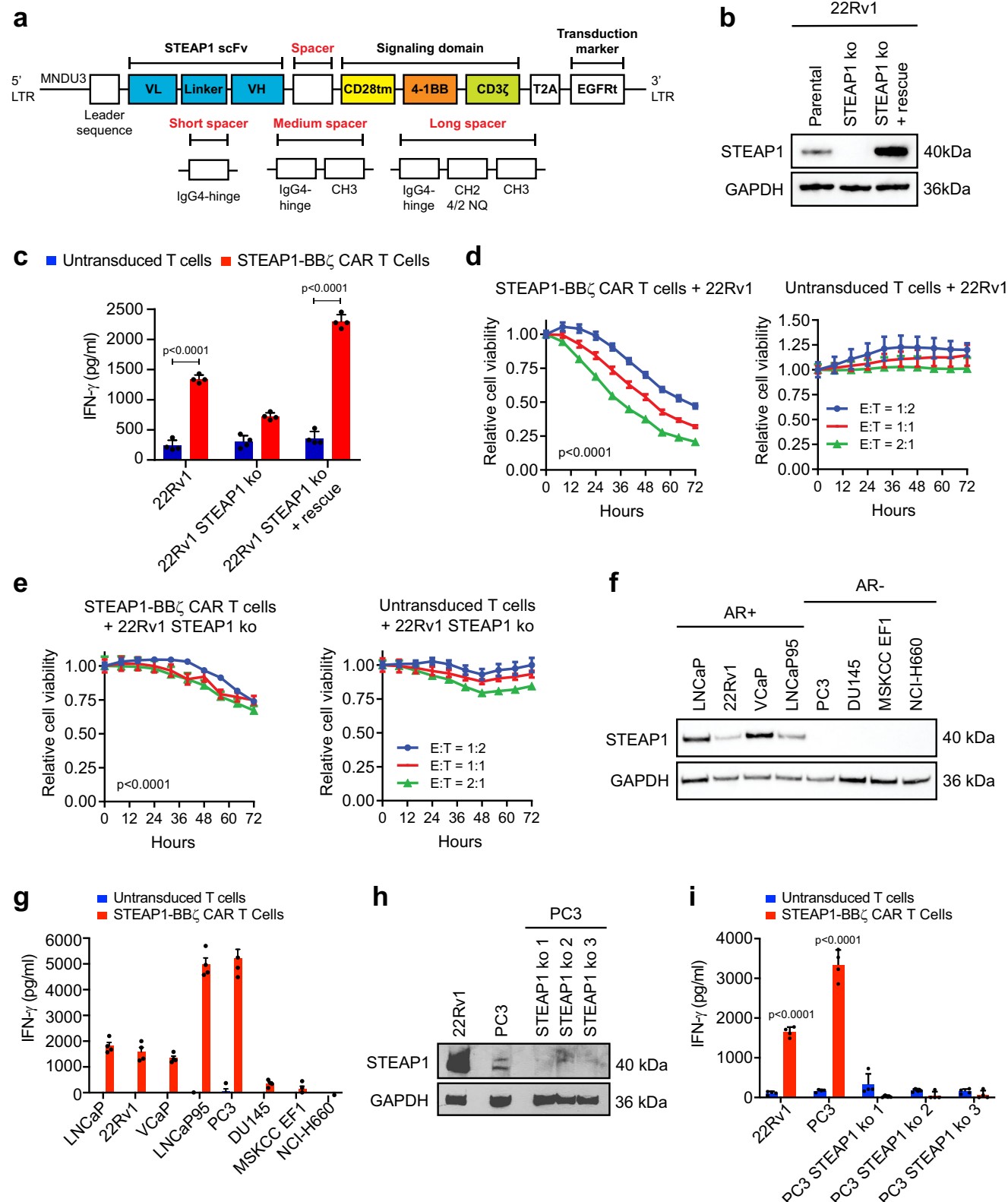

to interact with the Fab of 120.545 as part of an interaction hotspot based on a recent structure resolved by cryogenic electron microscropy[22].

Of the human STEAP family of proteins, STEAP1B has the greatest homology to STEAP1[45]. Three STEAP1B transcripts have been identified, of which all demonstrate complete conservation of the amino acid sequence of human STEAP1 ECD2 (Supplementary Fig. 5a). The consensus membrane topology prediction algorithm TOPCONS[46] predicted ECD2 domain sequences as being extracellular in the three STEAP1B protein isoforms (Supplementary Fig. 5b) albeit with low reliability scores for STEAP1B as compared to hSTEAP1 due to a lack of consensus between the five topology prediction models (OCTOPUS, Philius, PolyPhobius, SCAMPI, and SPOCTOPUS) used by TOPCONS (Supplementary Fig. 5c). Prior analysis using another in silico hidden

**Fig. 2 | Screening second-generation 4-1BB chimeric antigen receptors (CARs) to identify a lead for STEAP1 CAR T cell therapy. a** Schematic of the lentiviral STEAP1 CAR construct and variation based on short, medium, and long spacers. LTR long terminal repeat, MNDU3 Moloney murine leukemia virus U3 region, scFv single-chain variable fragment, VL variable light chain, VH variable heavy chain, tm transmembrane, EGFRt truncated epidermal growth factor receptor, 4/2 NQ = CH2 domain mutations to prevent binding to Fc-gamma receptors. **b** Immunoblots of STEAP1 in 22Rv1 parental cells, STEAP1 knockout (ko) cells, and STEAP1 ko cells with rescue of STEAP1. **c** IFN-γ enzyme-linked immunosorbent assay (ELISA) results from co-cultures of either untransduced T cells or STEAP1-BBζ CAR T cells with each of the 22Rv1 sublines at a 1:1 ratio at 24 h ($p < 0.001$). Relative cell viability of (**d**) 22Rv1 and (**e**) 22Rv1 STEAP1 ko target cells over time measured by fluorescence live cell imaging upon co-culture with (left) STEAP1-BBζ CAR T cells ($p < 0.001$) or (right) untransduced T cells at variable effector-to-target (E:T) cell ratios. **f** Immunoblots

demonstrating expression of STEAP1 in androgen receptor (AR)-positive human prostate cancer cell lines but not AR-negative prostate cancer cell lines. **g** IFN-γ quantification by ELISA from co-cultures of either untransduced T cells or STEAP1-BBζ CAR T cells with each of the human prostate cancer cell lines in (**f**) at a 1:1 ratio at 24 h. **h** Immunoblots for STEAP1 in 22Rv1, PC3, and PC3 STEAP1 ko sublines. **i** IFN-γ quantification by ELISA from co-cultures of either untransduced T cells or STEAP1-BBζ CAR T cells with each cell line in (**h**) at a 1:1 ratio at 24 h ($p < 0.001$). For panels (**c–e, g** and **i**) $n = 4$ biological replicates per conditions were used and error bars represent mean with SEM. Panel (**b, f, h**) displays results representative of $n = 3$ biological replicates. GAPDH was used as a protein loading control. For panel (**c**) and (**i**), two-way ANOVA with Sidak's multiple comparison test was used. For panels (**d**) and (**e**), two-way ANOVA with Tukey's multiple comparisons test was used. Source data are provided in the Source Data file.

Markov model-based topology prediction tool TMHMM[47] had also suggested that this sequence could be intracellular rather than extracellular in STEAP1B protein isoforms 1 and 2[45]. However, the crystal structure of STEAP1B has not yet been determined to directly substantiate these predictions. To functionally evaluate whether STEAP1-BBζ CAR T cells might also be reactive against STEAP1B, we performed co-cultures using DU145 lines engineered to express each of the three isoforms of STEAP1B. We did not identify evidence of STEAP1-BBζ CAR T cell activation (Supplementary Fig. 5d), suggesting that the STEAP1 epitope recognized by STEAP1-BBζ CAR T cells may not be presented as part of an ectodomain by STEAP1B despite apparent sequence homology.

## Characterization of STEAP1-BBζ CAR T cell products across a series of donors

We next profiled the expansion, transduction efficiency, and immunophenotype of STEAP1-BBζ CAR T cell products using three independent sets of peripheral blood mononuclear cells (PBMCs) collected from healthy donors. We generally observed a 20- to 40-fold expansion of STEAP1-BBζ CAR T cells within 11 days of culture (Supplementary Fig. 6a). The percentage of EGFRt[+] CD8 T cells ranged from 24.3 to 54.2% while the percentage of EGFRt[+] CD4 T cells was higher and ranged from 60.1 to 74.9% in our STEAP1-BBζ CAR T cell products (Supplementary Fig. 6b). We examined the expression of the T cell exhaustion markers PD-1 and LAG-3 in the untransduced and STEAP1-BBζ CAR T cell subsets and observed no significant increase in expression (Supplementary Fig. 6c). This finding suggested low or absent tonic signaling by the STEAP1-BBζ CAR which was encouraging as constitutive CAR signaling can negatively impact CAR T cell effector function[48].

Both stem cell memory T cell (Tscm) and central memory T cell (Tcm) phenotypes have been associated with the therapeutic efficacy of CAR T cell therapy as they promote sustained proliferation and persistence in vivo[49–51]. Immunophenotyping of untransduced and STEAP1-BBζ CAR T cell subsets demonstrated higher frequencies of Tscm cells compared to the T cell subsets in donor PBMCs from which the cell products were derived (Supplementary Fig. 6d). This effect is likely due to the addition of IL-7 and/or IL-15 to the T cell expansion media as these cytokines have been shown to preserve and enhance Tscm differentiation[51,52]. Our analysis also revealed an enrichment in Tcm populations particularly in the CD8 STEAP1-BBζ CAR T cells (Supplementary Fig. 6e).

## STEAP1-BBζ CAR T cells demonstrate substantial antitumor effects in disseminated prostate cancer models with native STEAP1 expression established in immunodeficient mice

As an initial screen for in vivo antitumor activity, we established 22Rv1 subcutaneous xenograft tumors in male NOD scid gamma (NSG) mice. When tumors grew to approximately 100 mm³, mice were treated with a single intratumoral injection of either $5 \times 10^6$ untransduced

T cells or STEAP1-BBζ CAR T cells. Intratumoral treatment with STEAP1-BBζ CAR T cells was associated with significant tumor growth inhibition that was statistically significant by day 18 of treatment (Fig. 3a). Mice were sacrificed on day 25 and residual tumors from mice treated with STEAP1-BBζ CAR T cells showed large areas of necrotic debris and regions of viable tumor were infiltrated with CD3[+] STEAP1-BBζ CAR T cells (Supplementary Fig. 7a). STEAP1 expression was conserved in the tumors across treatment groups (Supplementary Fig. 7b).

We transduced 22Rv1 cells with lentivirus to enforce firefly luciferase (fLuc) expression and $10^6$ 22Rv1-fLuc cells were injected into the tail veins of male NSG mice. Metastatic colonization was visualized by live bioluminescence imaging (BLI) after two weeks, at which point mice were treated with a single intravenous injection of either $5 \times 10^6$ untransduced T cells or STEAP1-BBζ CAR T cells (Fig. 3b). Serial BLI revealed rapid disease progression in mice treated with untransduced T cells while those receiving STEAP1-BBζ CAR T cells demonstrated a significant delay in tumor progression (Fig. 3c, d) and extension of survival (97 days versus 31 days, $p = 0.0018$ by log-rank test, Fig. 3e). There was no significant difference in mouse weights between treatment arms (Supplementary Fig. 7c). IHC staining of tumors at the end of study showed a significant reduction in STEAP1 expression (Supplementary Fig. 7d), indicating that antigen escape was a mechanism of resistance. However, this was unlikely a result of transdifferentiation to a variant prostate cancer state as we did not appreciate morphologic changes, loss of AR and PSMA expression[53], or gain of SYP expression (Supplementary Fig. 7d).

To investigate the global impact of STEAP1 loss in prostate cancer, we performed transcriptome profiling of the isogenic 22Rv1 wildtype (wt), 22Rv1 STEAP1 ko, and 22Rv1 STEAP1 ko + rescue cell lines we had previously prepared (Fig. 2b). Differential gene expression analysis comparing 22Rv1 STEAP1 ko cells with 22Rv1 wt cells identified ~1700 genes significantly downregulated (FDR ≤ 0.05, fold-change<2) with STEAP1 knockout. Rescue of STEAP1 expression in the 22Rv1 STEAP1 ko cells revealed that ~600 genes were significantly upregulated (FDR ≤ 0.05, fold-change>2) by STEAP1 addback (Supplementary Fig. 8a). Gene Set Enrichment Analysis (GSEA) nominated several biological pathways that may be dysregulated by modulation of STEAP1 expression. Prominent among these were cell cycle progression and multiple metabolic processes including the Kreb cycle and glycolysis which were negatively enriched by STEAP1 knockout and rescued upon addback of STEAP1 (Supplementary Fig. 8b). We applied the validated 31-gene cell cycle progression (CCP) signature[54] to our data which showed a significant downregulation of the CCP signature associated with STEAP1 knockout with a score of −0.8 which increased substantially to 0.5 with rescue of STEAP1 expression (Supplementary Fig. 8c). These data are consistent with a prior publication indicating that knockdown of STEAP1 in the LNCaP prostate cancer cell line impairs cell viability and proliferation while inducing apoptosis[37].

We also noted that antigen processing and presentation was one of the most significantly de-enriched KEGG pathways with STEAP1

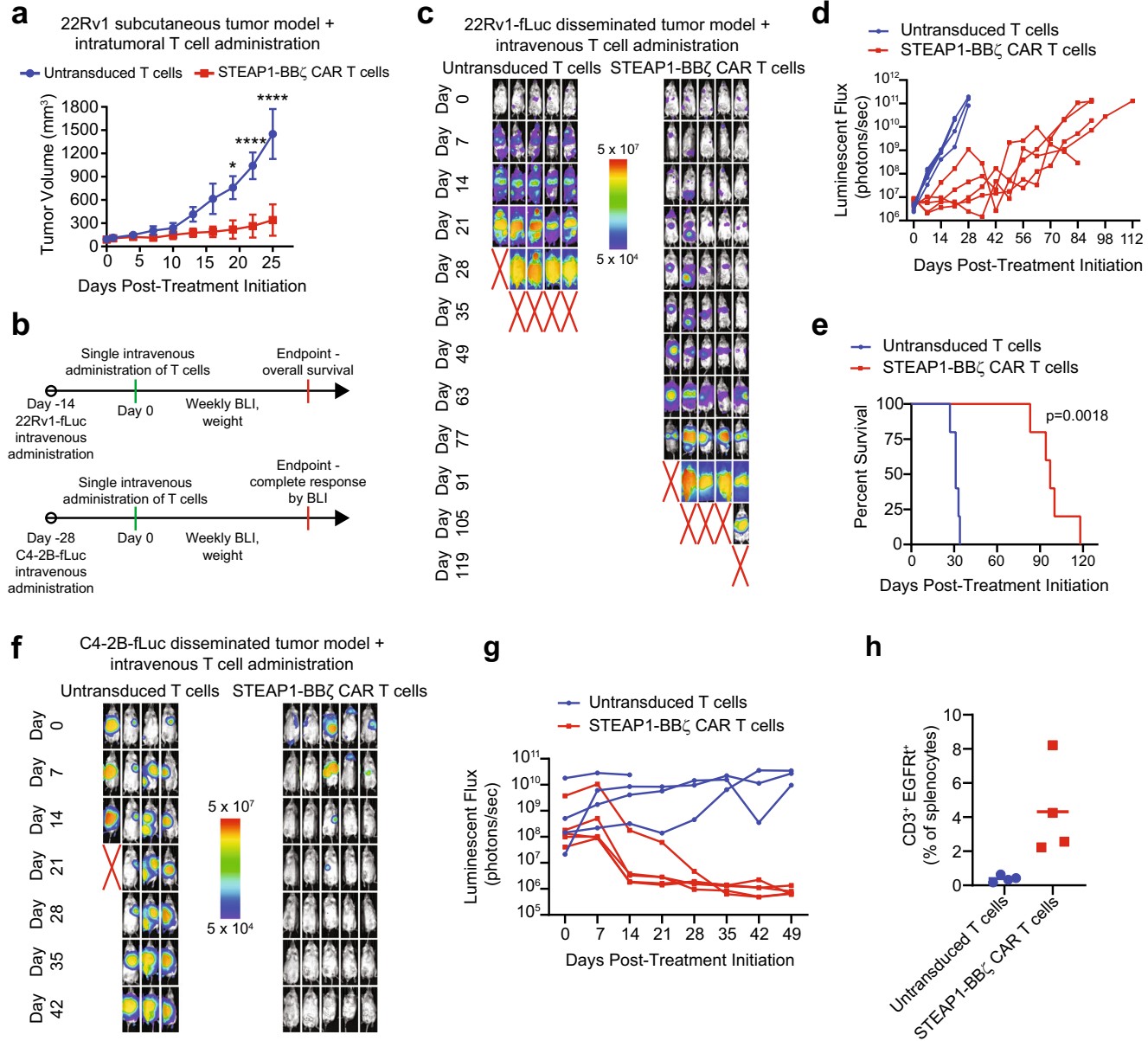

**Fig. 3 | In vivo antitumor activity of STEAP1-BBζ CAR T cell therapy in prostate cancer models with native STEAP1 expression. a** Volumes of 22Rv1 subcutaneous tumors in NSG mice ($n = 4$ for untransduced T cells group and $n = 5$ for STEAP1-BBζ CAR T cells group) over time after a single intratumoral injection of $5 \times 10^6$ untransduced T cells or STEAP1-BBζ CAR T cells at normal CD4/CD8 ratios. $p < 0.0001$ at day 20 and 25. Bars represent mean with SEM. **b** Schematic of tumor challenge experiments for 22Rv1 (top) and C4-2B (bottom) disseminated models. fLuc firefly luciferase, BLI bioluminescence imaging. **c** Serial live bioluminescence imaging (BLI) of NSG mice engrafted with 22Rv1-fLuc metastases and treated with a single intravenous injection of $5 \times 10^6$ untransduced T cells or STEAP1-BBζ CAR T cells at normal CD4/CD8 ratios on day 0. Red X denotes deceased mice. Radiance scale is shown. **d** Plot showing the quantification of total flux over time from live BLI of each mouse in (**c**). **e** Kaplan–Meier survival curves of mice in (**c**) with statistical

significance determined by log-rank (Mantel-Cox) test. For panels (**c**–**e**) $n = 5$ mice per condition were used. **f** Serial live BLI of NSG mice engrafted with C4-2B metastases and treated with a single intravenous injection of $5 \times 10^6$ untransduced T cells or STEAP1-BBζ CAR T cells at normal CD4/CD8 ratios on day 0. Red X denotes deceased mice. Radiance scale is shown. **g** Plot showing the quantification of total flux over time from live BLI of each mouse in (**f**). For panels (**f**, **g**) $n = 4$ mice were used in the untransduced T cells group and $n = 5$ mice in the STEAP1-BBζ CAR T cells group. **h** Quantification of CD3⁺EGFRt⁺ STEAP1-BBζ CAR T cells by flow cytometry from splenocytes of mice treated with STEAP1-BBζ CAR T cells ($n = 4$) at the end of experiment on day 49. Bars represent mean. For panel (**a**), two-way ANOVA with Sidak's multiple comparison test was used. Source data are provided in the Source Data file.

knockout (Supplementary Fig. 8b). We observed a significant down-regulation of genes including *PSME1* (proteasome activator subunit 1) which is a member of the immunoproteasome complex, *TAP1* (transporter 1, ATP binding cassette subfamily member) which is critical to the major histocompatibility complex (MHC) class I peptide-loading complex, and several MHC class I and II genes such as *MR1* (major histocompatibility complex, class I-related), *HLA-DQ-B1*, and *HLA-DQ-B2* (Supplementary Fig. 8d). We further investigated the tumors

collected from the 22Rv1 disseminated models treated with STEAP1-BBζ CAR T cell therapy that demonstrated antigen loss (Fig. 3c, Supplementary Fig. 7d) by transcriptome analysis. Differential gene expression analysis and subsequent GSEA comparing 22Rv1 metastatic tumors from mice treated with STEAP1-BBζ CAR T cell therapy to those treated with untransduced T cells showed negative enrichment of pathways involved in MHC, cytotoxic lymphocytes, and T cell activation (Supplementary Fig. 9a, b). We also specifically evaluated the

expression of MHC class I and II genes and observed their marked downregulation in 22Rv1 tumors treated with STEAP1-BBζ CAR T cell therapy (Supplementary Fig. 9c). This result was further substantiated by a signification reduction in HLA-A,B,C staining by IHC in these tumors (Supplementary Fig. 9d). The potential implication of these data is that treatment with STEAP1-BBζ CAR T cell therapy and resultant loss of STEAP1 tumor antigen expression in prostate cancer may result in further immunotherapy resistance through impaired antigen processing and presentation.

We also inoculated male NSG mice with C4-2B-fLuc cells by tail vein injection. C4-2B is a castration-resistant subline of LNCaP[55] with growth kinetics more in line with typical prostate cancer. Four weeks after injection, metastatic colonization was confirmed by BLI and mice were treated with a single intravenous injection of either $5 \times 10^6$ untransduced T cells or STEAP1-BBζ CAR T cells (Fig. 3b). Serial BLI showed a complete response in all mice who received STEAP1-BBζ CAR T cells within five weeks of treatment (Fig. 3f, g). We identified a trend of increased weight loss in the untransduced T cell treatment group (Supplementary Fig. 10a) but this was not statistically significant. Necropsy of mice treated with STEAP1-BBζ CAR T cells showed no macroscopic disease and ex vivo BLI of organs did not reveal any signal (Supplementary Fig. 10b), suggesting that these mice were likely cured. We identified peripheral persistence of STEAP1-BBζ CAR T cells at the end of the experiment based on the presence of detectable CD3[+]EGFRt[+] splenocytes (Fig. 3h).

## Mouse-in-mouse STEAP1 CAR T cell studies demonstrate antitumor therapeutic efficacy

The activation and cytolytic activity of STEAP1-BBζ CAR T cells observed in the very low STEAP1 antigen density (-1500 molecules/cell) context of the PC3 cell line (Fig. 2g–i, Supplementary Fig. 11a, b) and evidence of in vivo antitumor activity in a disseminated PC3-fLuc tumor model (Supplementary Fig. 11c–e) presented concerns about the potential for on-target off-tumor toxicities. To evaluate for potential toxicity in a tractable model organism, we generated a human STEAP1 knock-in (hSTEAP1-KI) mouse in which the human *STEAP1* gene was knocked into the mouse *Steap1* gene locus on the C57Bl/6 background (Fig. 4a). A mouse colony was established with genotyping performed by polymerase chain reaction (PCR) of tail DNA (Fig. 4b). Both homozygous and heterozygous hSTEAP1-KI mice exhibited no apparent phenotypic or reproductive abnormalities compared to wildtype littermates. A tissue survey for human STEAP1 expression based on quantitative reverse transcription PCR (qRT-PCR) was performed on male and female heterozygous hSTEAP1-KI (hSTEAP1-KI/+) mice and revealed greatest relative expression in the prostate, followed by the uterus and adrenal gland (Fig. 4c). Further in situ analysis by STEAP1 IHC of male hSTEAP1-KI/+ prostate and adrenal glands revealed human STEAP1 expression confined to luminal epithelial cells of the prostate (Fig. 4d) and expression in the adrenal cortex (Fig. 4e).

A murinized version of the STEAP1 CAR, called STEAP1-mBBζ CAR, in which the scFv and IgG4 hinge-CH2-CH3 spacer were retained but the CD28 transmembrane domain, 4-1BB costimulatory domain, and CD3ζ activation domain were replaced with their mouse orthologs was cloned into a gammaretroviral construct (Fig. 4f). In addition, the human EGFRt transduction marker was replaced with a truncated mouse CD19 (mCD19t) to minimize potential immunogenicity. We confirmed the efficient retroviral transduction of T cells enriched from mouse splenocytes (Fig. 4g) and demonstrated the capacity of mouse STEAP1-mBBζ CAR T cells to induce cytolysis of the RM9 mouse prostate cancer cell line[56] engineered to express human STEAP1 (RM9-hSTEAP1) by lentiviral transduction (Fig. 4h).

The in vivo efficacy of mouse STEAP1-mBBζ CAR T cells was validated in a disseminated RM9-STEAP1-fLuc tumor model in NSG mice (Supplementary Fig. 12a). One week after tail vein injection of RM9-STEAP1-fLuc cells, mice were treated with either $5 \times 10^6$ untransduced mouse T cells or mouse STEAP1-mBBζ CAR T cells by tail vein injection. Mice that received untransduced mouse T cells demonstrated unchecked disease progression, whereas those treated with STEAP1-mBBζ CAR T cells uniformly exhibited rapid disease regression which was followed by subsequent relapse ten days later (Supplementary Fig. 12b, c). STEAP1-mBBζ CAR T cell therapy was associated with a statistically significant survival benefit (22 days versus 12 days, $p = 0.0039$ by log-rank test, Supplementary Fig. 12d). Weight loss was evident in both treatment groups as tumor burden increased prior to death (Supplementary Fig. 12e, f). Analysis of mouse splenocytes collected at necropsy showed peripheral persistence of STEAP1-mBBζ CAR T cells with the detection of mCD3[+] mCD19t[+] cells up to 24 days after adoptive transfer (Supplementary Fig. 12g). Lungs were harvested from mice in both treatment groups and STEAP1 IHC showed loss of STEAP1 expression in pulmonary metastases from mice treated with STEAP1-mBBζ CAR T cells (Supplementary Fig. 12h).

We subsequently expanded clonal RM9-STEAP1-fLuc lines to determine whether the observed tumor antigen escape could be a result of pre-existing heterogeneity in STEAP1 expression. The experiment was repeated with a clonal, disseminated RM9-STEAP1-fLuc tumor model in NSG mice (Supplementary Fig. 13a). In this context, mice treated with STEAP1-mBBζ CAR T cells demonstrated a prompt and durable complete response (Supplementary Fig. 13b–d). These findings highlight the potency of STEAP1-mBBζ CAR T cells in eradicating STEAP1[+] prostate cancer and further suggest that adjunct therapeutic strategies may be needed to overcome resistance in subgroups of advanced prostate cancer patients where heterogeneity of STEAP1 expression is present (Fig. 1e).

## STEAP1 CAR T cell therapy is safe in a humanized STEAP1 mouse model

To investigate both the preclinical safety and efficacy of STEAP1-mBBζ CAR T cell therapy, we inoculated male heterozygous hSTEAP1-KI mice with syngeneic, non-clonal RM9-STEAP1-fLuc cells by tail vein injection (Fig. 5a). After confirmation of metastatic colonization by BLI about a week later, mice received pre-conditioning cyclophosphamide 100 mg/kg by intraperitoneal injection[57]. A day later, mice were randomized to treatment with either $5 \times 10^6$ untransduced mouse T cells or mouse STEAP1-mBBζ CAR T cells by tail vein injection. All mice that received mouse STEAP1-mBBζ CAR T cells demonstrated a decrease in tumor burden within the first week of treatment initiation based on BLI (Fig. 5b, c). The observed response was short-lived but led to a modest extension of survival (21 days versus 12 days, $p = 0.0138$ by log-rank test, Fig. 5d)–similar to findings from the non-clonal RM9-STEAP1-fLuc experiments in NSG mice (Supplementary Figure 12d). There were no gross toxicities or premature deaths specifically associated with mouse STEAP1-mBBζ CAR T cell therapy at this dose level where clear evidence of antitumor efficacy was observed. Weight loss was associated with increased tumor burden but common to both treatment arms (Fig. 5e, f).

To further assess potential toxicity of STEAP1-mBBζ CAR T cell therapy, a similar experiment was performed in parallel in heterozygous hSTEAP1-KI mice bearing RM9-hSTEAP1 tumors and no tumors. Non-tumor bearing mice treated with untransduced T cells or STEAP1-mBBζ CAR T cells did not demonstrate any differences in survival (Supplementary Fig. 14a) or gross toxicities including loss of body weight (Supplementary Fig. 14b). Because the STEAP1-BBζ CAR consists of a modified IgG4 spacer which might potentially be immunogenic, we evaluated for a mouse anti-human antibody (MAHA) response by collecting retroorbital bleeds from mice in this experiment. No anti-human IgG and IgM antibodies were detected in the sera of mice at day 8 after treatment with STEAP1-BBζ CAR T cells (Supplementary Fig. 14c).

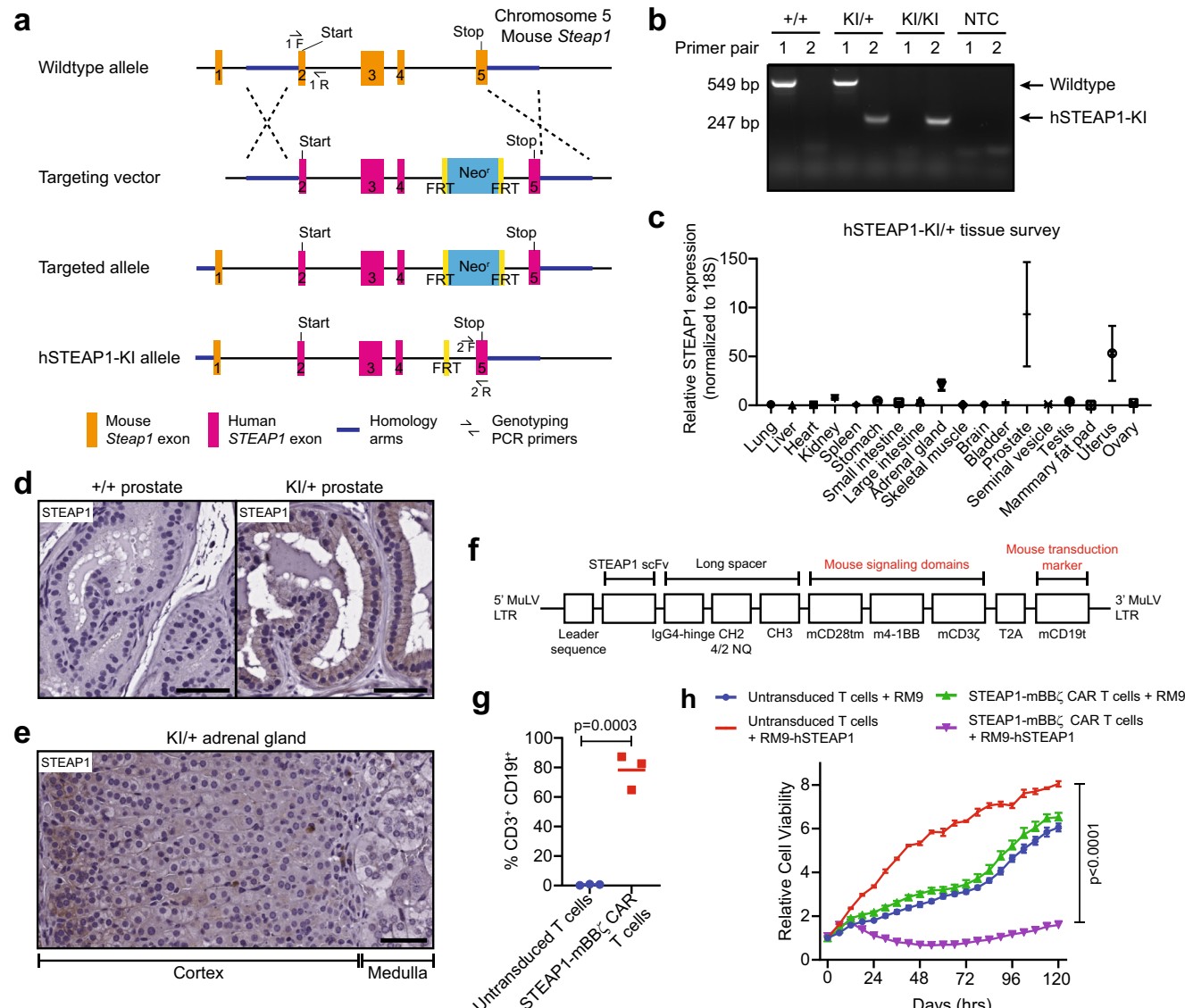

**Fig. 4 | Establishing a mouse-in-mouse system with a human STEAP1 knock-in (hSTEAP1-KI) mouse model and murinized STEAP1 CAR. a** Schematic showing the homologous recombination strategy using a targeting vector to knock-in human *STEAP1* exons 2–5 into the mouse *Steap1* locus on the C57Bl/6 background. FRT Flippase recognition target. **b** Visualization of PCR products from tail tip genotyping of wildtype (+/+), heterozygous (KI/+), or homozygous (KI/KI) mice using primer pairs intended to amplify portions of wildtype or hSTEAP1-KI alleles. NTC null template control. Representative gel image from *n* = 3 biologically independent experiments. **c** qPCR for human STEAP1 expression normalized to 18 S expression in a survey of tissues from hSTEAP1-KI/+ mice. *n* = 3 for sex-specific organs and *n* = 6 for common organs. Bars represent mean with SEM. Photomicrographs of STEAP1 IHC staining of (**d**) prostate tissues from (left) +/+ and (right) KI/+ mice and (**e**) an adrenal gland from a KI/+ mouse. Scale bars = 50 μm. For panel (**d**, **e**) STEAP1 immunostaining was performed on *n* = 3 biologically independent specimens. **f** Schematic of the retroviral murinized STEAP1 CAR construct. MuLV murine leukemia virus, mCD19t mouse truncated CD19. **g** Quantification of the efficiency of retroviral transduction of activated mouse T cells from three independent experiments based on the frequency of mouse CD3⁺CD19t⁺ cells by flow cytometry (*p* = 0.0003). **h** Relative cell viability of RM9 or RM9-hSTEAP1 target cells over time measured by fluorescence live cell imaging upon co-culture at a 1:1 ratio with mouse STEAP1-mBBζ CAR T cells or untransduced T cells (*p* < 0.0001). *n* = 4 biological replicates per condition. Error bars represent mean with SEM. For panel (**g**), unpaired two-tailed Student's *t* test with Welch's correction was used. In panel (**h**), two-way ANOVA with Sidak's multiple comparison test was used. Source data are provided in the Source Data file.

Importantly, heterozygous hSTEAP1-KI mice treated with STEAP1-mBBζ CAR T cells demonstrated no obvious tissue disruption or increased infiltration of CD3⁺ T cells in the prostate (Supplementary Fig. 15a, b) or adrenal gland (Supplementary Fig. 15c, d) relative to their counterparts treated with untransduced T cells, suggesting the absence of on-target off-tumor toxicities. Lungs collected at the end of the experiment showed human STEAP1 expression in RM9-hSTEAP1 pulmonary metastases with regional heterogeneity in mice treated with untransduced mouse T cells (Fig. 5g). On the other hand, tumors from mice treated with mouse

STEAP1-mBBζ CAR T cells again demonstrated an absence of human STEAP1 expression (Fig. 5h). To evaluate whether mouse STEAP1-mBBζ CAR T cell therapy and human STEAP1 antigen loss may also impact antigen presentation in the RM9-hSTEAP1 tumors, we performed IHC for murine beta-2-microglobulin (B2m) which is a key component of MHC class I molecules. We observed a significant downregulation of B2m expression in progressive tumors after treatment with mouse STEAP1-mBBζ CAR T cells compared to untransduced T cells (Supplementary Fig. 15e, f), consistent with our findings in the 22Rv1 model.

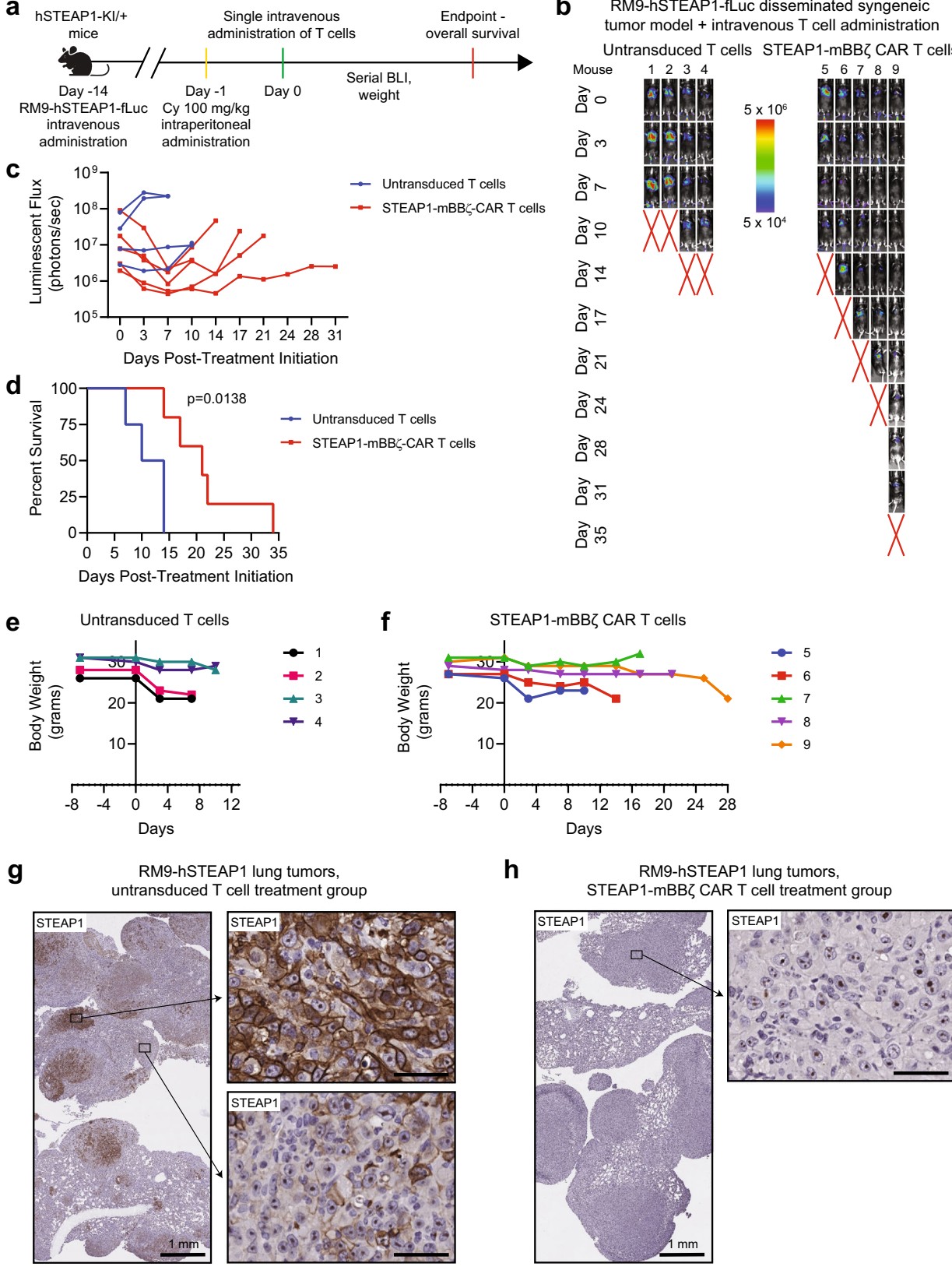

Collagen binding domain-IL-12 fusion cytokine elicits antitumor responses through enhanced T cell receptor signaling and antigen presentation

IL-12 is a heterodimeric cytokine consisting of p40 and p35 subunits that regulates T cell responses and leads to the production of IFN-γ. Striking antitumor responses associated with systemic IL-12 administration have been demonstrated in several preclinical models[58,59] but translation of this therapeutic approach to the clinic stalled due to dose-limiting toxicities and inefficacy[60–63]. Alternative strategies to circumvent these issues have been aimed at localizing IL-12 to tumor, either through intratumoral delivery or engineering IL-12 fusion proteins to leverage unique properties of the tumor

**Fig. 5 | Determination of the efficacy and safety of mouse STEAP1-mBBζ CAR T cells in hSTEAP1-KI mice bearing syngeneic, disseminated prostate cancer.** **a** Schematic of the tumor challenge experiment for the RM9-hSTEAP1 disseminated model in hSTEAP1-KI/ + mice. Cy cyclophosphamide (for preconditioning). Created with BioRender.com. **b** Serial live BLI of hSTEAP1-KI/ + mice engrafted with RM9-hSTEAP1-fLuc metastases and treated with a single intravenous injection of $5 \times 10^6$ mouse untransduced T cells or STEAP1-mBBζ CAR T cells on day 0. Red X denotes deceased mice. Radiance scale is shown. **c** Plot showing the quantification of total flux over time from live BLI of each mouse in (**b**). **d** Kaplan–Meier survival curves of mice in (**b**) with statistical significance determined by log-rank (Mantel–Cox) test. For panels (**b**–**d**), $n = 4$ mice were used in the untransduced T cells group and $n = 5$ mice in the STEAP1-BBζ CAR T cells group. Plots of weights for each mouse (numbered in (**b**)) over time in the (**e**) mouse untransduced T cells treatment group and (**f**) STEAP1-mBBζ CAR T cell treatment group. **g** Photomicrographs at (left) low and (right) high magnification of STEAP1 IHC staining of RM9-hSTEAP1 lung tumors after treatment with mouse untransduced T cells showing regions of strong homogenous STEAP1 expression and heterogeneous STEAP1 expression. Scale bars = 50 μm, unless otherwise noted. **h** Photomicrographs at (left) low and (right) high magnification of STEAP1 IHC staining of RM9-hSTEAP1 tumors after treatment with STEAP1-mBBζ CAR T cells showing no STEAP1 expression. Scale bars = 50 μm, unless otherwise noted. For panels (**g**, **h**) STEAP1 immunostaining was performed on $n = 4$ biologically independent specimens. Source data are provided in the Source Data file.

microenvironment. One recently described approach is fusion to the von Willebrand factor A3 domain which serves as a collagen binding domain (CBD, Fig. 6a) and enables binding of fusion proteins to exposed collagen in disordered tumor vasculature[64]. Systemic CBD-IL-12 therapy was shown to remodel the tumor microenvironments of immunologically "cold" murine breast cancer and melanoma models through enhanced IFN-γ signaling and to cooperate with anti-PD-1 immune checkpoint inhibition to induce tumor eradication[65].

We asked whether CBD-IL-12 could be effective in converting prostate cancer from "cold" to "hot" and inducing antitumor responses. Subcutaneous, syngeneic RM9 and Myc-CaP tumors were established in male C57Bl/6 and FVB mice, respectively, and mice were randomized to treatment with vehicle, anti-PD-1, or CBD-IL-12. Systemic CBD-IL-12 administration induced significant tumor growth inhibition in both syngeneic RM9 and Myc-CaP tumor models which were otherwise poorly responsive to anti-PD-1 therapy (Fig. 6b, Supplementary Fig. 16a). To gain further insight into the mechanisms of action of CBD-IL-12, we performed single-cell RNA-seq (scRNA-seq) analysis of RM9 tumors from mice treated with either vehicle or CBD-IL-12. Uniform Manifold Approximation and Projection plots revealed a substantial reduction in tumor epithelial, fibroblast, and endothelial cell compartments (Supplementary Fig. 16b) consistent with antitumor activity.

Marker-based profiling of innate and adaptive immune cell subsets (Fig. 6c,d) in the scRNA-seq data identified a substantial increase in CD8$^+$ T cells (7.55 vs. 0.76%), antigen cross-presenting cells including XCR1$^+$IRF8$^+$ conventional type 1 dendritic cells (cDC1, 1.33 vs. 0%), CD86$^+$INOS2$^+$ M1 polarized macrophages (2.26 vs 0%), and CD64$^+$F4/80$^+$ monocytes/macrophages (29.2 vs 4.43%) in tumors from the CBD-IL-12 treatment group compared to the vehicle control group. A reduction in immunosuppressive CD163$^+$CD206$^+$ M2 polarized macrophages (0 vs. 0.25%) and Ly6G+ neutrophils (0.52 vs. 1.08%) was also associated with CBD-IL-12 therapy. IHC analysis confirmed that RM9 tumors treated with vehicle generally lacked CD8$^+$ T cells while those treated with CBD-IL-12 showed a visibly significant infiltration of CD8$^+$ T cells (Supplementary Figure 16c). Tumor cells demonstrated enriched expression of the proteasome and immunoproteasome subunits *Psmb8* and *Psmb9* and major histocompatibility complex class I (MHC I) genes *H2-K1*, *H2-D1*, and *B2m*, consistent with upregulation of antigen processing and presentation machinery (Fig. 6e, Supplementary Figure 16d). In parallel, T cells exhibited enhanced gene expression associated with T cell receptor signaling and immunoregulatory interactions between a lymphoid and a non-lymphoid cell (Fig. 6f, Supplementary Fig. 16e). Cells of the mononuclear phagocyte system (MPS) showed enriched gene expression related to antigen processing and cross-presentation as well as cytokine signaling (Supplementary Fig. 16f, g). These studies established the antitumor activity of CBD-IL-12 in models of prostate cancer through reprogramming of the tumor microenvironment and engagement of both the innate and adaptive immune systems.

## Enhanced tumor control with concomitant STEAP1-mBBζ CAR T cell and CBD-IL-12 therapy

We next hypothesized that broadening the antitumor immune response with CBD-IL-12 therapy could improve the therapeutic efficacy of STEAP1-mBBζ CAR T cell therapy in prostate cancer by combating tumor antigen heterogeneity and rescue of antigen processing and presentation. We therefore established syngeneic, non-clonal RM9-STEAP1-fLuc metastases in male heterozygous hSTEAP1-KI mice and randomized to treatment with either $5 \times 10^6$ untransduced mouse T cells or mouse STEAP1-mBBζ CAR T cells by tail vein injection with or without weekly CBD-IL-12 therapy by retroorbital sinus injection. Groups that received CBD-IL-12 therapy alone or in combination with STEAP1-mBBζ CAR T therapy did not receive lymphodepleting cyclophosphamide (Fig. 7a). Serial BLI revealed rapid disease progression in mice treated with untransduced T cells and CBD-IL-12 while those receiving STEAP1-BBζ CAR T cells in combination with CBD-IL-12 therapy demonstrated a significant delay in tumor progression (Fig. 7b, c). Importantly, combined treatment with mouse STEAP1-mBBζ CAR T cells and weekly CBD-IL-12 was associated with a statistically significant extension in overall survival when compared to all other treatment groups (Fig. 7d). Plasma cytokine analysis on retro-orbital bleeds collected at day 0 and day 8 of treatment showed a significant increase in proinflammatory cytokines IFN-γ, TNF-α, IL-6, and IL-4 levels (Fig. 7e, Supplementary Fig. 17) in mice treated with STEAP1-mBBζ CAR T cells and CBD-IL-12.

Residual tumors were collected at necropsy and STEAP1 IHC showed antigen loss in tumors from mice treated with both STEAP1-mBBζ CAR T cells alone and in combination with CBD-IL-12 (Fig. 8a). In addition, we observed an increase in tumor B2m expression (Fig. 8a) associated with CBD-IL-12 therapy. CD3$^+$ T cells were also increased (Fig. 8b) in tumor conditions subjected to IL-12 therapy but, while a trend was appreciated, we did not find a statistically significant increase in intratumoral T cells between the STEAP1-mBBζ CAR T cell and combined STEAP1-mBBζ CAR T cell and CBD-IL-12 treatment groups.

Residual tumors including those from mice treated with combined STEAP1-mBBζ CAR T cell and CBD-IL-12 collected at maximal treatment response (nadir) at day 10 and at compassionate endpoints for tumor progression (relapse) were dissociated to single cells and immune cell subsets were characterized by multiparametric flow cytometry (Supplementary Figure 18). In concordance with our scRNA-seq results, CBD-IL-12 treatment either alone or in combination with STEAP1-mBBζ CAR T cells led to a significant increase in CD11b$^+$Ly6C$^{-/+}$F4/80$^+$MHC-II$^+$ macrophages (Fig. 8c, d, Supplementary fig. 19a). CD11b$^+$Ly6C$^-$F4/80$^+$MHC-II$^+$ cells represent mature antigen-presenting macrophages[66] and this population was preferentially enriched and demonstrated increased induced nitric oxide synthase (iNOS) expression as a marker of proinflammatory M1 polarization with CBD-IL-12 therapy. We also observed an expansion of the cDC1 population and a reduction in the conventional type 2 dendritic cell (cDC2) population (Fig. 8e, Supplementary Fig. 19b). cDC1 have been

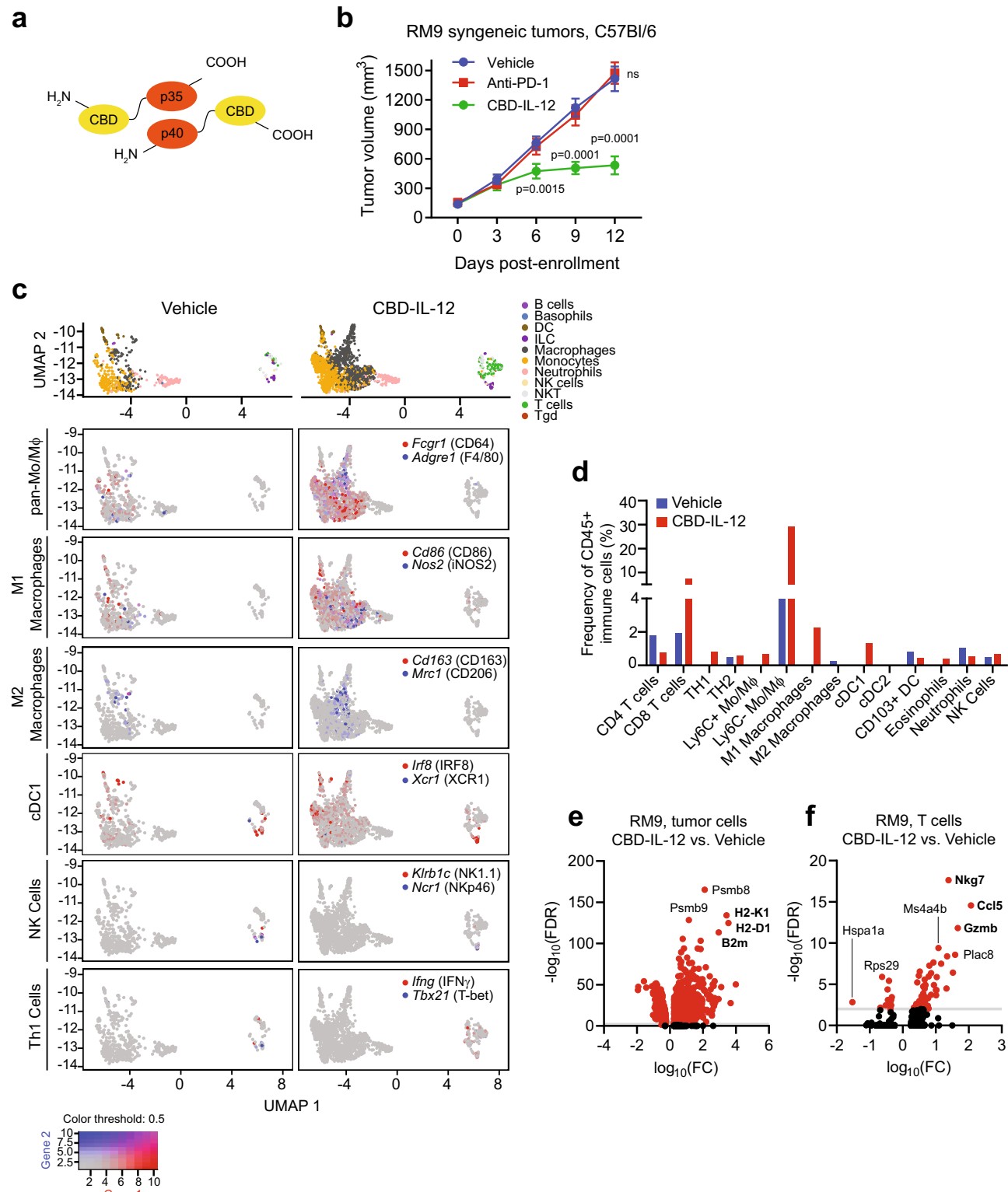

implicated in the activation of antitumor cytotoxic CD8+ T cells and T cell migration to tumor while cDC2 are important for the activation of CD4+ T cells including Tregs[67]. Consistent with the cDC2 findings, profiling of CD4+FOXP3+ Tregs revealed a decreased in Tregs associated with CBD-IL-12 therapy (Supplementary Figure 19c). Of interest, we observed that in relapsed tumors after combined STEAP1-mBBζ CAR T cell and CBD-IL-12 therapy there was a decrease in cDC1 and increase in Tregs, implicating reduced priming of cytotoxic CD8+ T cells and enrichment of immunosuppressive Treg signaling as

potential mechanisms of therapeutic resistance. We found no significant difference in the ratios of KLRG1+ and KLRG1- natural killer cells across treatment groups (Supplementary Fig. 19d). Frequencies of F40/80+SiglecF+ eosinophils were increased while Ly6G+ neutrophils were decreased with the addition of CBD-IL-12 to STEAP1-mBBζ CAR T cell therapy (Fig. 8f, g). Notably, the frequency of these tumor-associated neutrophils was heightened with treatment with STEAP1-mBBζ CAR T cells compared to untransduced T cells (54 vs. 34%), diminished in conditions with CBD-IL-12 therapy, but increased

**Fig. 6 | Systemic collagen-binding domain IL-12 (CBD-IL-12) cytokine fusion therapy inhibits prostate cancer tumor growth and reprograms the tumor immune microenvironment. a** Schematic of CBD-IL-12, composed of the p35 and p40 subunits fused to the CBD from von Willebrand factor domain A3. **b** Volumes of RM9 subcutaneous tumors in sygeneic C57Bl/6 mice over time with treatment with vehicle, anti-PD-1 (clone 29 F.1A12) 200 Mg by intraperitoneal injection every 5 days, or CBD-IL-12 25 Mg by intravenous injection every 5 days starting on day 0. $n = 7$ mice in vehicle and CBD-IL-12 treated groups and $n = 8$ mice in anti-PD1 treated group. $p < 0.0001$ at day 9 and 12. Bars represent mean with SEM. *P*-values are derived from two-way ANOVA with Dunnett's multiple comparisons test, ns not significant. **c** Uniform Manifold Approximation and Projection (UMAP) plots of different immune cell subsets (top) from single-cell RNA-seq (scRNA-seq) analysis of five RM9 tumors each aggregated from mice treated with vehicle or CBD-IL-12. UMAP plots colored with gene expression of immune cell subset-specific markers

for pan-monocytes/macrophages, M1 and M2 polarized marcophages, conventional type 1 dendritic cells (cDC1), natural killer (NK) cells, and T helper type 1 (Th1) cells. **d** Plots showing the frequency of specific immune cell populations (relative to CD45 + immune cells) identified by scRNA-seq analysis including CD4[+] and CD8[+] T cells, Th1 (Infg[+]Tbx21[+]) and Th2 (cMAF[+]Gata3[+]) cells, Ly6C[+/−] monocytes/macrophages (Ly6C[+/−]Adgre1[+]), M1 macrophages (CD80[+]CD86[+]INOS2[+]), M2 macrophages (CD163[+]Mrc1[+]cMAF[+]), cDC1 (XCR1[+]IRF8[+]), conventional type 2 dendritic cells (cDC2, CD1[+]IRF4[+]), migratory CD103[+] dendritic cells (Itgae[+]), eosinophils (SiglecF[+]), neutrophils (Ly6G[+]), and NK cells (Klrb1c[+]Ncr1[+]) in tumors treated with vehicle or CBD-IL-12. Volcano plots showing differential gene expression in (**e**) tumor cells and (**f**) T cells from RM9 tumors of mice treated CBD-IL-12 relative to those treated with vehicle. FC fold change, FDR false discovery rate. Source data are provided in the Source Data file.

upon tumor relapse after combined STEAP1-mBBζ CAR T cell and CBD-IL-12 therapy (19 to 42%). These findings suggest that the immunosuppressive properties[66] of tumor-associated neutrophils may play a prominent role in mediating treatment resistance and tumor progression. Overall, these analyses indicate that CBD-IL-12 treatment reverts the hostile immunosuppressive tumor milieu to a proinflammatory state and broadens antitumor activity in conjunction with adoptively transferred STEAP1-mBBζ CAR T cell therapy.

We further performed TCR repertoire analysis using multiplex PCR-based TCR beta chain sequencing[68] on lungs bearing tumors collected from each treatment group at necropsy. We observed a significant decrease in Simpson clonality in samples from mice treated with CBD-IL-12 alone and in combination with STEAP1-mBBζ CAR T cell therapy which was indicative of increased intratumoral T cell diversity (Fig. 8h). These findings establish that adding CBD-IL-12 as an adjunct to STEAP1 CAR T cell therapy may be beneficial through remodeling of the prostate cancer tumor microenvironment, enhancing antigen processing and presentation, and engaging host immunity to promote epitope spreading.

## Discussion

The effectiveness of CAR T cell therapy and other immune-based targeted therapeutics is highly dependent on consistent antigen expression on all or most cells comprising the tumor population within an individual patient. However, antigen heterogeneity is pronounced in solid tumors including prostate cancer, where progression to mCRPC and treatment resistance are associated with the emergence of divergent disease subtypes marked by distinct transcriptional programs[69–71] and cell surface antigen expression[17]. While PSMA is considered one of the foremost biomarkers in prostate cancer with significant overexpression found across the spectrum of disease progression, our work corroborates findings from a recent publication[14] indicating that PSMA expression is heterogeneous in lethal mCRPC. We show that STEAP1 is more broadly expressed than PSMA in this setting but is by no means expressed uniformly at high levels in all mCRPC tissues. No single antigen-targeted therapy including CAR T cell therapy may be able to overcome pre-existing tumor antigen heterogeneity in mCRPC. Thus, it is of critical importance to thoroughly credential additional therapeutic targets such as STEAP1 in mCRPC that may enable combinatorial therapies that exert insurmountable therapeutic pressure. These include dual antigen-targeted (e.g., PSMA and STEAP1) CAR T cell therapies or multimodal strategies combining CAR T cell therapies with ADCs, T-BsAbs, or other treatments that potently promote antigen-independent and -dependent tumor killing.

We engineered a STEAP1-targeted CAR T cell therapy that is highly antigen-specific and functionally localized the epitope recognized by the CAR to the second ECD of STEAP1. Our STEAP1-BBζ CAR T cells demonstrate substantial antitumor activity against multiple disseminated prostate cancer models both in human-in-mouse and mouse-in-mouse studies. Importantly, our STEAP1-BBζ CAR is capable

of inducing T cell activation and target cell cytolysis even in low antigen density conditions, as evidenced by reactivity against the PC3 prostate cancer model. However, this sensitivity of STEAP1-BBζ CAR T cells to low levels of STEAP1 expression may be advantageous from the perspective of enhancing antitumor efficacy but could also accentuate liabilities from on-target off-tumor toxicity. Systemic expression of STEAP1 has previously been reported as virtually absent in normal human tissues[18,72] except the prostate gland where membranous expression in prostate epithelial cells has been described[27]. To delve into the safety of STEAP1-BBζ CAR T cell therapy in the preclinical setting, we generated a humanized STEAP1 mouse model. The hSTEAP1-KI mouse model recapitulated human STEAP1 expression in the prostate gland and showed expression in the adrenal cortex. Reassuringly, STEAP1-BBζ CAR T cell therapy at a dose sufficient to induce antitumor activity did not lead to evident systemic toxicities in hSTEAP1-KI mice including on-target off-tumor toxicities at sites of human STEAP1 expression.

A recurring mechanism of prostate cancer relapse and progression after STEAP1-BBζ CAR T cell therapy in our studies was tumor antigen escape. On one hand, this finding underscores the overall potency of STEAP1-BBζ CAR T cell therapy. However, it is unclear whether the loss of tumor STEAP1 expression is solely due to inherent tumor antigen heterogeneity or whether there is also adaptive downregulation of STEAP1 expression. A recent publication showed that promoter methylation of *STEAP1* modulates STEAP1 expression and epigenetic deregulation by DNA methyltransferase and histone deacetylase inhibition was sufficient to significantly upregulate STEAP1 expression[73]. Treatment with epigenetic inhibitors in combination with STEAP1 CAR T cell therapy could simultaneously enhance tumor STEAP1 expression and reprogram CAR T cells to favorable exhaustion-resistant differentiation states[74,75], thereby mitigating tumor antigen loss and enhancing antitumor efficacy in prostate cancer. Our study also implicates STEAP1 as playing a functional role in regulating cell cycle progression and cellular metabolism in prostate cancer. STEAP1 is unique from other STEAP family members (STEAP2, 3, and 4) in that it lacks an intracellular oxidoreductase domain[22] which is necessary for metalloreductase activity. As a result, STEAP1 homotrimers, but not heterotrimers with other STEAP proteins, lack enzymatic function to reduce $Fe^{3+}$ to $Fe^{2+}$ and $Cu^{2+}$ to $Cu^{1+}$. Whether and how the involvement of STEAP1 in metal ions and cellular metabolism promotes cancer progression has yet to be determined and is worthy of further investigation.

The immunologically 'cold' tumor microenvironment of prostate cancer is a major barrier to the efficacy of cancer immunotherapies. For example, exploratory studies associated with a phase I clinical trial of PSMA CAR T cell therapy armored to express dominant-negative transforming growth factor-β receptor (TGFβR-DN) in mCRPC showed that the expression of immunosuppressive signaling molecules in the tumor microenvironment increases after CAR T infusion[13]. In our studies, a critical finding was the association between the loss of STEAP1

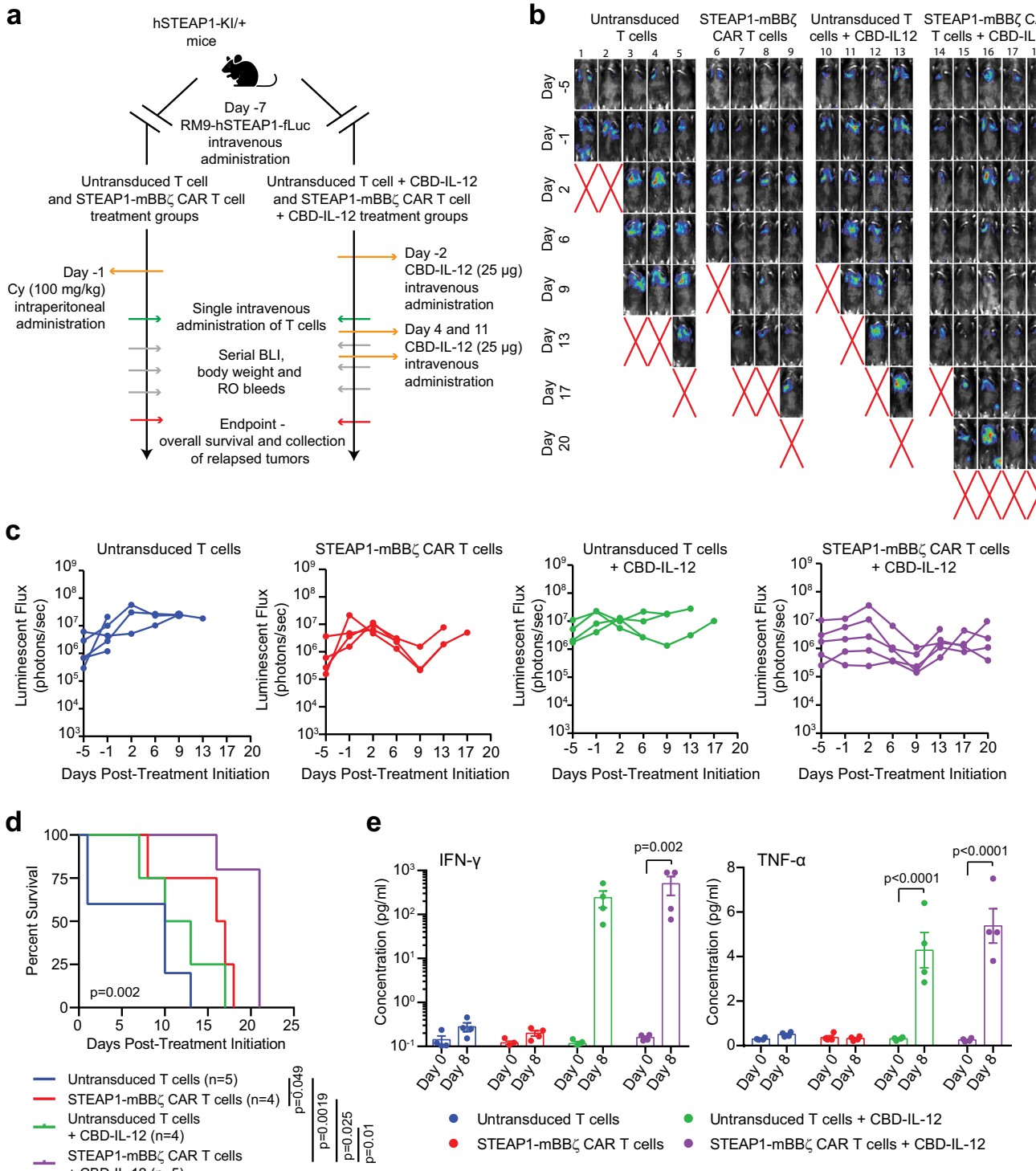

**Fig. 7 | Combining CBD-IL-12 with STEAP1-mBBζ CAR T cell therapy enhances overall survival and inflammatory cytokine levels. a** Schematic of the tumor challenge experiment for the RM9-hSTEAP1 disseminated model in hSTEAP1-KI/+ mice investigating the combination of CBD-IL-12 with STEAP1-mBBζ CAR T cell therapy. Cy cyclophosphamide (for preconditioning). Created with BioRender.com. **b** Serial live BLI of hSTEAP1-KI/+ mice engrafted with RM9-hSTEAP1-fLuc metastases and treated with a single intravenous injection of $5 \times 10^6$ mouse untransduced T cells or STEAP1-mBBζ CAR T cells on day 0 with or without CBD-IL-12 treatment weekly. Red X denotes deceased mice. The radiance scale is shown. (**c**) Plot showing the quantification of total flux over time from live BLI of each mouse in

(**b**). **d** Kaplan−Meier survival curves of mice in (**b**) with statistical significance determined by log-rank (Mantel-Cox) test ($p = 0.002$). **e** Plots showing serum cytokine levels of IFN-γ (left, $p = 0.002$) and TNF-α (right, $p < 0.0001$) based on ProcartaPlex immunoassays from retroorbital bleeds of hSTEAP1-KI/+ mice ($n = 4$ mice per group) bearing RM9-hSTEAP1-fLuc metastases prior to (day 0) and after treatment (day 8) with untransduced mouse T cells or mouse STEAP1-mBBζ CAR T cells with or without CBD-IL-12 therapy. Error bar represents mean with SEM. For panel (**e**), $p$-values were derived from two-way ANOVA with Sidak's multiple comparisons test. Source data are provided in the Source Data file.

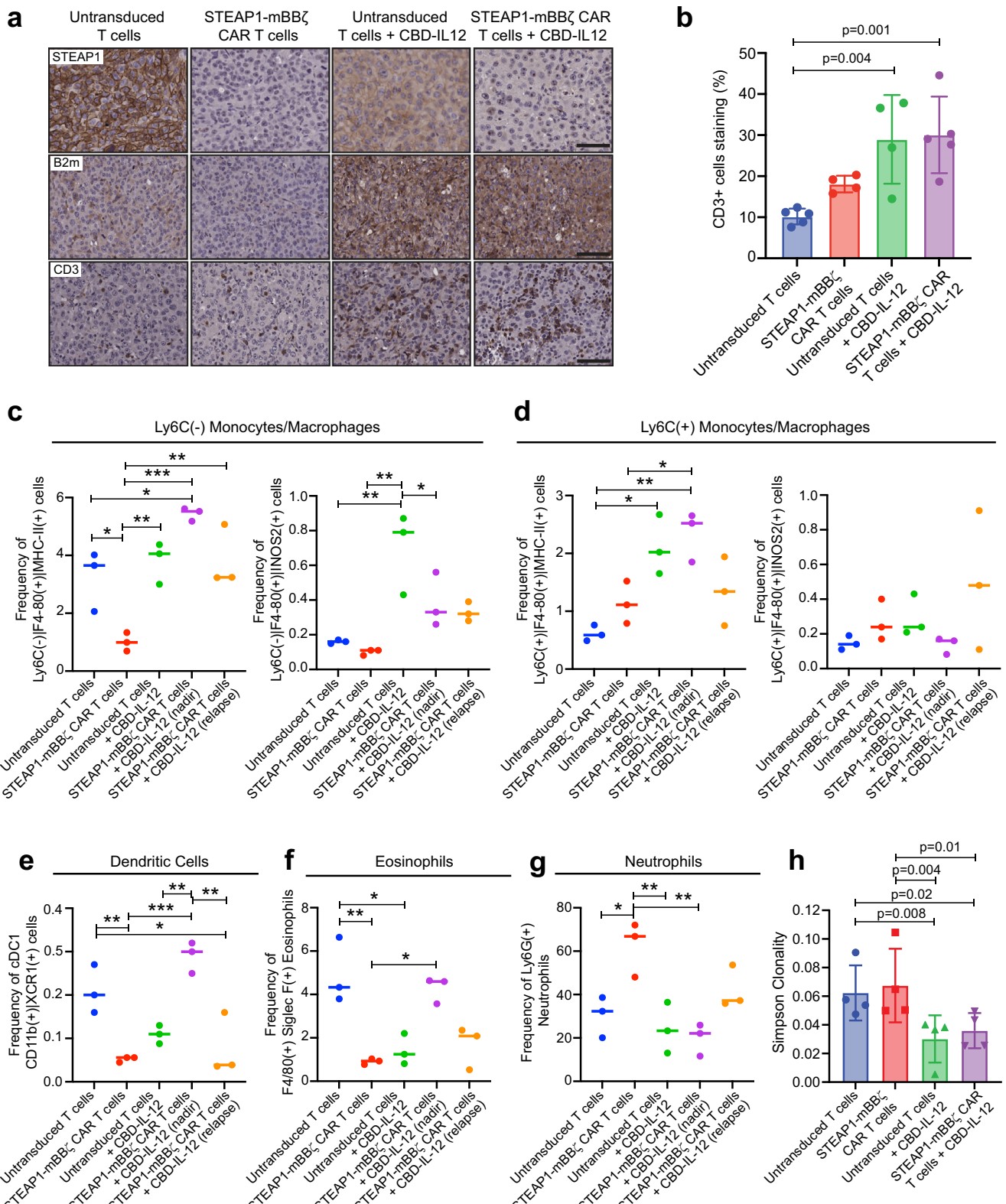

expression and downregulation of antigen processing and presentation, both in STEAP1 knockout cells and STEAP1 antigen loss tumors after STEAP1-BBζ CAR T cell therapy. Thus, STEAP1 antigen loss in prostate cancer promotes not only direct resistance to STEAP1-BBζ CAR T cell therapy but may also limit host adaptive antitumor immunity. Loss of tumor target antigen expression and conversion to a more immunosuppressive state with increased infiltration of Tregs and

higher expression of immunosuppressive molecules has also been observed in a phase I study of EGFR-vIII CAR T cells in participants with recurrent glioblastoma[76]. Additional work will be necessary to understand the functional mechanisms underlying STEAP1 antigen loss and more generally how dynamic effects of adoptive CAR T cell therapy may contribute to antigen loss and immunoediting by modulating tumor-immune-stromal interactions in solid tumors.

**Fig. 8 | Combining CBD-IL-12 with STEAP1-mBBζ CAR T cell therapy reprograms the tumor immune microenvironment and promotes antigen presentation and epitope spreading. a** Photomicrographs of STEAP1 (top), B2m (middle), and CD3 (bottom) IHC staining of RM9-hSTEAP1 lung tumors after treatment with mouse untransduced T cells or STEAP1-mBBζ CAR T cells with or without CBD-IL-12 treatment. Scale bars = 50 μm. **b** Bar plot showing IHC quantification of CD3 positive cells infiltrating the metastatic lung tumors (n = 4 tumors per group). One-way ANOVA p = 0.0021. Error bar represents mean with SD. Plots showing the frequencies of (**c**) Ly6C⁻F4/80⁺MHC-II⁺ (left, one-way ANOVA p = 0.0005) and Ly6C⁻F4/80⁺iNOS2⁺ (right, one-way ANOVA p = 0.0017) macrophages, (**d**) Ly6C⁺F4/80⁺MHC-II⁺ (left, one-way ANOVA p = 0.0038) and Ly6C⁺F4/80⁺iNOS2⁺ (right, one-way ANOVA p = ns) macrophages, (**e**) CD11b⁺XCR1⁺ cDC1 (one-way ANOVA p = 0.0003), (**f**) F4/80⁺SiglecF⁺ eosinophils (one-way ANOVA p = 0.0008), and (**g**) Ly6G⁺ neutrophils (one-way ANOVA p = 0.0035) normalized to total CD45⁺ cells as determined by multiparametric flow cytometry after treatment with untransduced T cells, STEAP1-mBBζ CAR T cells, untransduced T cells and CBD-IL-12, and STEAP1-mBBζ CAR T cells and CBD-IL-12 at maximal treatment response (nadir) and tumor relapse (relapse). **h** Bar plots representing Simpson clonality as a measure of 'evenness' of the TCR repertoire analyzed by TCRB sequencing on tumor infiltrating cells collected from mice in (**a**). n = 4 tumors per group. P-values for untransduced T cels + CBD-IL-12 compared to untransduced T cells and STEAP1-mBBζ CAR T cells are 0.008 and 0.02, respectively; and STEAP1-mBBζ CAR T cells + CBD-IL-12 compared to untransduced and STEAP1-mBBζ CAR T cells are 0.004 and 0.01, respectively. Error bar represents mean with SD. For panels (**c**–**g**) n = 3 tumors per group, *p < 0.05; **p < 0.01, ***p < 0.001, p-values in panels (**b**–**h**) are from one-way ANOVA with Dunn's multiple comparisons test. Source data are provided in the Source Data file.

To broaden the antitumor response, we investigated the systemic administration of CBD-IL-12 in combination with STEAP1-BBζ CAR T cell therapy in a disseminated, syngeneic RM9-hSTEAP1 tumor model in hSTEAP1-KI mice to approximate the immunosuppressive nature of mCRPC based on prior characterization of RM9 as a poorly immunogenic model[57,77]. Co-treatment with CBD-IL12 led to improvements in overall survival, cytokine production, tumor antigen presentation, and intratumoral T cell diversity consistent with epitope spreading. Close investigation of the tumor immune microenvironment revealed that adding CBD-IL-12 to STEAP1-BBζ CAR T cell therapy induces tumor stromal reversion and enhances activated macrophages and cDC1 while reducing tumor-associated neutrophils. However, cure was not achieved in our studies, and we highlight potential compensatory, adaptive mechanisms of resistance and tumor progression including increased frequencies of immunosuppressive neutrophils and cDC2 which may promote Treg induction. It is important to note that further optimization of the dose and administration schedule of CBD-IL-12 may be needed to maximize antitumor responses. However, these results support the investigation of combinatorial immunotherapeutic approaches such as armoring CAR T cells to express recombinant cytokines[78] (e.g., IL-2, IL-12, IL-15, or IL-18), concurrent treatment with immunomodulators (e.g., anti-PD-1/PD-L1 or anti-CTLA4), or tumor-directed radiotherapy to remodel the prostate cancer tumor microenvironment and enhance CAR T cell effector function.

While this manuscript was in preparation, a study from a group in Norway reported the preclinical development of a STEAP1 CAR T cell therapy with antitumor activity in a subcutaneous 22Rv1 model in immunocompromised mice[79]. The reported STEAP1 CAR differs from STEAP1-BBζ CAR as it incorporates a synthetic scFv called Oslo1 and a CD8α hinge and transmembrane domain. Another differentiating feature is that STEAP1-BBζ CAR T cells are prepared as a defined product with a normal ratio of CD4/CD8 T cells while the Oslo1 STEAP1 CAR T cells are not. Mechanisms of resistance to and safety studies related to the Oslo1 STEAP1 CAR T cell therapy were not reported. However, it will be of interest to see how these differences in CAR engineering and cell product composition may impact antitumor efficacy, persistence, and safety as both the Oslo1 STEAP1 CAR T cell therapy and our STEAP1-BBζ CAR T cell therapy programs are translated to the clinic.

The findings of our studies have led to a partnership with the National Cancer Institute (NCI) Experimental Therapeutics (NExT) Program with the goal of translating STEAP1-BBζ CAR T cell therapy to a first-in-human trial for men with mCRPC. Safety and efficacy signals from this early phase clinical trial will help determine whether there may also be value in investigating this therapeutic approach for other cancer types that highly express STEAP1.

## Methods

All studies were conducted in accordance with the ethical guidelines expressed in the World Medical Association Declaration of Helsinki.

### Cell lines

22Rv1 (CRL-2505), LNCaP (CRL-1740), PC3 (CRL-1435), DU145 (HTB-81), NCI-H660 (CRL-5813), C4-2B (CRL-3315), RM9 (RL-3312), HEK293T (CRL-3216), and Myc-CaP (CRL-3255) were obtained from the American Type Culture Collection. LNCaP95 cells were a gift from Stephen R. Plymate (University of Washington, Seattle). MSKCC EF1 were derived from the MSKCC PCa4 organoid line provided by Yu Chen (Memorial Sloan Kettering Cancer Center), as previously described[17]. Cell lines were maintained in RPMI 1640 medium supplemented with 10% FBS, 100 U/mL penicillin and 100 μg/mL streptomycin, and 4 mmol/L GlutaMAX (Thermo Fisher). PLAT-E (RV-101) Retroviral packaging cell line was obtained from Cell Biolabs, Inc. and were cultured in DMEM supplemented with 10% FBS media.

22Rv1 STEAP1 ko and PC3 STEAP1 ko cells were generated by transient transfection of 22Rv1 cells with a pool of PX458 (Addgene, #48138) plasmids each expressing one of four different sgRNA targeting sequences predicted from the Broad Institute Genetic Perturbation Platform sgRNA Designer[80]: (1) 5′-ATAGTCTGTCTTACCCAATG-3′; (2) 5′-CCTTTGTAGCATAAGGACAC-3′; (3) 5′-ATCCACTTATCCAACCAATG-3′; and (4) 5′-CATCAACAAAGTCTTGCCAA-3′. 48–72 h after transfection, GFP-positive cells were singly sorted on a Sony SH800 Cell Sorter into a 96-well plate and clonally expanded.

gBlocks encoding firefly luciferase (fLuc), hSTEAP1, mSTEAP1, mSTEAP1 hECD1, mSTEAP1 hECD2, mSTEAP1 hECD3, hSTEAP1B isoform 1, hSTEAP1B isoform 2, and hSTEAP1B isoform 3 were cloned into the EcoRI site of FU-CGW[81] by HiFi DNA Assembly. The FU-hSTEAP1-CGW plasmid was modified to excise the GFP cassette by digestion with AgeI and BsrGI. A PCR product encoding firefly (fLuc) was then inserted by HiFi DNA assembly to generate the FU-hSTEAP1-C-fLuc-GW plasmid. These lentiviruses were produced and titered as previously described[82]. Lentiviruses were used to introduce stable transgene expression in the cell lines noted above.

### Animal studies

All mouse studies were performed in accordance with protocols approved by the Fred Hutchinson Cancer Center Institutional Animal Care and Use Committee and regulations of Comparative Medicine.

For studies using immunocompromised mice, six- to eight-week-old male NSG (NOD-SCID-IL2Rγ-null) mice were obtained from The Jackson Laboratory. For the 22Rv1 subcutaneous tumor and intratumoral T cell administration model, 2 × 10⁶ 22Rv1 cells were suspended in 100 μl ice-cold Matrigel matrix (Corning) and injected subcutaneously into the flanks of NSG mice. Tumors were measured twice weekly using electronic calipers and tumor volume (TV) calculated based on the equation TV = ½ (L * W²). When TV was ~75–100 mm³, 5 × 10⁶ untransduced or STEAP1 CAR T cells at a defined CD4/CD8 composition of 1:1 and suspended in 100 μl of PBS was injected intratumorally. Mice were sacrificed 25 days after intratumoral T cell therapy. For disseminated human prostate cancer and intravenous T cell administration models, between 5 × 10⁵ and 10⁶ prostate cancer cells

were suspended in 100 µl of PBS and injected into the tail veins of NSG mice. Tumor burden was monitored by live bioluminescence imaging on an IVIS Spectrum (PerkinElmer) after intraperitoneal injection of XenoLight D-luciferin (PerkinElmer). When metastatic colonization was confirmed by imaging, $5 \times 10^6$ human untransduced or STEAP1-BBζ CAR T cells at a defined CD4/CD8 composition of 1:1 and suspended in 100 µl of PBS was injected by tail vein. Metastatic tumors and spleen were harvested when mice were euthanized at compassionate endpoints. In the disseminated C4-2B model, lungs and livers were collected from a subset of mice and placed individually in six-well plates in DMEM media supplemented with 10% FBS and GlutaMAX. Ex vivo bioluminescence imaging was performed by introducing XenoLight D-luciferin into the media and quantifying signal on an IVIS Spectrum. Metastatic tumors were formalin-fixed and paraffin-embedded. Splenocytes were harvested from spleens to perform flow cytometry to evaluate the peripheral persistence of CAR T cells.

For animal studies using hSTEAP1-KI mice, heterozygous hSTEAP1-KI mice were generated by crossing homozygous hSTEAP1-KI mice with wildtype C57Bl/6 mice. Genotyping of all hSTEAP1-KI mice was performed by PCR of 10 ng of tail DNA using the Taq 2X Master Mix (New England Biolabs) and visualization of PCR products by gel electrophoresis on a 2% agarose gel. Primers used for genotyping PCR reactions are (1) wildtype: forward 5′-CTAGGTGGCTGAAGCCGTA-3′ and reverse 5′-GCGATGACCAAAAGTGACTTC-3′, (2) hSTEAP1-KI: forward 5′-CAGATGAGGTAGGATGGGATAAAC-3′ and reverse 5′-CCTCAAGCATGGCAGGAATAG-3′. Thermocycler conditions for genotyping PCR are 95 °C × 30 sec; (95 °C × 30 sec, 58 °C × 30 sec, 68 °C × 70 sec) × 35 cycles; 68 °C × 5 min; and 12 °C hold.

For disseminated RM9-hSTEAP1-fLuc mouse prostate cancer and intravenous T cell administration models, $5 \times 10^5$ RM9-hSTEAP1-fLuc cells were suspended in 100 µl of PBS and injected into the tail veins of either NSG or heterozygous hSTEAP1-KI mice. Tumor burden was monitored by live bioluminescence imaging on an IVIS Spectrum (PerkinElmer) after intraperitoneal injection of XenoLight D-luciferin (PerkinElmer). When metastatic colonization was confirmed by imaging, $5 \times 10^6$ mouse untransduced or STEAP1-mBBζ CAR T cells suspended in 100 µl of PBS was injected by tail vein. Retroorbital blood was collected using heparinized capillary tubes into polystyrene tubes containing an EDTA/PBS solution. After collection, retroorbital blood samples were incubated at room temperature for 15–20 min and centrifuged in a tabletop centrifuge at $2000 \times g$ for 10 min. Plasma was collected and stored at −80 °C. Lungs, spleen, prostate, and adrenal glands were harvested when mice were euthanized at compassionate endpoints. Lungs, prostate, and adrenal glands were formalin-fixed and paraffin-embedded. Splenocytes were harvested from spleens to perform flow cytometry to evaluate the peripheral persistence of CAR T cells.

For animal studies to assess the antitumor efficacy of systemic CBD-IL-12 treatment in prostate cancer, $10^6$ syngeneic RM9 and Myc-CaP murine prostate cancer cells were subcutaneously injected into the flanks of C57Bl/6 and FVB mice, respectively. Tumors were measured using electronic calipers twice weekly. Once TV was ~100 mm³, mice were randomized to treatment with (a) PBS control, (b) 200 Mg anti-PD-1 (BioXCell, clone 29 F.1A12) by intraperitoneal injection every 5 days, and (c) 25 µg CBD-1L-12 by retroorbital sinus injection every 5 days. Tumors were harvested at the end of the experiment, dissociated viably to single cells, and subjected to single cell RNA-sequencing. Tumors were also collected and processed for paraffin embedding for IHC analysis.

For animal studies testing combined CBD-IL-12 and STEAP1-mBBζ CAR T cell therapy, heterozygous hSTEAP1-KI mice were inoculated with RM9-hSTEAP1-fLuc cells by tail vein injection as described above. When metastatic colonization was confirmed, mice were randomized into four groups: (a) untransduced T cells, (b) STEAP1-mBBζ CAR T cells, (c) untransduced T cells + CBD-IL-12, and (d) STEAP1-mBBζ CAR T cells + CBD-IL-12. 48 h prior to T cell infusion, groups c and d were pre-treated with 25 µg CBD-IL-12 by retroorbital sinus injection and then weekly thereafter. 24 h prior to T cell infusion, groups a and b received preconditioning cyclophosphamide 100 mg/kg. $5 \times 10^6$ mouse untransduced or STEAP1-mBBζ CAR T cells suspended in 100 µl of PBS was injected by tail vein. Tumor burden monitoring and collection of retroorbital blood and tissue samples were performed as described above.

For all mice experiments, animals were routinely monitored for disease progression and overt toxicity such as >10% decline in body weights, lethargy, immobility, fur ruffling, and fur loss. For the subcutaneous mouse tumor model, mice were euthanized when the tumor size approached 2.0 cm in diameter which is the maximum tumor size allowed for a mouse bearing a single tumor. For metastatic mouse models, mice were sacrificed when tumor burden based on bioluminescence imaging reached $>1 \times 10^{12}$ photons/sec/cm²/steradian or institutional compassionate euthanasia criteria were met. Maximal tumor size/burden was not exceeded.

## Immunohistochemical studies

Tissue sections were deparaffinized in xylene and rehydrated in 100, 95, 75% ethanol, and finally TBS with 0.1% Tween 20 (TBST). Antigen retrieval was conducted in Citrate-Based Antigen Unmasking Solution (Vector Labs) using a pressure cooker at 95 °C for 30 min. Tissue sections were blocked with Dual Endogenous Enzyme-Blocking Reagent (Agilent Technologies) and incubated for 10 min followed by three washes with TBST. Slides were incubated with primary antibody in a humidified chamber at 37 °C for one hour. Primary antibodies and dilutions used for this study are rabbit anti-STEAP1 polyclonal antibody (LS Bio, LS-C291740, 1:500), rabbit anti-CD3 antibody (Thermo Fisher, MA5-14524, 1:100), mouse anti-PSMA antibody (Dako, Clone 3E6, M3620, 1:50), rabbit anti-Androgen Receptor antibody (Abcam, Clone EPR1535(2), ab133273), mouse anti-SYP/Synaptophysin Antibody (Santa Cruz, sc-17750 (D4), 1:50), rabbit anti-beta-2-microglobulin antibody (Clone EPR21752-214, Abcam, ab218230, 1:1000) and mouse anti-HLA Class 1 ABC antibody (Clone EMR8-5, Abcam, ab70328, 1:3000). Slides were washed with TBST and incubated with Power-Vision Poly-HRP anti-rabbit IgG or anti-mouse IgG (Leica Biosystems) in a humidified chamber at 37 °C for 30 min. Slides were washed with TBST and incubated with 3,3′-Diaminobenzidine (DAB) (Sigma Aldrich) at room temperature for 10 min. DAB was quenched in deionized water. Slides were stained in Dako hematoxylin (Agilent Technologies) at room temperature for one minute and washed with deionized water for five minutes. Slides were dehydrated in 75, 95, 100% ethanol, and finally xylene. Slides were mounted using Permount mounting medium (Fisher Chemical), and cover slipped. Stained slides were scanned using Ventana DP200 (Roche). QuPath 0.2.3 was used to analyze the IHC images.

## Prostate cancer tissue microarray analysis

University of Washington mCRPC Tissue Acquisition Necropsy (TAN) tissue microarrays (Prostate Cancer Biorepository Network) were used for immunohistochemical studies. Human tissues studies were approved by the Institutional Review Board of the University of Washington (Protocol #2341). All rapid autopsy tissues represented on the tissue microarray were collected from patients who provided informed written consent under the Prostate Cancer Donor Program at the University of Washington. Each core stained with either STEAP1 or PSMA was scored by an experienced pathologist (M.P.R) and assigned an intensity score of 0, 1, 2, or 3 and frequency of positive cell staining ranging from 0 to 100%. H-scores were generated for each core by multiplying the intensity score by the frequency of positive cell staining resulting in a minimum of 0 and maximum of 300. The average H-score of replicate cores represented in the tissue microarray was determined for each mCRPC tissue.

## Immunoblotting

Protein extracts were collected in a 9M urea lysis buffer and quantified using the Pierce Rapid Gold BCA Protein Assay Kit (Thermo Fisher). Protein samples were fractionated with SDS-PAGE using Bolt 4–12% Bis-Tris Plus Gels (Thermo Fisher) and transferred to nitrocellulose membranes using the Invitrogen Mini Blot module (Thermo Fisher). Membranes were blocked with 5% non-fat milk in PBS + 0.5% Tween 20 (PBST) on a shaker at room temperature for 30 min. Membranes were then incubated with primary antibody on a shaker at 4 °C overnight. Primary antibodies used for this study are mouse anti-STEAP (Clone B-4, Santa Cruz, sc-271872, 1:1000) for human STEAP1, Polyclonal Rabbit anti-STEAP1 antibody (BioRad, AHP1438) for murine STEAP1, and GAPDH (Clone GT239, GeneTex, GX627408, 1:5000). Membranes were washed with PBST and incubated with goat anti-mouse IgG (H + L) secondary antibody conjugated with horseradish peroxidase (Thermo Fisher, 31430, 1:10,000) on a shaker at room temperature for 1 h. Membranes were washed with PBST, incubated with Immobilon Western Chemiluminescent HRP Substrate (EMD Millipore) at room temperature for three minutes, and visualized on a ChemiDoc MP Imaging System (Bio-Rad Laboratories).

## Absolute STEAP1 quantification

Flow cytometric quantification of STEAP1 antigen density across human prostate cancer cell lines was performed using Quantum Simply Cellular microspheres (Bangs Laboratories) per the manufacturer's protocol. Vandortuzumab (Creative Biolabs) was used as the primary antibody and rat anti-human IgG Fc antibody conjugated to APC (BioLegend) was used as the secondary antibody. Stained cells and beads were analyzed on a BD FACSCanto II (BD Biosciences). The Geo Mean or Median channel values for each population were recorded into the provided QuickCal spreadsheet yielding a regression coefficient of 0.998.

## Chimeric antigen receptors (CAR) expression plasmids

gBlocks (Integrated DNA Technologies) encoding the GM-CSF leader, DSTP3086S scFv (VL-[G4S]3-VH), IgG4 spacers, CD28 transmembrane domain, 4-1BB costimulatory domain, CD3ζ chain, EGFRt or mCD19t, and WPRE were cloned into the EcoRI site of pCCL-c-MNDU3-X (gift from Donald Kohn, Addgene plasmid #81071) or pMYs (Cell Biolabs) by HiFi DNA Assembly (New England Biolabs). Sequences involving cloning junctions and open reading frames were validated by Sanger sequencing at the Fred Hutch Genomics & Bioinformatics Shared Resource.

## CAR lentivirus and retrovirus production

For STEAP1-BBζ CAR lentivirus production, HEK 293 T cells (ATCC) were thawed, cultured, and expanded in DMEM media supplemented with 10% FBS and GlutaMAX. HEK 293 T cells were seeded on plates coated with Cultrex Poly-L-Lysine (R&D Systems) prior to transfection with the pCCl-c-MNDU3 STEAP1-BBζ CAR lentiviral plasmid and the packaging plasmids pMDL, pVSVg, and pREV using the TransIT-293 transfection reagent (Mirus Bio). About 18 h after transfection, sodium butyrate and HEPES were added to each plate to a final concentration of 20 mM each. Eight hours later, media was aspirated from the plates and each plate was washed with PBS. DMEM media supplemented with 10% FBS, GlutaMAX, and 20 mM HEPES was added to each plate. Lentiviral supernatant was collected at 48 h after transfection, vacuum filtered through a 0.22 μm filter, and concentrated by ultracentrifugation in polypropylene Konical tubes (Beckman Coulter) at 85,929 g at 4 °C for two hours in an Optima XE 90 (Beckman Coulter). Lentiviral pellets were resuspended in the minimal residual media present after aspirating off supernatant, aliquoted in cryovials and stored at −80 °C.

For STEAP1-mBBζ CAR retrovirus production, PLAT-E cells (Cell Biolabs) were thawed and cultured in DMEM media supplemented with 10% FBS, GlutaMAX, 1 μg/ml puromycin, and 10 μg/ml blasticidin. One day prior to seeding cells for transfection, PLAT-E cells were washed and seeded in antibiotic-free DMEM media supplemented with 10% FBS and GlutaMAX. PLAT-E cells were transfected with the pMYs STEAP1-mBBζ CAR retroviral construct using the FuGENE HD transfection reagent (Promega). 48 and 72 h after transfection, supernatants containing retrovirus were passed through a 0.22 μm syringe filter prior to use in transduction.

## Human CAR T cell manufacturing

Peripheral blood mononuclear cells (PBMCs) from three de-identified healthy donor used in this study was reviewed by the Institutional Review Board of the Fred Hutchinson Cancer Center and deemed Not Human Subjects Research. PBMCs were collected from participants who provided informed written consent under the Co-Operative Center for Excellence in Hematology at Fred Hutchinson Cancer Center. PBMCs were thawed and washed with pre-warmed TCM base media consisting of AIM-V media (Gibco) supplemented with 55 mM beta-mercaptoethanol, human male AB plasma (Sigma), and GlutaMAX. PBMCs were centrifuged in a tabletop centrifuge at 600 g for 5 min. Cells were resuspended in TCM base media and counted on a hemacytometer. Dynabeads CD8 and CD4 Positive Isolation Kits were used per manufacturer's protocol to separate CD8 and CD4 T cells. After bead detachment, CD8 T cells were seeded in CD8 media (TCM base media supplemented with 50 U/ml human IL-2 and 0.5 ng/ml human IL-15) and CD4 T cells were seeded in CD4 media (TCM base media supplemented with 0.5 ng/ml human IL-15 and 5 ng/ml human IL-7). CD8 and CD4 T cells were activated and expanded with Dynabeads Human T-Activator CD3/CD28 (Thermo Fisher) per manufacturer's protocol. After two to four days, CD8 and CD4 T cells were counted and transduced with STEAP1-BBζ CAR lentivirus at a relative multiplicity-of-infection of 10 based on the infectious titer on HEK 293 T cells. Lentiviral transduction was performed in the presence of 10 μg/ml protamine sulfate. 48 h after transduction, T cells were collected, activation beads removed, and transduction efficiency of the T cells evaluated by flow cytometry. CAR-modified CD4 and CD8 T cells were counted every two days and maintained at a density of 10^6 cells/ml in their respective CD4 and CD8 media.

## Mouse CAR T cell manufacturing

Splenocytes were harvested from the manual dissociation of spleens obtained from heterozygous hSTEAP1-KI mice. Splenocytes were passed through a 70 μm strainer and pelleted by centrifugation at 600 g for six minutes. Cells were resuspended in RBC lysis buffer (BioLegend) and incubated on ice for five minutes. Cells were washed with PBS and pelleted by centrifugation. Murine CD3 T cells were isolated using Mouse CD3 + T Cell Enrichment Columns (R&D Systems) per the manufacturer's protocol. T cells were cultured in RPMI 1640 media containing 10% FBS, 50 U/ml human IL-2, 10 ng/ml murine IL-7, and 50 μM beta-mercaptoethanol. T cells were activated and expanded with Dynabeads Mouse T-Activator CD3/CD28 (Thermo Fisher) per manufacturer's protocol. 48 and 72 h later, mouse T cells were transduced with filtered pMYs STEAP1-mBBζ CAR retroviral supernatants via spinoculation on a tabletop centrifuge at 2000 × g at 30 °C for two hours. On day six of culture, beads were magnetically removed, and T cell transduction efficiency was determined by flow cytometry prior to use in functional assays.

## Immunophenotyping CAR T cell products and assessment of peripheral persistence

$1.5 \times 10^5$ human untransduced or STEAP1-BBζ CAR T cells, mouse untransduced or STEAP1-BBζ CAR T cells, or splenocytes were incubated with fluorophore conjugated antibodies on ice for 20 min. Cells were washed with PBS after antibody staining and acquired on a BD

FACSCanto II or BD Symphony 4. Data were analyzed on FlowJo v.10.8 (Treestar).

Antibodies used for immunophenotyping transduced human T cells were mouse anti-human CD3 conjugated to APC (1:100, clone SK7, Thermo Fisher, 47-0036-42), mouse anti-human CD8 conjugated to FITC (1:100, clone RPA-T8, BD Biosciences, 555366), rabbit anti-human EGFR (1:100, clone C225, cetuximab) conjugated to PE (Novus Biologicals, NBP2-52671PE).

Antibodies used for immunophenotyping murine transduced splenocytes were rat anti-mouse CD8a conjugated to FITC (1:100, clone 53-6.7, BioLegend, 100706) and rat anti-mouse CD19 conjugated to PE (1:100, clone 6D5, BioLegend, 115508).

For immunophenotyping differentiation and exhaustion state of STEAP1-BBζ CAR T cells from different donors were mouse anti-human CD3 conjugated to BUV395 (5 μl per condition, clone UCHT1BD, Biosciences, 563548), mouse anti-human CD4 conjugated to BV605 (0.5 μl per condition, clone SK3, BioLegend, 344645), mouse anti-human CD8 conjugated to BUV805 (5 μl per condition, clone SK1, BD Biosciences, 564912), mouse anti-human CD45RO conjugated to BV510 (2 μl per condition, clone UCHL1, BioLegend, 304246), mouse anti-human CD45RA conjugated to BV711 (0.1 μl per condition, clone HI100, BioLegend, 304138), mouse anti-human PD-1 conjugated to APC (2 μl per condition, clone A17188B, BioLegend, 621610), mouse anti-human CD95 conjugated to BUV615 (2 μl per condition, clone DX2, BD Biosciences, 752346), mouse anti-human CXCR3 conjugated to BV421 (2 μl per condition, clone G025H7, BioLegend, 353716), mouse anti-human CD62L conjugated to BV785 (2 μl per condition, clone SK11, BD Biosciences, 565311), and mouse anti-human LAG-3 conjugated to BV421 (2 μl per condition, clone T47-530, BD Biosciences, 565721).

## Tumor immune microenvironment analysis using multiparametric flow cytometry

Tumor infiltrating lymphocytes from RM9-hSTEAP1 tumors treated with untransduced T cells and STEAP1-mBBζ CAR T cells with or without CBD-IL-12 were isolated using mechanical dissociation of tumor chunks followed by dispase/collagenase (10 mg/ml) digestion at 37 °C for 1 h. Dissociated cells were treated briefly with trypsin for 5 min followed by syringe dissociation and incubation in 1x RBC lysis buffer for 5 min. Isolated single cells were frozen or processed for downstream analysis. For tumor immune cell phenotyping analysis, $2.5 \times 10^6$ cells were counted and resuspended in PBS followed by incubation in Fc Block buffer (TruStain FcX,101319) for 15 min at room temperature. Cells were subsequently washed twice with PBS and were stained with Fixable Viability Dye (Zombie UV, 1:1000) for 15 min. After washing, cells were stained with a cocktail of anti-mouse cell-surface antibodies including pan-CD45-BUV805 (1:320, clone 30-F11, BD Biosciences); Dump APCF-750 including antibodies against B220/CD45R (1:200, clone RA3-6B2, eBioscience), CD19 (1:200, clone 6D5, BioLegend), CD3 (1:200, clone 145-2C11, BioLegend), CD4 (1:200, clone RM4-5, BioLegend), and CD8a (1:200, clone 53−6.7, BioLegend); CD103-BUV395 (1:200, clone 2E7, BD Biosciences); CD24-BUV496 (1:100, clone M1/69, BD Horizon); Ly6G-BUV737 (1:160, clone 1A8, BD Biosciences); NK1.1-BV421 (1:80, clone PK136, BioLegend); KLRG1-BV510 (1:100, clone 2F1, BD Biosciences); CD11b-BV605 (1:100, clone M1/70, BioLegend); XCR1-BV785 (1:100, clone ZET, BioLegend); F4/80 AF488 (1:100, clone T45-2342, BD Biosciences); Ly6C PerCP-Cy5.5 (1:200, clone HK1.4, BioLegend); CD69 PE (1:100, clone H1.2F3, BioLegend); Siglec-F PE-CF594 (1:400, clone E50-2440, BD Biosciences); and MHC-II A700 (1:400, clone M5/114.15.2, BioLegend). Cells were incubated with the antibody cocktail for 30 min followed by two washes with FACS buffer. Cells were incubated overnight in the Perm wash buffer for membrane permeabilization at 4 ˚C. On the following day, cells were washed once and stained for intracellular proteins including anti Ki-67 BV650 (1:100, clone B56, BD Biosciences), FOXP3 e450 (1:100, clone

FJK-16s, eBioscience) and iNOS APC (1:100, clone CXNFT, eBiosciences) for 30 min. Cells were washed and were analyzed on Cytek® Aurora System. The detailed gating strategy for analysis of different immune cell subsets including neutrophils, eosinophils, monocytes/macrophages, and dendritic cells is provided in Supplementary Fig. 18. The frequencies of individual populations were plotted and statistically analyzed using GraphPad Prism v9.4.1.

## Immunologic co-culture assays

CAR T cell functional assays were performed by co-culturing prostate cancer cells engineered to express GFP with either human or mouse untransduced or STEAP1 CAR T cells at variable effector-to-target (E:T) ratios in 96-well plates. For cytotoxicity assays, 96-well clear bottom black wall plates (Corning) were coated with Cultrex Poly-L-Lysine for 30 min and seeded with prostate cancer cells. Plates were incubated at 37 °C for one hour. Effector cells were then counted and seeded into wells with tumor cells at specified E:T ratios. The plates were placed into a BioTek BioSpa 8 Automated Incubator (Agilent Technologies) and read by brightfield and fluorescence imaging on a BioTek Cytation 5 Cell Imaging Multi-Mode Reader (Agilent Technologies) every six hours for a total of six days. Target cells were quantified based on the number of GFP+ objects identified per scanned area using BioTek Gen5 Imager Software (Agilent Technologies). To assess T cell activation based on cytokine release, 25−50 μl of co-culture supernatants were collected at 24 and 48 h of co-culture and stored at −30 °C.

## Mitogen stimulation of T cells

T cells were stimulated with 5 ng/ml of phorbol 12-myristate 13-acetate (PMA, Sigma Aldrich) and 250 ng/ml of ionomycin (Sigma Aldrich) or with 5 μg/ml of phytohemagglutinin-L (PHA-L, Sigma Aldrich). Supernatants were collected at 24 h for use as a positive control for IFN-γ ELISA studies or T cells were used as a positive control for induction of the exhaustion markers PD-1 and LAG-3 as assessed by flow cytometry.

## Enzyme-linked immunosorbent assay

To determine IFN-γ levels in co-culture supernatants, samples were thawed and sandwich ELISA for human or mouse IFN-γ levels was performed using the BD Human IFN-gamma ELISA Set (BD Biosciences, 555142) or BD Mouse IFN-gamma ELISA Set (BD Biosciences, 555138) according to the manufacturer's protocol. Plates were read at 450 nm and 560 nm wavelengths using a BioTek Cytation 3 Cell Imaging Multi-Mode Microplate Reader (Agilent Technologies). Plasma samples isolated from retroorbital bleeds were used for ProcartaPlex immunoassays (Thermo Fisher) to quantify levels of mouse IFN-γ, IL-2, IL-6, IL-4 and TNF-α according to the manufacturer's protocol. Samples were assayed on a Luminex 100/200 System (Luminex). Anti-human IgG and IgM antibodies from the sera of mice treated with untransduced T cells or STEAP1-mBBζ CAR T cells were quantified using MAHA (Mouse Anti-Human Antibody) ELISA Assay Kit (Eagle Biosciences, MAH11-K01).

## Bulk RNA-sequencing and analysis

Total RNA was isolated from 22Rv1 wt, 22Rv1 STEAP1 ko, and 22Rv1 STEAP1 ko + rescue cells using the Purelink™ RNA Mini Kit (Thermo Scientific). Samples with RNA integrity number (RIN) greater than 8.0 were processed for library preparation using the Illumina TruSeq Stranded mRNA kit and sequencing on an Illumina NovaSeq 6000 with $2 \times 100$ bp paired-end reads by the Fred Hutch Genomics and Bioinformatics Shared Resource.

For RNA-seq on the 22Rv1 metastatic tumor samples from animals treated with untransduced T cells or STEAP1-BBζ CAR T cells, total RNA was isolated using the RNeasy FFPE Kit (Qiagen). DV200 were assessed by TapeStation (Agilent) and tumor samples with DV200 > 30% were selected for library preparation using the Illumina TruSeq RNA Exome

kit and sequencing on an Illumina NovaSeq 6000 with $2 \times 50$ bp paired-end reads by the Fred Hutch Genomics and Bioinformatics Shared Resource.

Data quality check for RNA-seq results were performed using NovaSeq Control Software (v1.8.0) and demultiplexing was performed with the Illumina Bcl2fastq2 (v.2.20) program. FASTQ files were processed using the UCSC TOIL (v5.7.1) processing pipeline[83]. Reads were assessed for quality control using FASTQC and aligned to the hg38 human reference genome. A count matrix filtered for coding genes using HUGO and containing raw log2 transformed gene expression values was generated. Differential gene expression analysis was performed using DESeq2 (v3.15) with default parameters[84]. Pathway analysis was performed using Gene Set Enrichment Analysis (v4.3.1)[85] and the cell cycle progression (CCP) score was calculated as previously described[54]. Bioinformatic analysis was performed using R (v4.0.3) and R studio (v1.3.1093).

## Single-cell RNA-sequencing and analysis

Single-cell libraries of RM9 tumor cells from mice treated with vehicle or CBD-IL-12 were generated per the 10x Genomics Chromium platform and sequenced on an Illumina NovaSeq 6000 in the Fred Hutch Genomics and Bioinformatics Shared Resource. Reads were aligned to the GRCm38/mm10 mouse reference genome and quality control and data processing was performed using Seurat V4[86]. Annotation of cell type was performed using the Bioconductor package SingleR v1.10.0[87] based on the ImmGenData reference.

## TCR repertoire analysis

Genomic DNA from metastatic lung tumors were isolated using the GeneJET Genomic DNA Purification Kit (Thermo Fisher). Subsequent TCRV-β sequencing was performed on the immunoSEQ platform (Adaptive Biotechnologies). Productive (in-frame) TCR rearrangements were used for the clonality assessment based on Simpson clonality index using immunoSEQ Analyzer 3.0 (Adaptive Biotechnologies). Non-parametric multiple comparison Dunn's test was used to estimate the significance between means of each group and adjusted p-values are reported.

## Reporting summary

Further information on research design is available in the Nature Portfolio Reporting Summary linked to this article.

## Data availability

Sequencing data pertaining to this study is available from Gene Expression Omnibus (GEO) as SuperSeries GSE214585. RNA-seq data from the parental 22Rv1, 22Rv1 STEAP1 ko, and 22Rv1 STEAP ko + rescue cells lines and 22Rv1 tumors treated with untransduced T cells or STEAP1-BBζ CAR T cells (resulting in STEAP1 antigen loss) is available from accession number GSE214583. Single-cell RNA-seq (scRNA-seq) data for RM9 tumors treated with vehicle or CBD-IL-12 is available from accession number GSE214584. T cell receptor (TCR) beta sequencing profiles are available from accession number GSE228884 and also as part of the ImmuneCODE data resource available to be downloaded from the Adaptive Biotechnologies immuneACCESS site under the immuneACCESS Terms of Use at: https://doi.org/10.21417/VB2023NC. Source data are provided with this manuscript in the Source Data file. All remaining data related to the study are included in the article, Supplementary Information and Source Data file. Source data are provided with this paper.

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

## Acknowledgements

We are grateful first and foremost to the patients and their families for their contributions, without which this research would not have been possible. We acknowledge Celestia Higano, Evan Yu, Heather Chang, Bruce Montgomery, Elahe Mostaghel, Andrew Hsieh, Daniel Lin, Funda Vakar-Lopez, Xiaotun Zhang, Lawrence True, and the rapid autopsy teams for the contributions to the University of Washington Prostate Cancer Donor Rapid Autopsy Program. We thank the Fred Hutch Co-Operative Center for Excellence in Hematology (supported by U54 DK106829). We also thank the Fred Hutch Genomics Shared Resource, Comparative Medicine Shared Resource, Flow Cytometry Shared Resource, Experimental Histopathology Shared Resource, and Immune Monitoring Shared Resource (supported by NIH/NCI Cancer Center Support Grant P30 CA015704). This research was funded in part by a Department of Defense Prostate Cancer Research Program Award (W81XWH-21-1-0581 to J.K.L.), a Fred Hutch/University of Washington Cancer Consortium Safeway Pilot Award (J.K.L.), a Seattle Cancer Care Alliance/Swim Across America Award (J.K.L.), the Pacific Northwest Prostate Cancer SPORE P50 CA097186 (P.S.N., L.T., and R.G.), the Institute for Prostate Cancer Research (M.P.R), the Doris Duke Charitable Foundation (2021184 to M.C.H), NCI R50 CA221836 (R.G.), JSPS Overseas Research Fellowships (202160429 to K.S.), Prostate Cancer Foundation Young Investigator Award (V.B.), Department of Defense Prostate Cancer Research Program Early Investigator Research Award (HT9425-23-1-0089 to V.B.), and a Movember Foundation-Prostate Cancer Foundation Challenge Award (P.S.N. and J.K.L.).

## Author contributions

V.B. Data curation, formal analysis, validation, investigation, visualization, methodology, writing-original draft, writing-review and editing. N.V.K. Data curation, formal analysis, investigation, visualization, and methodology. T.E.P. Conceptualization, data curation, formal analysis, investigation, visualization, and methodology. L.W.: Data curation and investigation. A.T. Data curation and investigation. H.S. Data curation and investigation. G.J. Data curation and investigation. S.N. Data curation and investigation. I.C. Data curation and investigation. K.S. Data curation and investigation. L.H. Data curation and investigation. A.Z. Data curation and investigation. D.R.: Data curation and investigation. R.G. Formal analysis. R.P. Data curation and investigation. M.P.R. Formal analysis. L.T. Data curation and writing-review and editing. S.S. Data curation and investigation. C.M.M. Data curation and investigation. M.C.H. Data curation, formal analysis, and writing-review and editing. P.S.N. Conceptualization and writing-review and editing. S.J.P. Formal analysis, methodology, writing-review and editing. J.I. Formal analysis, validation, investigation, and writing-review and editing. J.K.L. Conceptualization, data curation, formal analysis, validation, investigation, visualization, methodology, writing-original draft, writing-review and editing.

## Competing interests

T.E.P. and J.K.L. are inventors on U.S. Provisional Patent Application No. 63/309389 entitled "Chimeric Antigen Receptors Binding STEAP1"

related to this work. J.I. is a co-founder and shareholder of Arrowimmune, Inc. J.I. is a scientific advisor of Libo Pharma Corp. The remaining authors declare no competing interests.
