## [Peer Review File · Nature Communications]

Targeting advanced prostate cancer with STEAP1 chimeric antigen receptor T cell and tumor-localized IL-12 immunotherapyREVIEWER COMMENTS

Reviewer #1 (Remarks to the Author): with expertise in prostate cancer, STEAP1

It is a great article focusing the potential use of STEAP1 as immunotherapeutic target for prostate cancer. The authors started to evaluate and compare the expression of STEAP1 and PSMA proteins in lethal metastatic prostate cancer, demonstrating that targeting STEAP1 presents advantages in comparison with PSMA. Next, the authors used the potential therapy based in "chimeric antigen receptor (CAR) T cell to recognize the cell overexpressing the STEAP1 protein. With this construct it was observed that STEAP1 antigen loss was associated with diminished tumor antigen processing and presentation. Then, the authors have engineered the construct with IL-12 as an adjunct to STEAP1 CAR T cell therapy, showing promising results in in vitro and in vivo preclinical models. The experimental design is very well established and the conclusions are in accordance with obtained results. Several experimental controls were used to validate the results. I suggest some improvements before acceptance for publication:

1. In supplemental Figure 1D and 1E, it must be added immunohistochemical image staining the AR and synaptophysin. Ideally, it should be evaluated the co-localization of STEAP1 and AR/synaptophysin by confocal microscopy.
2. The text should be reviewed in order to eliminate some grammatical mistakes.

Reviewer #2 (Remarks to the Author): with expertise in prostate cancer, immunotherapy, CAR-T

The authors describe a new CAR T cell approach for targeting STEAP1, an antigen which they showed to be highly expressed in tissues of patients with metastatic prostate cancer. The CAR T cell therapy was tested in different mouse models. Antigen escape was identified as mechanism of therapeutic resistance. Combination of CAR T cell therapy with IL-12 immunotherapy was used for the remodeling of the tumor microenvironment and enhanced therapeutic outcome by engagement of host immunity and epitope spreading. The manuscript is well written and the findings from the experiments are easy to follow. At a few points, more information is necessary from a methodological point of view. Moreover, more data should be presented from the three CAR T cell versions that were generated, discussion about crossreactivity of the CAR T cells with murine STEAP1 and hSTEAP1B isoforms needs more clarity and Figure 7 needs some revisions.

Major points of revision

1. Anti-STEAP1 CARs with different spacers are presented in Fig. 2A. The authors write that only the candidate with the long spacer showed activity. Comparative data from all three candidates should be added in a new suppl. figure.
2. Lines 271-292: In the comparison between human and mouse STEAP1 the authors found that the ECD2 domain with high homology between the two species was associated with anti STEAP1 CAR T cell activation. Moreover, they identified Q198 and/or I209 of human STEAP1 as critical to productive recognition by the CAR T cells (suppl. Fig. 3F). On the other side, no CAR T cell activation was found on PC cells expressing the STEAP1B isoforms that share the ECD2 domain with STEAP1 including Q198 and I209 (suppl. Fig. 4A). This should be clarified.
3. In Fig. 7B, only BLI pictures from days -1, 9, and 17 are shown. Fig. 7C shows that more BLI pictures were made on other days of the therapy. These picture should be added to Fig. 7B.
4. Fig. 7C and 7D do not seem to correspond. For example, In Fig. 7C one animal survived until day 17 in the „untransduced cells + CBD-IL-12“ group. In Fig. 7D, however, it seems that two animals of this group survived until day 17. This should be corrected.
5. In Fig. 7F, there is no significant difference in the number of CD3+ cells in tumors treated with CAR T cells alone compared to CAR T cells + CDB-IL-12. This means that there is no enhanced CAR T cell infiltration into tumors treated with CDB-IL-12. This should be discussed in the context of the experiment.

Minor points of revision

1. Lines 93-104: References should be added.

2. line 139: the NCT number should be added.
3. line 214: what is meant with "1 nM affinity" (Kd value?)
4. lines 238-240: STEAP1 expression of DU145 STEAP1 cells should be shown in an additional suppl. figure.
5. Lines 324-326: The authors state a statistically significant inhibition of tumor growth in the CAR T cell group by day 16, however it seems that the significant effect occurred only from day 18 on (Fig. 3A). This should be corrected.
6. For a better understanding PMA/I should be described in the legends of suppl. Fig. 2B and 3C.
7. Termination criteria for killing of the animals with disseminated tumors in the experiments investigating OS should be added in the methods section.
8. Suppl. Fig. 11E: there seems to be a wrong numbering of the animals of the CAR T cells group (no. 6-10 instead of 5-9).
9. In Fig. 11G, data of only two animals in the group treated with untransduced T cells are shown, although five animals were used in the experiment. This should be clarified or corrected.
10. Fig. 9: It is not clear what is in the dishes (organs from one or more mice?). The control image is over-illuminated so that no organs or tumor lesions are visible. The images should therefore be improved.
11. Suppl. Fig. 15: The bars showing INF- γ release in the „untransduced T cells + CDB-IL-12 group“ (day 0 vs day 8) suggest a significant difference. If so, the p value should be added.

Reviewer #3 (Remarks to the Author): with expertise in IL12 immunotherapy, CAR-T

- What are the noteworthy results?

Histology based the authors acknowledge/confirm STEAP1 as a superior target for PC, present on especially those cancer cells that are identified in relapse/disseminated disease, metastases, late stage disease. STEAP is involved in tumor progression, antibodies inhibit local and metastatic Ewing sarcoma growth. Would be interesting to know whether tumor stem cells also express it.

The authors then developed STEAP1-directed chimeric antigen receptor (CAR) T cells with fully humanized scFv from vedotin which possesses high binding affinity. To tune CAR activity different hinge regions were implemented and different constructs tested. Only the long spacer STEAP1 CAR met expectations regarding IFN release and killing activity.

Lead STEAP1 CAR T cells then demonstrated reactivity towards even low antigen density and killed diverse metastatic human and mouse prostate cancers. Application seemed safe in a human STEAP1 knock-in mouse model.

STEAP1 CAR were not cross reactive towards xenogeneic mouse STEAP1 or human STEAP2. Bhati et al. identified ECD2 as mandatory for STEAP1 recognition on the target, and this domain did not recognize 3 STEAP1B variants despite high sequence homology to STEAP1.

Next, the authors characterize STEAP1 CAR T cell products generated with T cells from different donors. They report a 20-40fold expansion of STEAP CAR T in 11 days, with a slightly higher proliferative capacity in CD4+ CAR T than CD8+ CAR T. PD-1 and LAG-3 expression was the same in un-transduced versus STEAP1 CAR T cells, which the authors assume is due to absent tonic (constitutive) signaling, which is positive for effector function (less exhaustion).

Immunophenotyping of un-transduced versus STEAP1 transduced CAR T subset revealed a higher proportion of Tscm in the latter. The authors suggest that IL-7 and or IL-15 in the culture might preserve /enhance this phenotype. Tcm was particularly enriched in the CD8+ STEAP CAR compartment.

STEAP CAR monotherapy applied once induced delay of tumor growth within 16 days. Day 25 explanted tumors showed necrotic areas and CAR T infiltration in the treatment cohort.

The authors applied CAR T intratumorally here: why? If the target is superior, specific and expressed? Also, with regard to clinical application: in most disseminated disease no intra tumoral application is possible.

In vivo disseminated disease was then treated intravenously with CAR T and was associated with reduced and delayed marker expression of labelled cancer cells. Histology showed loss of target antigen day 97 versus 31.

The authors then provide transcriptomics associated with STEAP loss, which modulates most impressively cell cycle regulation genes, Krebs cycle/Glycolysis, downregulation of antigen processing and presentation, MHC expression. The latter was confirmed also by histologic analysis in CAR T treated tumors, which is a key finding!

STEAP CAR therapy resulting in antigen loss and impaired antigen processing and presentation. This raises the question of a general mechanism of action, which is also paramount in hematological malignancies treated with CAR T.

Castration resistant PC with growth kinetics typical for human PC in vivo was established as disseminated disease. Single CAR T iv induced remission in all individuals of the treatment cohort, and was confirmed macroscopically and with BLI. Survivors spleens harbored respective CAR effector cells.

Due to reactivity to even very low antigen density the authors in a hSTEAP1 transgene mouse examined potential off-target toxicities of iv administered CAR T (engineered with respective adaptations to the model system) in a model system.

Treatment in NSG mice showed rapid disease regression followed by relapse 10 days later, survival was statistically significantly better (22 vs 12 d). Interestingly: Antigen loss was also observed in this model in pulmonary disseminated disease in the CAR T cohort. Whether this is a general side effect/phenomenon of CAR T therapy or the physiological response to an immune response irrespective of whether mediated by CAR T, NK or ab T cells would be interesting to investigate. Should be discussed.

The authors conclude that adjunct therapies are needed to overcome resistance in subgroups of advanced cancer patients where inter- or intra tumoral STEAP1 heterogeneity is present. This is a major contribution to the field highlighting that further therapy is necessary to intercept the undesirable formation of resistance by CAR. To my knowledge this is also the first report which sees CAR T as causal for the antigen loss, and compares treatment cohort with ctrl. in this respect.

Safety and efficacy of therapy were also examined in a humanized mouse model with non-clonal RM9-STEAP1 cells which gave results similar to that seen in the NSG model.

CAR T induced no cross toxicities or premature death of treated individuals while at the same time showing clear evidence for anti-tumor efficacy. Comparable infiltration rate of CD3+ and same degree of integrity of tissues (prostate and adrenal gland) in treatment and ctrl cohort. Again, and interestingly: Lung explants at the end of experiment show hSTEAP1 in some regions of lung metastases in untreated cohort, yet complete antigen loss in CAR T cohort, associated with significant downregulation of MHC class I.

This finding let the authors seek for additional therapeutic intervention to counter regulate antigen loss.

The authors introduce fusion cytokine CBD-IL12 which showed efficacy to revert tumor stroma in vivo, the authors show in RM9 tumors on scRNA level: restructure of the TME, enhanced proteasomal activity, enhanced MHC expression and antigen presentation, and via histology: increase in T cell infiltration.

Next the authors tested un-transduced T, vs STEAP CAR T vs un-transduced T + CBD-IL12 vs STEAP CAR T + CBD-IL12 in non-clonal RM9 STEAP1 metastases in hSTEAP1-KI, preconditioned with CPA. Administration of therapy was 1/w for 3 weeks. Only CAR T combined with IL-12 increased survival and was associated with increase in plasma TH1 cytokines. Antigen loss was observed in CAR T only and CAR T+IL-12. Increased MHC class I and Ti CD3+ were observed in CAR T and CAR T+IL12.

comments to the authors:

The authors may mention that antigen loss might be due to tumor editing which would be in line with that IL12 alone and IL-12+CAR T increased TCR diversity in tumor metastases of the lung.

The work would clearly benefit if histological analyses were shown that indicate reversion of the stroma on protein level: Infiltration of T cells, M1, M2 macrophages, NK, iNKT cells, Treg cells. scRNA analysis was not used to analyse the type of infiltrating T cells, e.g. based on the lineage TFs, cytokines, cytolytic markers, expression of activating NKps or corresponding ligands on

Tumor cells, why? Transcriptomics can give a hint but do not correlate with protein expression.

20 days post therapy was the endpoint of the combination therapy study, why? This is too short for immunotherapeutic interventions and to show CAR T plus tumor targeted IL12 can induce long-lasting remission/survival in solid malignancy. It would also have been interesting to analyse the one tumorbearing mouse: was the stroma reverted?, less infiltration, less marker expressed that may make the Tumor susceptible to innate and adaptive immunity, less TH1 cytokines in periphery?, did the Tumor show signs of senescence (terminal growth arrest due to cellular senescence induced by TH1 cytokines? which would positively impact survival....

Thus for comprehensiveness of the study more details on stroma reversion by immune cytokine CBD-IL12 are needed.

- Will the work be of significance to the field and related fields? How does it compare to the established literature? If the work is not original, please provide relevant references.

The work is of significance to the field in several aspects:

The authors use a new target for CAR T therapy, which is also on aggressive disseminated cancer cells, thus a "superior" target on lethal PC.

In contrast to the previous gold standard PSMA, STEAP1 is also found on treatment-resistant mCRPC.

The authors acknowledge that CAR T alone is insufficient to cure the disease, which most authors do not address with such clarity

They pinpoint CAR T as causative for antigen loss, which is a relevant important finding!

Immunogenicity, delineated from proteasome activity, MHC class I and II expression, antigen processing and presentation, is negatively enriched in CAR T treated tumors, a finding that is of superior significance.

They try to compensate antigen loss by adjuvant tumor targeted IL-12, since IL12 - as they show in RM9 tumors treated with IL-12FP -upregulates Ag presentation machinery and MHC class I.

The authors may discuss the fact that also IL12 therapy lead to antigen loss, thus did not fulfill the authors expectation to prevent antigen loss.

However, the important question of whether the cause is the same in both cases has not been clarified!

Is antigen loss observed because antigen positive tumour cells downregulate the antigen in the presence of CAR T or because they are killed? The former being more likely and consistent with reduced "antigen presenting machinery" results.

(STEAP1 is involved in proliferation which can be modulated by TH1 cytokines/metabolism/... in a reciprocal crosstalk between tumour and immune compartment).

In the IL12 settings, tumor editing may have caused loss of antigen in STEAP1 heterogenous Tumor cells due to increased effector function of innate and adaptive immune cells and CAR T cells, thus quantitative killing of all STEAP1 positive Tumor cells may have occurred.

In general:the authors may emphasise that this question has to be answered and that the cross talk of tumour and effector cells and/or stroma needs to be investigated in more detail.

The production of different STEAP1 targeting CAR T constructs is gaplessly reported, and the CAR T characterized in detail regarding killing efficacy, required antigen density.

The CAR T is produced in different donors and immunophenotyping reveals preferential TSCM type.Cross-reactivities were evaluated in elaborated in vivo models and were negative.

The authors aim to develop an efficient immunotherapeutic approach combining CAR T and immune cytokine, which is a new and promising therapeutic option. They use state of the art and elaborated techniques for preclinical testing of the constructs to make the application safe. A step further towards clinical application of combined immune therapies.

- Does the work support the conclusions and claims, or is additional evidence needed?

It is a well-founded, detailed and comprehensive study in terms of the identification of the CAR target, the production of the CAR and also the question of safety and efficacy of the CAR in monotherapies. The part where the CAR is combined with stromal reversion (IL-12 fusion cytokines) is not totally convincing since stroma reversion is downregulated/extinguished at the cellular level by CPA and tumor growth negatively impacted, thus true efficacy of IL-12 fusion cytokine is not clear. CPA is an unknown variable that has furthermore only been used in some but not all experimental groups. Without being commented or justified.

Testing with full preservation of stromal reversion plus CAR combination therapy is missing, yet essential and is required to find out the potential of combined therapy.

- Are there any flaws in the data analysis, interpretation and conclusions? Do these prohibit publication or require revision?

Testing combination therapy:

Line 753: For animal testing combined IL12 and CAR therapy ... group d) is incorrect, should also include CBD-IL-12.

Why did group a) (Un-transduced T cells) and d) (CAR +IL12) receive preconditioning?

That clearly creates different experimental conditions, doesn't it?

Endogenous activated proliferating NK and T cells are very likely negatively modulated (eliminated?) by CPA treatment, the full potential of the treatment (CAR + IL12) seems not shown here.

In d) the effect of stroma reversion is more or less (degree of impact is not analyzed) reduced to antigen-presentation machinery, which may lead to optimum tumor editing by CAR T (might explain loss of antigen?). The crosstalk of endogenous innate and adaptive immunity with stroma and tumor remains biased/incomplete/manipulated.

CPA also negatively impacts tumor growth and proliferation (to which extend?)

In this context: It should be discussed why antigen loss in "a priori" "heterogeneous STEAP1-expressing RM9 tumors" is quantitative in CAR treated tumor metastasis, but not so in ctrl. (see above).

Duration/treatment phase of the combination treatment study is very short, this and the manipulation of tumor and immune compartment with CPA might have made it less likely to identify the full potential of the combination therapy.

Therapy for a period of time of at least 35-42 days (5-6 weeks instead of 3) might have shown more of the potential of respective treatments.

Combination therapy resulting in better but not good outcome is not sufficient with regard to clinical application. The authors clearly address this in the discussion.

They may hypothesise why it is that no better results are achieved?

Other reports combine IL12 immune cytokines with irradiation, and or additional cytokines (IL-2) to support T cell responses established by a reverted stroma.

Also CI might be mentioned as an additional therapeutic add on.

- Is the methodology sound? Does the work meet the expected standards in your field?

Methodology is state of the art and comprehensive. Except for CPA preconditioning (see above)

- Is there enough detail provided in the methods for the work to be reproduced?

Yes.

Taken together: The questions raised in the comments should be answered. However, this work is of paramount interest to a broad readership including clinicians, translational researchers, immunologists.

Reviewer #4 (Remarks to the Author): with expertise in CAR-T

Comment for the authors

The authors present a comprehensive data set and very compelling story of targeting STEAP1 in prostate cancer with CAR T cells. The manuscript is very well written, the methods are elaborate and even though prostate cancer is not “cured” (even in combination of STEAP1 CAR T cells with the CBD-IL12), this likely provides a realistic perspective also for clinical translation.

Major comments:

1. STEAP1 expression: The authors state that up to three cores from the same patient, but obtained from different tumor sites (primary vs. metastatic) were analyzed. Can the authors add data on the homogeneity/heterogeneity of STEAP1 expression in these different tumor sites in a given patient and comment on how this may impact efficacy?
2. STEAP1 vs. PSMA expression: The authors show loss of STEAP1 expression in their in vivo models. At the time of relapse with (potential neuroendocrine trans-differentiation) and STEAP1 loss – is PSMA expression retained? Any data on dual-antigen targeting of STEAP1 together with PSMA?
3. CAR design: The authors use a modified IgG4 spacer that due to the elimination of Fc motifs may be immunogenic in humans. Also in mice, this spacer is very likely immunogenic. Please include any data on immunogenicity or immune responses that have been observed, at least in the immunocompetent mouse models.
4. Toxicology testing: Can the authors present data on STEAP CAR T cell transfer into non-tumor bearing mice (that have the human STEAP1 KI)? Since all of the mice in the current experiment die from progressive tumor, the claim of “no deaths” from targeting STEAP1 is hard to hold up.
5. CBD-IL12: Any data on the effect of CBD-IL-12 alone in the immunocompetent mouse models? Since there is systemic administration and biodistribution – any toxicity? Any data on innate immune cell activation and migration to the tumor site(s)? Any data to demonstrate that CBD-IL12-activated innate immune cells are capable of eliminating STEAP1-positive and negative tumor cells?

Response to Reviewers' Comments

Reviewer 1 Comments (with expertise in prostate cancer, STEAP1):

It is a great article focusing the potential use of STEAP1 as immunotherapeutic target for prostate cancer. The authors started to evaluate and compare the expression of STEAP1 and PSMA proteins in lethal metastatic prostate cancer, demonstrating that targeting STEAP1 presents advantages in comparison with PSMA. Next, the authors used the potential therapy based in "chimeric antigen receptor (CAR) T cell to recognize the cell overexpressing the STEAP1 protein. With this construct it was observed that STEAP1 antigen loss was associated with diminished tumor antigen processing and presentation. Then, the authors have engineered the construct with IL-12 as an adjunct to STEAP1 CAR T cell therapy, showing promising results in in vitro and in vivo preclinical models. The experimental design is very well established, and the conclusions are in accordance with obtained results. Several experimental controls were used to validate the results. I suggest some improvements before acceptance for publication:

Comment 1. In supplemental Figure 1D and 1E, it must be added immunohistochemical image staining the AR and synaptophysin. Ideally, it should be evaluated the co-localization of STEAP1 and AR/synaptophysin by confocal microscopy.

Response: We thank the reviewer for the very positive response. We have addressed this comment by now including **Fig. 1e** which summarizes the expression of STEAP1 and PSMA in metastatic tumor cores coded by molecular subtype based on androgen receptor (AR) and synaptophysin (SYP) expression (AR+/SYP+, AR+/SYP-, AR-/SYP+, AR-/SYP-) and **Supplementary Fig. 2a** which shows representative tumor sections stained by IHC for AR, SYP, and STEAP1 expression.

Comment 2: The text should be reviewed to eliminate some grammatical mistakes.

Response: We have reviewed the manuscript text and edited for grammar.

Reviewer 2 Comments (with expertise in prostate cancer, immunotherapy, CAR-T):

The authors describe a new CAR T cell approach for targeting STEAP1, an antigen which they showed to be highly expressed in tissues of patients with metastatic prostate cancer. The CAR T cell therapy was tested in different mouse models. Antigen escape was identified as mechanism of therapeutic resistance. Combination of CAR T cell therapy with IL-12 immunotherapy was used for the remodeling of the tumor microenvironment and enhanced therapeutic outcome by engagement of host immunity and epitope spreading. The manuscript is well written and the findings from the experiments are easy to follow. At a few points, more information is necessary from a methodological point of view. Moreover, more data should be presented from the three CAR T cell versions that were generated, discussion about crossreactivity of the CAR T cells with murine STEAP1 and hSTEAP1B isoforms needs more clarity and Figure 7 needs some revisions.

Major points of revision:

Comment 1: Anti-STEAP1 CARs with different spacers are presented in Fig. 2A. The authors write that only the candidate with the long spacer showed activity. Comparative data from all three candidates should be added in a new suppl. figure.

Response: We thank the reviewer for the careful review of this manuscript and constructive comments. We have added **Supplementary Fig. 3c** to show data from the co-culture of short and medium spacer STEAP1 CAR T cells with parental 22Rv1, 22Rv1 STEAP1 ko, and 22Rv1 STEAP1 ko + rescue cells.

Comment 2: Lines 271-292: In the comparison between human and mouse STEAP1 the authors found that the ECD2 domain with high homology between the two species was associated with anti STEAP1 CAR T cell activation. Moreover, they identified Q198 and/or I209 of human STEAP1 as critical to productive recognition by the CAR T cells (suppl. Fig. 3F). On the other side, no CAR T cell activation was found on PC cells expressing the STEAP1B isoforms that share the ECD2 domain with STEAP1 including Q198 and I209 (suppl. Fig. 4A). This should be clarified.

Response: We appreciate the opportunity to clarify these points. STEAP1 CAR T cells are not activated when co-cultured with DU145 cells expressing mouse Steap1 (**Supplementary Fig. 4d, e**). When we take the mouse Steap1, replace its ECD2 with the ECD2 of human STEAP1, and express this chimeric protein (mSteap1 hECD2) in DU145 cells, we find that STEAP1 CAR T cells are activated in co-cultures with DU145-mSteap1 hECD2 cells (**Supplementary Fig. 4g**). While there is homology between the ECD2 of human STEAP1 and the STEAP1B isoforms, the crystal structure of STEAP1B has not been resolved and it is unknown whether this region of homology is extracellular in STEAP1B isoforms. We show data from an *in silico* membrane topology prediction tool TOPCON showing low reliability in the prediction that this region of homology in STEAP1B isoforms is extracellular. To empirically determine whether STEAP1 CAR T cells may cross-react with STEAP1B, we overexpressed each of the STEAP1B isoforms in DU145 cells and performed co-cultures with STEAP1 CAR T cells. Our data indicated that STEAP1 CAR T cells are activated in the presence of human STEAP1 but not mouse Steap1 or human STEAP1B isoforms (**Supplementary Fig. 5d**). We have clarified these findings in the Results section of the revised manuscript.

Comment 3: In Fig. 7B, only BLI pictures from days -1, 9, and 17 are shown. Fig. 7C shows that more BLI pictures were made on other days of the therapy. These picture should be added to Fig. 7B.

Response: As per the reviewer's suggestion we have added the complete set of BLI images for the mice experiment in **Fig. 7b** in the revised manuscript.

Comment 4: Fig. 7C and 7D do not seem to correspond. For example, In Fig. 7C one animal survived until day 17 in the "untransduced cells + CBD-IL-12" group. In Fig. 7D, however, it seems that two animals of this group survived until day 17. This should be corrected.

Response: We apologize for this error while plotting the graphs. We have corrected this in the revised manuscript.

Comment 5: In Fig. 7F, there is no significant difference in the number of CD3+ cells in tumors treated with CAR T cells alone compared to CAR T cells + CDB-IL-12. This means that there is no enhanced CAR T cell infiltration into tumors treated with CDB-IL-12. This should be discussed in the context of the experiment.

Response: We agree with the reviewer that a significant difference in intratumoral CD3⁺ cells in tumors treated with CAR T cells alone compared to CAR T cells + CBD-IL-12 was not observed. However, we now show in **Fig. 8c-g** and **Supplementary Fig. 19** that CBD-IL-12 clearly modulates other aspects of the tumor immune microenvironment including increases in M1 macrophages and antigen cross-presenting dendritic cells as well as decreases in immunosuppressive neutrophils and T regulatory cells (Tregs). These results are discussed in the context of the experiment in the revised manuscript.

Minor points of revision:

Minor Comment 1: Lines 93-104: References should be added.

Response: The relevant references have been added in the revised manuscript.

Minor Comment 2: line 139: the NCT number should be added.

Response: The relevant clinical trial number NCT04221542 has been added in the revised manuscript.

Minor Comment 3: line 214: what is meant with “1 nM affinity” (Kd value?)

Response: The 1 nM affinity was reported previously (*Challita-Eid PM et al. Cancer Res. 2007.*). Per the publication, cells were incubated with increasing concentrations of STEAP1 antibody, washed, incubated with phycoerythrin-conjugated secondary antibody, and analyzed by flow cytometry. The affinity was calculated based on nonlinear regression of antibody concentration plotted against mean fluorescence intensity.

Minor Comment 4: lines 238-240: STEAP1 expression of DU145 STEAP1 cells should be shown in an additional suppl. figure.

Response: Immunoblot analysis showing the expression of human STEAP1 and mouse STEAP1 in engineered DU145 cells is now presented in **Supplementary Fig. 4a**.

Minor Comment 5: Lines 324-326: The authors state a statistically significant inhibition of tumor growth in the CAR T cell group by day 16, however it seems that the significant effect occurred only from day 18 on (Fig. 3A). This should be corrected.

Response: We have corrected this in the text of the revised manuscript.

Minor Comment 6: For a better understanding PMA/I should be described in the legends of suppl. Fig. 2B and 3C.

Response: We have added a description of the PMA/I abbreviation in the supplementary figure legends.

Minor Comment 7: Termination criteria for killing of the animals with disseminated tumors in the experiments investigating OS should be added in the methods section.

Response: The termination criteria for euthanizing the mice in the disseminated mice model studies have been elaborated in the revised material and method section of the manuscript.

Minor Comment 8: Suppl. Fig. 11E: there seems to be a wrong numbering of the animals of the CAR T cells group (no. 6-10 instead of 5-9).

Response: We have corrected the numbering in the revised **Supplementary Fig. 12e, f**.

Minor Comment 9: In Fig. 11G, data of only two animals in the group treated with untransduced T cells are shown, although five animals were used in the experiment.

Response: Unfortunately, we were unable to collect the spleens from the remaining three mice in the untransduced group as they were found dead prior to compassionate euthanasia endpoints. Thus, the graph consists of data from remaining two mice treated with untransduced T cells.

Minor Comment 10: Suppl. Fig. 9: It is not clear what is in the dishes (organs from one or more mice?). The control image is over-illuminated so that no organs or tumor lesions are visible. The images should therefore be improved.

Response: The wells each contain liver or lungs from a single mouse. We have modified the figure to include photographic images with and without the bioluminescence overlay to **Supplementary Fig. 10b** to enable better visualization of organs and tumors in the control group.

Minor Comment 11: Suppl. Fig. 15: The bars showing INF- γ release in the “untransduced T cells + CDB-IL-12 group” (day 0 vs day 8) suggest a significant difference. If so, the p value should be added.

Response: Statistical analysis of IFN- γ release showed no significant difference between untransduced T cells + CBD-IL-12 treatment (day 0 vs day 8). This is likely due to the large standard deviation between the mouse samples.

Reviewer 3 Comments (with expertise in IL12 immunotherapy, CAR-T):

What are the noteworthy results?

Histology based the authors acknowledge/confirm STEAP1 as a superior target for PC, present on especially those cancer cells that are identified in relapse/disseminated disease, metastases, late-stage disease. STEAP is involved in tumor progression, antibodies inhibit local and metastatic Ewing sarcoma growth. Would be interesting to know whether tumor stem cells also express it. The authors then developed STEAP1-directed chimeric antigen receptor (CAR) T cells with fully humanized scFv from vedotin which possesses high binding affinity. To tune CAR activity different hinge regions were implemented and different constructs tested. Only the long spacer STEAP1 CAR met expectations regarding IFN release and killing activity. Lead STEAP1 CAR T cells then demonstrated reactivity towards even low antigen density and killed diverse metastatic human and mouse prostate cancers. Application seemed safe in a human STEAP1 knock-in mouse model. STEAP1 CAR were not cross reactive towards xenogeneic mouse STEAP1 or human STEAP2. Bhatia et al. identified ECD2 as mandatory for STEAP1 recognition on the target, and this domain did not recognize 3 STEAP1B variants despite high sequence homology to STEAP1.

Next, the authors characterize STEAP1 CAR T cell products generated with T cells from different donors. They report a 20-40fold expansion of STEAP CAR T in 11 days, with a slightly higher proliferative capacity in CD4+ CAR T than CD8+ CAR T. PD-1 and LAG-3 expression was the same in un-transduced versus STEAP1 CAR T cells, which the authors assume is due to absent tonic (constitutive) signalling, which is positive for effector function (less exhaustion). Immunophenotyping of untransduced versus STEAP1 transduced CAR T subset revealed a higher proportion of Tscm in the latter. The authors suggest that IL-7 and or IL-15 in the culture might preserve /enhance this phenotype. Tcm was particularly enriched in the CD8+ STEAP CAR compartment.

STEAP CAR monotherapy applied once induced delay of tumor growth within 16 days. Day 25 explanted tumors showed necrotic areas and CAR T infiltration in the treatment cohort. The authors applied CAR T intratumorally here: why? If the target is superior, specific, and expressed? Also, with regard to clinical application: in most disseminated disease no intra tumoral application is possible.

In vivo disseminated disease was then treated intravenously with CAR T and was associated with reduced and delayed marker expression of labelled cancer cells. Histology showed loss of target antigen day 97 versus 31. The authors then provide transcriptomics associated with

STEAP loss, which modulates most impressively cell cycle regulation genes, Krebs cycle/Glycolysis, downregulation of antigen processing and presentation, MHC expression. The latter was confirmed also by histologic analysis in CAR T treated tumors, which is a key finding! STEAP CAR therapy resulting in antigen loss and impaired antigen processing and presentation. This raises the question of a general mechanism of action, which is also paramount in haematological malignancies treated with CAR T.

Castration resistant PC with growth kinetics typical for human PC *in vivo* was established as disseminated disease. Single CAR T *iv* induced remission in all individuals of the treatment cohort and was confirmed macroscopically and with BLI. Survivors' spleens harboured respective CAR effector cells. Due to reactivity to even very low antigen density the authors in a hSTEAP1 transgene mouse examined potential off-target toxicities of *iv* administered CAR T (engineered with respective adaptations to the model system) in a model system. Treatment in NSG mice showed rapid disease regression followed by relapse 10 days later, survival was statistically significantly better (22 vs 12 d). Interestingly: Antigen loss was also observed in this model in pulmonary disseminated disease in the CAR T cohort. Whether this is a general side effect/phenomenon of CAR T therapy or the physiological response to an immune response irrespective of whether mediated by CAR T, NK or ab T cells would be interesting to investigate. Should be discussed.

The authors conclude that adjunct therapies are needed to overcome resistance in subgroups of advanced cancer patients where inter- or intra tumoral STEAP1 heterogeneity is present. This is a major contribution to the field highlighting that further therapy is necessary to intercept the undesirable formation of resistance by CAR. To my knowledge this is also the first report which sees CAR T as causal for the antigen loss and compares treatment cohort with ctrl. in this respect.

Safety and efficacy of therapy were also examined in a humanized mouse model with non-clonal RM9-STEAP1 cells which gave results like that seen in the NSG model. CAR T induced no cross toxicities or premature death of treated individuals while at the same time showing clear evidence for anti-tumor efficacy. Comparable infiltration rate of CD3+ and same degree of integrity of tissues (prostate and adrenal gland) in treatment and ctrl cohort. Again, and interestingly: Lung explants at the end of experiment show hSTEAP1 in some regions of lung metastases in untreated cohort, yet complete antigen loss in CAR T cohort, associated with significant downregulation of MHC class I. This finding let the authors seek for additional therapeutic intervention to counter regulate antigen loss. The authors introduce fusion cytokine CBD-IL12 which showed efficacy to revert tumor stroma *in vivo*, the authors show in RM9 tumors on scRNA level: restructure of the TME, enhanced proteasomal activity, enhanced MHC expression and antigen presentation, and via histology: increase in T cell infiltration. Next the authors tested un-transduced T, vs STEAP CAR T vs un-transduced T + CBD-IL12 vs STEAP CAR T + CBD-IL12 in non-clonal RM9 STEAP1 metastases in hSTEAP1-KI, preconditioned with CPA. Administration of therapy was 1/w for 3 weeks. Only CAR T combined with IL-12 increased survival and was associated with increase in plasma TH1 cytokines. Antigen loss was observed in CAR T only and CAR T+IL-12. Increased MHC class I and tumor infiltrating CD3+ were observed in CAR T and CAR T+IL12.

Response: We would like to thank the reviewer for critically assessing our work and appreciating the significance and impact of this research.

We agree that intratumoral administration of STEAP1 CAR T cells is not clinically applicable but was performed in the 22Rv1 subcutaneous xenograft model as a first pass to evaluate for evidence of *in vivo* activity. The results related to disseminated models are certainly more clinically relevant to advanced prostate cancer.

We appreciate the reviewer's point about the important finding of STEAP1 antigen loss associated with STEAP1 CAR T cell therapy in this study. While beyond the scope of this manuscript, we are actively investigating underlying mechanisms, including the determination of whether this may be specific to CAR T cell therapy or generalizable to other adoptive cellular therapies.

As suggested by the reviewer we have modified our discussion in lines 738-745 and 752-760, addressing relevant literature on tumor antigen loss, underscoring the need for additional studies on tumor-immune-stromal interactions that may contribute to antigen loss, and further characterization of the tumor immune microenvironment in response to combined CBD-IL-12 and STEAP1 CAR T cell therapy.

Comments to the authors:

Comment 1: *The authors may mention that antigen loss might be due to tumor editing which would be in line with that IL12 alone and IL-12+CAR T increased TCR diversity in tumor metastases of the lung.*

Response: We appreciate the reviewer's suggestion and believe that antigen loss may be a mechanism of tumor editing upon treatment with STEAP1 CAR T cell therapy. Antigen loss and the increase in TCR diversity could also be conceptualized through two independent mechanisms, the first an escape mechanism from stringent immunologic pressure of CAR T cell therapy and the latter an effect of CBD-IL-12. This is supported by our findings showing that CBD-IL-12 treatment alone did not lead to loss of STEAP1 expression (**Fig. 8a**) and the absence of an observable additive/synergistic effect on TCR diversity upon combining CBD-IL-12 with STEAP1 CAR T cell therapy (**Fig. 8h**).

Comment 2: *The work would clearly benefit if histological analyses were shown that indicate reversion of the stroma on protein level: Infiltration of T cells, M1, M2 macrophages, NK, iNKT cells, Treg cells.*

Response: We agree wholeheartedly with this comment as it would strengthen the rationale for the use of CBD-IL-12 to induce stromal reversion and its combination with STEAP1 CAR T cell therapy. We show in **Fig. 8a, b** that CD3⁺ T cells are increased based on IHC staining of tumors in the groups with CBD-IL-12 treatment relative to the control group of untransduced T cell treatment. We have now added multiparametric flow cytometry data (**Fig. 8c-g** and **Supplementary Fig. 19**) to further characterize the immune microenvironment of RM9-hSTEAP1 tumors treated with untransduced T cells, STEAP1 CAR T cells, untransduced T cells and CBD-IL-12, or CBD-IL-12 and STEAP1-mBBζ CAR T cells (both at tumor nadir and at relapse). Briefly, our findings reveal increased antitumorigenic F4/80⁺MHC-II⁺INOS2⁺ M1 macrophages and CD11b⁺XCR1⁺ antigen cross-presenting conventional type 1 dendritic cells associated with the combination of CBD-IL-12 and STEAP1 CAR T cell therapy. We also show reduced intratumoral neutrophils and conventional type 2 dendritic cells with combination treatment. CD4⁺FOXP3⁺ Tregs were reduced overall in conditions with CBD-IL-12 treatment. In contrast, NK cells were unchanged in frequency.

Comment 3: *scRNA analysis was not used to analyse the type of infiltrating T cells, e.g., based on the lineage TFs, cytokines, cytolytic markers, expression of activating NKps or corresponding ligands on Tumor cells, why? Transcriptomics can give a hint but do not correlate with protein expression.*

Response: In our revised manuscript, we present additional analysis of the scRNA-seq data to define different T cell subsets, M1 and M2 macrophages, dendritic cells, NK cells, eosinophils, and neutrophils. The UMAP plots and frequencies of these immune cell populations are plotted in **Fig. 6c, d**. Importantly, the findings from the scRNA-seq data are

largely corroborated by our multiparametric flow cytometry data presented in **Fig. 8c-g** and **Supplementary Fig. 19**.

Comment 4: 20 days post therapy was the endpoint of the combination therapy study, why? This is too short for immunotherapeutic interventions and to show CAR T plus tumor targeted IL12 can induce long-lasting remission/survival in solid malignancy. It would also have been interesting to analyse the one tumor bearing mouse: was the stroma reverted? less infiltration, less marker expressed that may make the Tumor susceptible to innate and adaptive immunity, less TH1 cytokines in periphery? did the Tumor show signs of senescence (terminal growth arrest due to cellular senescence induced by TH1 cytokines? which would positively impact survival. Thus, for comprehensiveness of the study more details on stroma reversion by immune cytokine CBD-IL12 are needed.

Response: We appreciate the reviewer's concern and suggestion on the role of stromal reversion and its impact on overall survival. The disseminated RM9 tumor model is highly aggressive and marked by rapid seeding and proliferation. Thus, the survival of mice after inoculation with RM9 tumor cells is only 20-30 days overall. In the present study, we have used only a single concentration of CBD-IL-12 therapy (25 mg/kg) based on data from our RM9 subcutaneous tumor model and prior work from the development of CBD-IL-12 (Mansurov A, Ishihara J, et al. *Nat Biomed Eng.* 2020.). Future optimization of the dose and administration schedule of CBD-IL-12 may be necessary to achieve more pronounced therapeutic effects.

As described in our response to Comment 3, we have added additional data to evaluate the stromal reversion of tumors associated with CBD-IL-12 treatment either alone or in combination with STEAP1 CAR T cells.

Will the work be of significance to the field and related fields? How does it compare to the established literature? If the work is not original, please provide relevant references.

Comment: *The work is of significance to the field in several aspects: (a) The authors use a new target for CAR T therapy, which is also on aggressive disseminated cancer cells, thus a "superior" target on lethal PC. (b) In contrast to the previous gold standard PSMA, STEAP1 is also found on treatment-resistant mCRPC. (c) The authors acknowledge that CAR T alone is insufficient to cure the disease, which most authors do not address with such clarity. They pinpoint CAR T as causative for antigen loss, which is a relevant important finding!*

Immunogenicity, delineated from proteasome activity, MHC class I and II expression, antigen processing and presentation, is negatively enriched in CAR T treated tumors, a finding that is of superior significance. They try to compensate antigen loss by adjuvant tumor targeted IL-12, since IL12 - as they show in RM9 tumors treated with IL-12FP -upregulates Ag presentation machinery and MHC class I. The authors may discuss the fact that also IL12 therapy led to antigen loss, thus did not fulfil the authors expectation to prevent antigen loss. However, the important question of whether the cause is the same in both cases has not been clarified! Is antigen loss observed because antigen positive tumour cells downregulate the antigen in the presence of CAR T or because they are killed? The former being more likely and consistent with reduced "antigen presenting machinery" results.

(STEAP1 is involved in proliferation which can be modulated by TH1 cytokines/metabolism/... in a reciprocal crosstalk between tumour and immune compartment). In the IL12 settings, tumor editing may have caused loss of antigen in STEAP1 heterogenous Tumor cells due to increased effector function of innate and adaptive immune cells and CAR T cells, thus quantitative killing of all STEAP1 positive Tumor cells may have occurred. In general: the authors may emphasise that this question has to be

answered and that the cross talk of tumour and effector cells and/or stroma needs to be investigated in more detail.

The production of different STEAP1 targeting CAR T constructs is gaplessly reported, and the CAR T characterized in detail regarding killing efficacy, required antigen density. The CAR T is produced in different donors and immunophenotyping reveals preferential TSCM type. Cross-reactivities were evaluated in elaborated in vivo models and were negative. The authors aim to develop an efficient immunotherapeutic approach combining CAR T and immune cytokine, which is a new and promising therapeutic option. They use state of the art and elaborated techniques for preclinical testing of the constructs to make the application safe. A step further towards clinical application of combined immune therapies.

Response: We thank the reviewer for underscoring the significance of our research findings and considering them relevant for the development of effective CAR T cell therapy for prostate cancer. We would like to clarify that we did not expect CBD-IL-12 to reverse STEAP1 antigen loss due to STEAP1 CAR T cell therapy. Instead, we hoped that CBD-IL-12 would overcome the downregulation of antigen processing and presentation due to STEAP1 antigen loss and engage components of endogenous innate and adaptive antitumor immunity. These expectations were fulfilled based on our analyses but were insufficient to achieve cure in our disseminated tumor model. We agree that the major takeaways from these studies are 1) STEAP1 CAR T cell therapy is potent in its ability to induce effective killing of STEAP1-positive cancer cells with relapse of STEAP1-negative disease and 2) evidence of tumor evolution to compensate for immune pressure by downregulating antigen processing and presentation machinery. Indeed, additional work is necessary and ongoing to deconvolute the mechanisms underlying these findings.

Does the work support the conclusions and claims, or is additional evidence needed?

Comment: *It is a well-founded, detailed, and comprehensive study in terms of the identification of the CAR target, the production of the CAR and the question of safety and efficacy of the CAR in monotherapies. The part where the CAR is combined with stromal reversion (IL-12 fusion cytokines) is not totally convincing since stroma reversion is downregulated/extinguished at the cellular level by CPA and tumor growth negatively impacted, thus true efficacy of IL-12 fusion cytokine is not clear. CPA is an unknown variable that has furthermore only been used in some but not all experimental groups. Without being commented or justified. Testing with full preservation of stromal reversion plus CAR combination therapy is missing, yet essential and is required to find out the potential of combined therapy.*

Response: We appreciate this comment regarding the use of CPA with CBD-IL-12 treatment. We apologize for this error in our presentation of the methods related to this experiment. CPA preconditioning would absolutely abrogate the potential for stromal reversion induced by CBD-IL-12. To clarify, mice treated with untransduced T cells + CBD-IL-12 or STEAP1-mBB ζ CAR T cells + CBD-IL-12 did not receive CPA for preconditioning. We have corrected this error in the methods section and updated the schematic for the study design in **Fig. 7a**.

As described in our response to Comment 3, we have added additional data to evaluate the stromal reversion of tumors associated with CBD-IL-12 treatment either alone or in combination with STEAP1 CAR T cells.

Are there any flaws in the data analysis, interpretation and conclusions? Do these prohibit publication or require revision?

Comment: *Testing combination therapy: Line 753: For animal testing combined IL12 and CAR therapy ... group d) is incorrect, should also include CBD-IL-12. Why did group a) (Untransduced T cells) and d) (CAR +IL12) receive preconditioning? That clearly creates*

different experimental conditions, doesn't it? Endogenous activated proliferating NK and T cells are very likely negatively modulated (eliminated?) by CPA treatment, the full potential of the treatment (CAR + IL12) seems not shown here. In d) the effect of stroma reversion is more or less (degree of impact is not analyzed) reduced to antigen-presentation machinery, which may lead to optimum tumor editing by CAR T (might explain loss of antigen?). The crosstalk of endogenous innate and adaptive immunity with stroma and tumor remains biased/incomplete/manipulated. CPA also negatively impacts tumor growth and proliferation (to which extend?) In this context: It should be discussed why antigen loss in "a priori" "heterogeneous STEAP1-expressing RM9 tumors" is quantitative in CAR treated tumor metastasis, but not so in ctrl. (See above).

48 hours prior to T cell infusion, groups c and d were pre-treated with 25 µg CBD-IL-12 by retroorbital sinus injection and then weekly thereafter. 24 hours prior to T cell infusion, groups a and d received preconditioning cyclophosphamide 100 mg/kg.

Duration/treatment phase of the combination treatment study is very short, this and the manipulation of tumor and immune compartment with CPA might have made it less likely to identify the full potential of the combination therapy. Therapy for a period of at least 35-42 days (5-6 weeks instead of 3) might have shown more of the potential of respective treatments. Combination therapy resulting in better but not good outcome is not sufficient with regard to clinical application. The authors clearly address this in the discussion. They may hypothesise why it is that no better results are achieved? Other reports combine IL12 immune cytokines with irradiation, and or additional cytokines (IL-2) to support T cell responses established by a reverted stroma. Also, CI might be mentioned as an additional therapeutic add on.

Response: We apologize for the ambiguity raised due to our error in defining the treatment groups receiving CPA preconditioning. As described in the previous response, we have corrected this in the methods section of the revised manuscript and included a detailed schematic of the study design to provide additional clarity.

Based on the reviewer's comments, we have added additional data to the revised manuscript detailing stromal reversion associated with CBD-IL-12 therapy including in combination with STEAP1 CAR T cell therapy. Importantly, we observe a loss of stromal reversion when tumors relapse/progress after combined STEAP1 CAR T cell and CBD-IL-12 therapy. Overall, we present a realistic perspective that combination therapy may improve outcomes but does not result in cures within our tumor model. As mentioned previously, additional optimization of dose or administration schedule of CBD-IL-12 may be necessary to achieve maximal therapeutic benefit. We also agree that the use of additional cytokines including IL-2, radiotherapy, or immune checkpoint inhibitors may be strategies to deepen antitumor responses as mentioned now in lines 760-765 of the discussion.

Is the methodology sound? Does the work meet the expected standards in your field?

Comment: Methodology is state of the art and comprehensive. Except for CPA preconditioning (see above).

Response: We have addressed this important comment and oversight related to the description of CPA preconditioning in treatment conditions involving CBD-IL-12 treatment in the previous responses.

Is there enough detail provided in the methods for the work to be reproduced?

Comment: Yes. Taken together: The questions raised in the comments should be answered. However, this work is of paramount interest to a broad readership including clinicians, translational researchers, immunologists.

Response: We thank the reviewer for this very positive assessment of our work.

Reviewer 4 Comments (with expertise in CAR-T):

The authors present a comprehensive data set and very compelling story of targeting STEAP1 in prostate cancer with CAR T cells. The manuscript is very well written, the methods are elaborate and even though prostate cancer is not “cured” (even in combination of STEAP1 CAR T cells with the CBD-IL12), this likely provides a realistic perspective also for clinical translation.

Major comment 1: STEAP1 expression: *The authors state that up to three cores from the same patient, but obtained from different tumor sites (primary vs. metastatic) were analysed. Can the authors add data on the homogeneity/heterogeneity of STEAP1 expression in these different tumor sites in each patient and comment on how this may impact efficacy?*

Response: We appreciate the reviewer's suggestion to add data on the homogeneity/heterogeneity of STEAP1 expression in different tumor sites in each patient. We would first clarify that the tissue microarray analyzed is composed of tissues from metastatic sites and does not include the primary site (prostate). In the revised manuscript, we provide new analysis (**Fig. 1e**) showing inter-patient and intra-patient heterogeneity in STEAP1 and PSMA expression using hypergeometric, Simpson, and Shannon diversity scores. These results show that about 70% of patients had STEAP1 expression (H-score >30) across all metastatic sites whereas approximately 30% of patients showed heterogeneous expression. On the contrary, 23% of patients showed no PSMA expression in their metastatic tissues, 32% demonstrated heterogeneous PSMA expression, and 45% showed PSMA expression across all metastatic sites. These results indicate broader expression of STEAP1 compared to PSMA in end-stage metastatic castration-resistant prostate cancer. STEAP1 heterogeneity is likely to be a major mechanism of resistance to STEAP1 CAR T cell therapy and thus we believe that combination therapies (i.e., delivery of CBD-IL-12) may be necessary to overcome this issue.

Major comment 2: STEAP1 vs. PSMA expression: *The authors show loss of STEAP1 expression in their in vivo models. At the time of relapse with (potential neuroendocrine trans-differentiation) and STEAP1 loss – is PSMA expression retained? Any data on dual antigen targeting of STEAP1 together with PSMA?*

Response: We appreciate these important questions raised by the reviewer. We have shown that 22Rv1 tumors with STEAP1 antigen loss after STEAP1 CAR T cell therapy retain the expression of PSMA (**Supplementary Fig. 7d**). In addition, we did not observe any changes in tumor cell morphology or expression of SYP and AR (**Supplementary Fig. 7d**) to indicate neuroendocrine transdifferentiation. Due to the findings of retained PSMA expression after STEAP1 antigen loss and the common co-expression of STEAP1 and PSMA in lethal metastatic castration-resistant prostate cancer (**Fig. 1b**), we are actively developing and optimizing dual STEAP1 and PSMA CAR T cell therapy strategies for prostate cancer. However, this work is premature for inclusion here and we intend to submit this work as a future manuscript.

Major comment 3: CAR design: *The authors use a modified IgG4 spacer that due to the elimination of Fc motifs may be immunogenic in humans. Also in mice, this spacer is very likely immunogenic. Please include any data on immunogenicity or immune responses that have been observed, at least in the immunocompetent mouse models.*

Response: Thank you for this excellent suggestion to assess the immunogenicity of the modified IgG4 spacer in our CAR design. We assayed for anti-human IgG and IgM antibodies produced at day 8 compared to day 0 in the hSTEAP1-KI immunocompetent mice model harbouring RM9-hSTEAP1 tumors treated with untransduced T cells and STEAP1-mBBζ CAR T cells. We observed no anti-human IgG or IgM antibodies (**Supplementary Fig.**

14c) above the minimum detection limit of the commercial Mouse Anti-Human Antibody ELISA kit.

Major comment 4: Toxicology testing: Can the authors present data on STEAP CAR T cell transfer into non-tumor bearing mice (that have the human STEAP1 KI)? Since all the mice in the current experiment die from progressive tumor, the claim of “no deaths” from targeting STEAP1 is hard to hold up.

Response: We conducted this experiment at the request of the reviewer to provide additional toxicology testing. Untransduced T cells and STEAP1-mBBζ CAR T cells were introduced into hSTEAP1-KI mice bearing RM9-hSTEAP1 tumors or no tumors. The results are provided in **Supplementary Fig. 14a, b** where we appreciated no deaths, loss of body weight, or evidence of gross toxicity in non-tumor bearing mice beyond one month after treatment.

Major comment 5: CBD-IL12: Any data on the effect of CBD-IL-12 alone in the immunocompetent mouse models? Since there is systemic administration and biodistribution – any toxicity? Any data on innate immune cell activation and migration to the tumor site(s)? Any data to demonstrate that CBD-IL 12-activated innate immune cells can eliminate STEAP1-positive and negative tumor cells?

Response: Our scientific collaborator and co-senior author Dr. Jun Ishihara at Imperial College London has previously published on the effect of CBD-IL-12 in enhancing tumor inflammation and driving complete responses in breast and melanoma mouse models (*Mansurov A, Ishihara J, et al. Nat Biomed Eng. 2020.*). In this publication, the biodistribution of CBD-12 (relative to IL-12) was characterized and showed increased localization of CBD-IL-12 to tumor stroma which was associated with diminished levels of systemic IFN-γ. Further, CBD-IL-12 was shown to activate innate and adaptive immunity with increased intratumoral enrichment of cytotoxic CD8⁺ T cells and MHC-II⁺CD80⁺ macrophages as well as a reduced number of Tregs in a pulmonary metastatic model of B16F10 melanoma.

We have shown that CBD-IL-12 demonstrates antitumor effects in two independent mouse prostate cancer models, RM9 and Myc-CaP (**Fig. 6b** and **Supplementary Fig. 16a**). Further, single-cell RNA-seq analysis of RM9 tumors treated with CBD-IL-12 demonstrated enrichment of innate and adaptive immune cells indicative of an inflamed tumor microenvironment (**Fig. 6c, d**) and consistent with the prior publication (*Mansurov A, Ishihara J, et al. Nat Biomed Eng. 2020.*). Our data also indicated that the addition of CBD-IL-12 to STEAP1-mBBζ CAR T cell therapy in the disseminated RM9-hSTEAP1 mouse model (where antigen loss is the evident mechanism of resistance to STEAP1-mBBζ CAR T cell therapy) leads to enhanced survival. We have also provided additional data as per the response to Reviewer 3, Comment 2 on the effects on the tumor immune microenvironment. Further, we have demonstrated that the repertoire of T cell receptors (TCRs) associated with intratumoral T cells is enhanced with CBD-IL-12 therapy (**Fig. 8h**). Taken together, these data indicate that CBD-IL-12 can broaden the antitumor immune responses through effects on both innate and adaptive immunity to combat both STEAP1-positive and -negative disease.

REVIEWERS' COMMENTS

Reviewer #2 (Remarks to the Author):

The manuscript has been revised according to my suggestions. The manuscript can now be published. There are only two new comments from my side on minor issues:

Minor Comment 1: Lines 93-104: References should be added.

Response: The relevant references have been added in the revised manuscript.

New comment: References should be added not only to the text in lines 97-102, but also to the text in lines 102-108.

Minor Comment 11: Suppl. Fig. 15: The bars showing INF- γ release in the “untransduced T cells + CDB-IL-12 group” (day 0 vs day 8) suggest a significant difference. If so, the p value should be added.

Response: Statistical analysis of IFN- γ release showed no significant difference between untransduced T cells + CBD-IL-12 treatment (day 0 vs day 8). This is likely due to the large standard deviation between the mouse samples.

New comment: Fig. 7e (formerly Suppl. Fig15): It cannot be understood that the green bars (INF γ release of untransduced T cells + CBD-IL-12, day 0 vs. day 8), should not be significantly different, as no large standard deviations can be seen. However, since the authors confirm that there is no significance, this is to be accepted.

Reviewer #3 (Remarks to the Author):

Dear authors,

Response to comment to reviewer3 regarding tumor antigen loss is not given as indicated by the authors in lines 738-745 and 752 – 760. Discussion section already ends line 744.

Moreover the response of the authors where they describe how they correct false description of preconditioning they mention that they now include in the manuscript that “ additional cytokines, radiotherapy, CI may be strategies to deepen anti-tumor responses” in lines 760-765. There is no line 760-765, in line 745 begins already Method section.

I have read all your comments and the newly submitted manuscript and have the following comments. These are in red, imbedded in the rebuttal letter from the authors.

Please find your rebuttal letter with my my comments in the attached docx file below.

Response to Reviewers' Comments

Reviewer 1 Comments (with expertise in prostate cancer, STEAP1):

It is a great article focusing the potential use of STEAP1 as immunotherapeutic target for prostate cancer. The authors started to evaluate and compare the expression of STEAP1 and PSMA proteins in lethal metastatic prostate cancer, demonstrating that targeting STEAP1 presents advantages in comparison with PSMA. Next, the authors used the potential therapy based in "chimeric antigen receptor (CAR) T cell to recognize the cell overexpressing the STEAP1 protein. With this construct it was observed that STEAP1 antigen loss was associated with diminished tumor antigen processing and presentation. Then, the authors have engineered the construct with IL-12 as an adjunct to STEAP1 CAR T cell therapy, showing promising results in in vitro and in vivo preclinical models. The experimental design is very well established, and the conclusions are in accordance with obtained results. Several experimental controls were used to validate the results. I suggest some improvements before acceptance for publication:

Comment 1. In supplemental Figure 1D and 1E, it must be added immunohistochemical image staining the AR and synaptophysin. Ideally, it should be evaluated the co-localization of STEAP1 and AR/synaptophysin by confocal microscopy.

Response: We thank the reviewer for the very positive response. We have addressed this comment by now including **Fig. 1e** which summarizes the expression of STEAP1 and PSMA in metastatic tumor cores coded by molecular subtype based on androgen receptor (AR) and synaptophysin (SYP) expression (AR+/SYP+, AR+/SYP-, AR-/SYP+, AR-/SYP-) and **Supplementary Fig. 2a** which shows representative tumor sections stained by IHC for AR, SYP, and STEAP1 expression.

Comment 2: The text should be reviewed to eliminate some grammatical mistakes.

Response: We have reviewed the manuscript text and edited for grammar.

Reviewer 2 Comments (with expertise in prostate cancer, immunotherapy, CAR-T):

The authors describe a new CAR T cell approach for targeting STEAP1, an antigen which they showed to be highly expressed in tissues of patients with metastatic prostate cancer. The CAR T cell therapy was tested in different mouse models. Antigen escape was identified as mechanism of therapeutic resistance. Combination of CAR T cell therapy with IL-12 immunotherapy was used for the remodeling of the tumor microenvironment and enhanced therapeutic outcome by engagement of host immunity and epitope spreading. The manuscript is well written and the findings from the experiments are easy to follow. At a few points, more information is necessary from a methodological point of view. Moreover, more data should be presented from the three CAR T cell versions that were generated, discussion about crossreactivity of the CAR T cells with murine STEAP1 and hSTEAP1B isoforms needs more clarity and Figure 7 needs some revisions.

Major points of revision:

Comment 1: Anti-STEAP1 CARs with different spacers are presented in Fig. 2A. The authors write that only the candidate with the long spacer showed activity. Comparative data from all three candidates should be added in a new suppl. figure.

Response: We thank the reviewer for the careful review of this manuscript and constructive comments. We have added **Supplementary Fig. 3c** to show data from the co-culture of short and medium spacer STEAP1 CAR T cells with parental 22Rv1, 22Rv1 STEAP1 ko, and 22Rv1 STEAP1 ko + rescue cells.

Comment 2: Lines 271-292: In the comparison between human and mouse STEAP1 the authors found that the ECD2 domain with high homology between the two species was associated with anti STEAP1 CAR T cell activation. Moreover, they identified Q198 and/or I209 of human STEAP1 as critical to productive recognition by the CAR T cells (suppl. Fig. 3F). On the other side, no CAR T cell activation was found on PC cells expressing the STEAP1B isoforms that share the ECD2 domain with STEAP1 including Q198 and I209 (suppl. Fig. 4A). This should be clarified.

Response: We appreciate the opportunity to clarify these points. STEAP1 CAR T cells are not activated when co-cultured with DU145 cells expressing mouse Steap1 (**Supplementary Fig. 4d, e**). When we take the mouse Steap1, replace its ECD2 with the ECD2 of human STEAP1, and express this chimeric protein (mSteap1 hECD2) in DU145 cells, we find that STEAP1 CAR T cells are activated in co-cultures with DU145-mSteap1 hECD2 cells (**Supplementary Fig. 4g**). While there is homology between the ECD2 of human STEAP1 and the STEAP1B isoforms, the crystal structure of STEAP1B has not been resolved and it is unknown whether this region of homology is extracellular in STEAP1B isoforms. We show data from an *in silico* membrane topology prediction tool TOPCON showing low reliability in the prediction that this region of homology in STEAP1B isoforms is extracellular. To empirically determine whether STEAP1 CAR T cells may cross-react with STEAP1B, we overexpressed each of the STEAP1B isoforms in DU145 cells and performed co-cultures with STEAP1 CAR T cells. Our data indicated that STEAP1 CAR T cells are activated in the presence of human STEAP1 but not mouse Steap1 or human STEAP1B isoforms (**Supplementary Fig. 5d**). We have clarified these findings in the Results section of the revised manuscript.

Comment 3: In Fig. 7B, only BLI pictures from days -1, 9, and 17 are shown. Fig. 7C shows that more BLI pictures were made on other days of the therapy. These picture should be added to Fig. 7B.

Response: As per the reviewer's suggestion we have added the complete set of BLI images for the mice experiment in **Fig. 7b** in the revised manuscript.

Comment 4: Fig. 7C and 7D do not seem to correspond. For example, In Fig. 7C one animal survived until day 17 in the "untransduced cells + CBD-IL-12" group. In Fig. 7D, however, it seems that two animals of this group survived until day 17. This should be corrected.

Response: We apologize for this error while plotting the graphs. We have corrected this in the revised manuscript.

Comment 5: In Fig. 7F, there is no significant difference in the number of CD3+ cells in tumors treated with CAR T cells alone compared to CAR T cells + CDB-IL-12. This means that there is no enhanced CAR T cell infiltration into tumors treated with CDB-IL-12. This should be discussed in the context of the experiment.

Response: We agree with the reviewer that a significant difference in intratumoral CD3⁺ cells in tumors treated with CAR T cells alone compared to CAR T cells + CBD-IL-12 was not observed. However, we now show in **Fig. 8c-g** and **Supplementary Fig. 19** that CBD-IL-12 clearly modulates other aspects of the tumor immune microenvironment including increases in M1 macrophages and antigen cross-presenting dendritic cells as well as decreases in immunosuppressive neutrophils and T regulatory cells (Tregs). These results are discussed in the context of the experiment in the revised manuscript.

Minor points of revision:

Minor Comment 1: Lines 93-104: References should be added.

Response: The relevant references have been added in the revised manuscript.

Minor Comment 2: line 139: the NCT number should be added.

Response: The relevant clinical trial number NCT04221542 has been added in the revised manuscript.

Minor Comment 3: line 214: what is meant with “1 nM affinity” (Kd value?)

Response: The 1 nM affinity was reported previously (*Challita-Eid PM et al. Cancer Res. 2007.*). Per the publication, cells were incubated with increasing concentrations of STEAP1 antibody, washed, incubated with phycoerythrin-conjugated secondary antibody, and analyzed by flow cytometry. The affinity was calculated based on nonlinear regression of antibody concentration plotted against mean fluorescence intensity.

Minor Comment 4: lines 238-240: STEAP1 expression of DU145 STEAP1 cells should be shown in an additional suppl. figure.

Response: Immunoblot analysis showing the expression of human STEAP1 and mouse STEAP1 in engineered DU145 cells is now presented in **Supplementary Fig. 4a**.

Minor Comment 5: Lines 324-326: The authors state a statistically significant inhibition of tumor growth in the CAR T cell group by day 16, however it seems that the significant effect occurred only from day 18 on (Fig. 3A). This should be corrected.

Response: We have corrected this in the text of the revised manuscript.

Minor Comment 6: For a better understanding PMA/I should be described in the legends of suppl. Fig. 2B and 3C.

Response: We have added a description of the PMA/I abbreviation in the supplementary figure legends.

Minor Comment 7: Termination criteria for killing of the animals with disseminated tumors in the experiments investigating OS should be added in the methods section.

Response: The termination criteria for euthanizing the mice in the disseminated mice model studies have been elaborated in the revised material and method section of the manuscript.

Minor Comment 8: Suppl. Fig. 11E: there seems to be a wrong numbering of the animals of the CAR T cells group (no. 6-10 instead of 5-9).

Response: We have corrected the numbering in the revised **Supplementary Fig. 12e, f**.

Minor Comment 9: In Fig. 11G, data of only two animals in the group treated with untransduced T cells are shown, although five animals were used in the experiment.

Response: Unfortunately, we were unable to collect the spleens from the remaining three mice in the untransduced group as they were found dead prior to compassionate euthanasia endpoints. Thus, the graph consists of data from remaining two mice treated with untransduced T cells.

Minor Comment 10: Suppl. Fig. 9: It is not clear what is in the dishes (organs from one or more mice?). The control image is over-illuminated so that no organs or tumor lesions are visible. The images should therefore be improved.

Response: The wells each contain liver or lungs from a single mouse. We have modified the figure to include photographic images with and without the bioluminescence overlay to **Supplementary Fig. 10b** to enable better visualization of organs and tumors in the control group.

Minor Comment 11: Suppl. Fig. 15: The bars showing INF- γ release in the “untransduced T cells + CDB-IL-12 group” (day 0 vs day 8) suggest a significant difference. If so, the p value should be added.

Response: Statistical analysis of IFN- γ release showed no significant difference between untransduced T cells + CBD-IL-12 treatment (day 0 vs day 8). This is likely due to the large standard deviation between the mouse samples.

Reviewer 3 Comments (with expertise in IL12 immunotherapy, CAR-T):

What are the noteworthy results?

Histology based the authors acknowledge/confirm STEAP1 as a superior target for PC, present on especially those cancer cells that are identified in relapse/disseminated disease, metastases, late-stage disease. STEAP is involved in tumor progression, antibodies inhibit local and metastatic Ewing sarcoma growth. Would be interesting to know whether tumor stem cells also express it. The authors then developed STEAP1-directed chimeric antigen receptor (CAR) T cells with fully humanized scFv from vedotin which possesses high binding affinity. To tune CAR activity different hinge regions were implemented and different constructs tested. Only the long spacer STEAP1 CAR met expectations regarding IFN release and killing activity. Lead STEAP1 CAR T cells then demonstrated reactivity towards even low antigen density and killed diverse metastatic human and mouse prostate cancers. Application seemed safe in a human STEAP1 knock-in mouse model. STEAP1 CAR were not cross reactive towards xenogeneic mouse STEAP1 or human STEAP2. Bhatia et al. identified ECD2 as mandatory for STEAP1 recognition on the target, and this domain did not recognize 3 STEAP1B variants despite high sequence homology to STEAP1.

Next, the authors characterize STEAP1 CAR T cell products generated with T cells from different donors. They report a 20-40fold expansion of STEAP CAR T in 11 days, with a slightly higher proliferative capacity in CD4+ CAR T than CD8+ CAR T. PD-1 and LAG-3 expression was the same in un-transduced versus STEAP1 CAR T cells, which the authors assume is due to absent tonic (constitutive) signalling, which is positive for effector function (less exhaustion). Immunophenotyping of untransduced versus STEAP1 transduced CAR T subset revealed a higher proportion of Tscm in the latter. The authors suggest that IL-7 and or IL-15 in the culture might preserve /enhance this phenotype. Tcm was particularly enriched in the CD8+ STEAP CAR compartment.

STEAP CAR monotherapy applied once induced delay of tumor growth within 16 days. Day 25 explanted tumors showed necrotic areas and CAR T infiltration in the treatment cohort. The authors applied CAR T intratumorally here: why? If the target is superior, specific, and expressed? Also, with regard to clinical application: in most disseminated disease no intra tumoral application is possible.

In vivo disseminated disease was then treated intravenously with CAR T and was associated with reduced and delayed marker expression of labelled cancer cells. Histology showed loss of target antigen day 97 versus 31. The authors then provide transcriptomics associated with

STEAP loss, which modulates most impressively cell cycle regulation genes, Krebs cycle/Glycolysis, downregulation of antigen processing and presentation, MHC expression. The latter was confirmed also by histologic analysis in CAR T treated tumors, which is a key finding! STEAP CAR therapy resulting in antigen loss and impaired antigen processing and presentation. This raises the question of a general mechanism of action, which is also paramount in haematological malignancies treated with CAR T.

Castration resistant PC with growth kinetics typical for human PC in vivo was established as disseminated disease. Single CAR T iv induced remission in all individuals of the treatment cohort and was confirmed macroscopically and with BLI. Survivors' spleens harboured respective CAR effector cells. Due to reactivity to even very low antigen density the authors in a hSTEAP1 transgene mouse examined potential off-target toxicities of iv administered CAR T (engineered with respective adaptations to the model system) in a model system. Treatment in NSG mice showed rapid disease regression followed by relapse 10 days later, survival was statistically significantly better (22 vs 12 d). Interestingly: Antigen loss was also observed in this model in pulmonary disseminated disease in the CAR T cohort. Whether this is a general side effect/phenomenon of CAR T therapy or the physiological response to an immune response irrespective of whether mediated by CAR T, NK or ab T cells would be interesting to investigate. Should be discussed.

The authors conclude that adjunct therapies are needed to overcome resistance in subgroups of advanced cancer patients where inter- or intra tumoral STEAP1 heterogeneity is present. This is a major contribution to the field highlighting that further therapy is necessary to intercept the undesirable formation of resistance by CAR. To my knowledge this is also the first report which sees CAR T as causal for the antigen loss and compares treatment cohort with ctrl. in this respect.

Safety and efficacy of therapy were also examined in a humanized mouse model with non-clonal RM9-STEAP1 cells which gave results like that seen in the NSG model. CAR T induced no cross toxicities or premature death of treated individuals while at the same time showing clear evidence for anti-tumor efficacy. Comparable infiltration rate of CD3+ and same degree of integrity of tissues (prostate and adrenal gland) in treatment and ctrl cohort. Again, and interestingly: Lung explants at the end of experiment show hSTEAP1 in some regions of lung metastases in untreated cohort, yet complete antigen loss in CAR T cohort, associated with significant downregulation of MHC class I. This finding let the authors seek for additional therapeutic intervention to counter regulate antigen loss. The authors introduce fusion cytokine CBD-IL12 which showed efficacy to revert tumor stroma in vivo, the authors show in RM9 tumors on scRNA level: restructure of the TME, enhanced proteasomal activity, enhanced MHC expression and antigen presentation, and via histology: increase in T cell infiltration. Next the authors tested un-transduced T, vs STEAP CAR T vs un-transduced T + CBD-IL12 vs STEAP CAR T + CBD-IL12 in non-clonal RM9 STEAP1 metastases in hSTEAP1-KI, preconditioned with CPA. Administration of therapy was 1/w for 3 weeks. Only CAR T combined with IL-12 increased survival and was associated with increase in plasma TH1 cytokines. Antigen loss was observed in CAR T only and CAR T+IL-12. Increased MHC class I and tumor infiltrating CD3+ were observed in CAR T and CAR T+IL12.

Response: We would like to thank the reviewer for critically assessing our work and appreciating the significance and impact of this research.

We agree that intratumoral administration of STEAP1 CAR T cells is not clinically applicable but was performed in the 22Rv1 subcutaneous xenograft model as a first pass to evaluate for evidence of *in vivo* activity. The results related to disseminated models are certainly more clinically relevant to advanced prostate cancer. **Trafficking to the spot of interest and staying there i.e. in the tumor nest is of significance, no matter whether the disease is disseminated**

or not. It is dependent on the amount of target expressed by the tumor, the exclusive specificity of the target to the tumor, and on lack of cross reactivity of the CAR to other molecules. Labelled CAR T can show whether they are able to enrich in a tumor lesion and/or stay there.

We appreciate the reviewer's point about the important finding of STEAP1 antigen loss associated with STEAP1 CAR T cell therapy in this study. While beyond the scope of this manuscript, we are actively investigating underlying mechanisms, including the determination of whether this may be specific to CAR T cell therapy or generalizable to other adoptive cellular therapies.

As suggested by the reviewer we have modified our discussion in lines 738-745 and 752-760, these lines do not exist in the discussion section. However I found the respective changes in the manuscript. Good completion. addressing relevant literature on tumor antigen loss, underscoring the need for additional studies on tumor-immune-stromal interactions that may contribute to antigen loss, and further characterization of the tumor immune microenvironment in response to combined CBD-IL-12 and STEAP1 CAR T cell therapy.

Comments to the authors:

Comment 1: The authors may mention that antigen loss might be due to tumor editing which would be in line with that IL12 alone and IL-12+CAR T increased TCR diversity in tumor metastases of the lung.

Response: We appreciate the reviewer's suggestion and believe that antigen loss may be a mechanism of tumor editing upon treatment with STEAP1 CAR T cell therapy. Antigen loss and the increase in TCR diversity could also be conceptualized through two independent mechanisms, the first an escape mechanism from stringent immunologic pressure of CAR T cell therapy and the latter an effect of CBD-IL-12. This is supported by our findings showing that CBD-IL-12 treatment alone did not lead to loss of STEAP1 expression (Fig. 8a) and the absence of an observable additive/synergistic effect on TCR diversity upon combining CBD-IL-12 with STEAP1 CAR T cell therapy (Fig. 8h).

Comment 2: The work would clearly benefit if histological analyses were shown that indicate reversion of the stroma on protein level: Infiltration of T cells, M1, M2 macrophages, NK, iNKT cells, Treg cells.

Response: We agree wholeheartedly with this comment as it would strengthen the rationale for the use of CBD-IL-12 to induce stromal reversion and its combination with STEAP1 CAR T cell therapy. We show in Fig. 8a, b that CD3⁺ T cells are increased based on IHC staining of tumors in the groups with CBD-IL-12 treatment relative to the control group of untransduced T cell treatment. We have now added multiparametric flow cytometry data (Fig. 8c-g and Supplementary Fig. 19) to further characterize the immune microenvironment of RM9-hSTEAP1 tumors treated with untransduced T cells, STEAP1 CAR T cells, untransduced T cells and CBD-IL-12, or CBD-IL-12 and STEAP1-mBBζ CAR T cells (both at tumor nadir and at relapse). Briefly, our findings reveal increased antitumorigenic F4/80⁺MHC-II⁺INOS2⁺ M1 macrophages and CD11b⁺XCR1⁺ antigen cross-presenting conventional type 1 dendritic cells associated with the combination of CBD-IL-12 and STEAP1 CAR T cell therapy. We also show reduced intratumoral neutrophils and conventional type 2 dendritic cells with combination treatment. CD4⁺FOXP3⁺ Tregs were reduced overall in conditions with CBD-IL-12 treatment. In contrast, NK cells were unchanged in frequency.

The already originally submitted staining of CD3 is not of the highest quality. Low magnification, low contrast, positive cells not clearly distinguishable, can only be approximated.

The now subsequently submitted multiparametric phenotype analysis is based on flow and is thus statistical, why do the authors not provide a multiparametric (classical) histological analysis of the cells in the TME? FC analysis is very questionable, especially DCs and macrophages are very difficult to "isolate". This method is never quantitative, not reproducible and different for each tumour as it depends on the particular connective tissue portion of the tumour. Also, the phenotype changes if any form of digestion is used. Ultra-high-content imaging Macsima or PhenoCycler (formerly Codex) allows deep spatial phenotyping with target cells in the relevant context and in the respective cell-typical quantity ratios. This and conventional histology are much more informative than phenotyping cells that have been manipulated and taken out of their context. Stroma reversion is not convincingly demonstrated.

Comment 3: scRNA analysis was not used to analyse the type of infiltrating T cells, e.g., based on the lineage TFs, cytokines, cytolytic markers, expression of activating NKps or corresponding ligands on Tumor cells, why? Transcriptomics can give a hint but do not correlate with protein expression.

Response: In our revised manuscript, we present additional analysis of the scRNA-seq data to define different T cell subsets, M1 and M2 macrophages, dendritic cells, NK cells, eosinophils, and neutrophils. The UMAP plots and frequencies of these immune cell populations are plotted in **Fig. 6c, d**. Importantly, the findings from the scRNA-seq data are largely corroborated by our multiparametric flow cytometry data presented in **Fig. 8c-g** and **Supplementary Fig. 19**. **Very nice addition to the study**

Comment 4: 20 days post therapy was the endpoint of the combination therapy study, why? This is too short for immunotherapeutic interventions and to show CAR T plus tumor targeted IL12 can induce long-lasting remission/survival in solid malignancy. It would also have been interesting to analyse the one tumor bearing mouse: was the stroma reverted? less infiltration, less marker expressed that may make the Tumor susceptible to innate and adaptive immunity, less TH1 cytokines in periphery? did the Tumor show signs of senescence (terminal growth arrest due to cellular senescence induced by TH1 cytokines? which would positively impact survival. Thus, for comprehensiveness of the study more details on stroma reversion by immune cytokine CBD-IL12 are needed.

Response: We appreciate the reviewer's concern and suggestion on the role of stromal reversion and its impact on overall survival. The disseminated RM9 tumor model is highly aggressive and marked by rapid seeding and proliferation. Thus, the survival of mice after inoculation with RM9 tumor cells is only 20-30 days overall. In the present study, we have used only a single concentration of CBD-IL-12 therapy (25 mg/kg) based on data from our RM9 subcutaneous tumor model and prior work from the development of CBD-IL-12 (Mansurov A, Ishihara J, et al. *Nat Biomed Eng.* 2020.). Future optimization of the dose and administration schedule of CBD-IL-12 may be necessary to achieve more pronounced therapeutic effects. **25µg/g body weight is high dose therapy. Perhaps it makes sense to take a tumour cell line next time that does not close the time window before successful stroma reversion.**

As described in our response to Comment 3, we have added additional data to evaluate the stromal reversion of tumors associated with CBD-IL-12 treatment either alone or in combination with STEAP1 CAR T cells.

Will the work be of significance to the field and related fields? How does it compare to the established literature? If the work is not original, please provide relevant references.

Comment: The work is of significance to the field in several aspects: (a) The authors use a new target for CAR T therapy, which is also on aggressive disseminated cancer cells, thus a

“superior” target on lethal PC. (b) In contrast to the previous gold standard PSMA, STEAP1 is also found on treatment-resistant mCRPC. (c) The authors acknowledge that CAR T alone is insufficient to cure the disease, which most authors do not address with such clarity. They pinpoint CAR T as causative for antigen loss, which is a relevant important finding!

Immunogenicity, delineated from proteasome activity, MHC class I and II expression, antigen processing and presentation, is negatively enriched in CAR T treated tumors, a finding that is of superior significance. They try to compensate antigen loss by adjuvant tumor targeted IL-12, since IL12 - as they show in RM9 tumors treated with IL-12FP -upregulates Ag presentation machinery and MHC class I. The authors may discuss the fact that also IL12 therapy led to antigen loss, thus did not fulfil the authors expectation to prevent antigen loss. However, the important question of whether the cause is the same in both cases has not been clarified! Is antigen loss observed because antigen positive tumour cells downregulate the antigen in the presence of CAR T or because they are killed? The former being more likely and consistent with reduced "antigen presenting machinery" results.

(STEAP1 is involved in proliferation which can be modulated by TH1 cytokines/metabolism/... in a reciprocal crosstalk between tumour and immune compartment). In the IL12 settings, tumor editing may have caused loss of antigen in STEAP1 heterogenous Tumor cells due to increased effector function of innate and adaptive immune cells and CAR T cells, thus quantitative killing of all STEAP1 positive Tumor cells may have occurred. In general: the authors may emphasise that this question has to be answered and that the cross talk of tumour and effector cells and/or stroma needs to be investigated in more detail.

The production of different STEAP1 targeting CAR T constructs is gaplessly reported, and the CAR T characterized in detail regarding killing efficacy, required antigen density. The CAR T is produced in different donors and immunophenotyping reveals preferential TSCM type. Cross-reactivities were evaluated in elaborated in vivo models and were negative. The authors aim to develop an efficient immunotherapeutic approach combining CAR T and immune cytokine, which is a new and promising therapeutic option. They use state of the art and elaborated techniques for preclinical testing of the constructs to make the application safe. A step further towards clinical application of combined immune therapies.

Response: We thank the reviewer for underscoring the significance of our research findings and considering them relevant for the development of effective CAR T cell therapy for prostate cancer. We would like to clarify that we did not expect CBD-IL-12 to reverse STEAP1 antigen loss due to STEAP1 CAR T cell therapy. Instead, we hoped that CBD-IL-12 would overcome the downregulation of antigen processing and presentation due to STEAP1 antigen loss and engage components of endogenous innate and adaptive antitumor immunity. These expectations were fulfilled based on our analyses but were insufficient to achieve cure in our disseminated tumor model. We agree that the major takeaways from these studies are 1) STEAP1 CAR T cell therapy is potent in its ability to induce effective killing of STEAP1-positive cancer cells with relapse of STEAP1-negative disease and 2) evidence of tumor evolution to compensate for immune pressure by downregulating antigen processing and presentation machinery. Indeed, additional work is necessary and ongoing to deconvolute the mechanisms underlying these findings.

Does the work support the conclusions and claims, or is additional evidence needed?

Comment: It is a well-founded, detailed, and comprehensive study in terms of the identification of the CAR target, the production of the CAR and the question of safety and efficacy of the CAR in monotherapies. The part where the CAR is combined with stromal reversion (IL-12 fusion cytokines) is not totally convincing since stroma reversion is downregulated/extinguished at the cellular level by CPA and tumor growth negatively impacted, thus true efficacy of IL-12 fusion cytokine is not clear. CPA is an unknown variable

that has furthermore only been used in some but not all experimental groups. Without being commented or justified. Testing with full preservation of stromal reversion plus CAR combination therapy is missing, yet essential and is required to find out the potential of combined therapy.

Response: We appreciate this comment regarding the use of CPA with CBD-IL-12 treatment. We apologize for this error in our presentation of the methods related to this experiment. CPA preconditioning would absolutely abrogate the potential for stromal reversion induced by CBD-IL-12. To clarify, mice treated with untransduced T cells + CBD-IL-12 or STEAP1-mBBζ CAR T cells + CBD-IL-12 did not receive CPA for preconditioning. We have corrected this error in the methods section and updated the schematic for the study design in **Fig. 7a**.

As described in our response to Comment 3, we have added additional data to evaluate the stromal reversion of tumors associated with CBD-IL-12 treatment either alone or in combination with STEAP1 CAR T cells.

Are there any flaws in the data analysis, interpretation and conclusions? Do these prohibit publication or require revision?

Comment: Testing combination therapy: Line 753: For animal testing combined IL12 and CAR therapy ... group d) is incorrect, should also include CBD-IL-12. Why did group a) (Untransduced T cells) and d) (CAR +IL12) receive preconditioning? That clearly creates different experimental conditions, doesn't it? Endogenous activated proliferating NK and T cells are very likely negatively modulated (eliminated?) by CPA treatment, the full potential of the treatment (CAR + IL12) seems not shown here. In d) the effect of stroma reversion is more or less (degree of impact is not analyzed) reduced to antigen-presentation machinery, which may lead to optimum tumor editing by CAR T (might explain loss of antigen?). The crosstalk of endogenous innate and adaptive immunity with stroma and tumor remains biased/incomplete/manipulated. CPA also negatively impacts tumor growth and proliferation (to which extend?) In this context: It should be discussed why antigen loss in "a priori" "heterogeneous STEAP1-expressing RM9 tumors" is quantitative in CAR treated tumor metastasis, but not so in ctrl. (See above).

48 hours prior to T cell infusion, groups c and d were pre-treated with 25 µg CBD-IL-12 by retroorbital sinus injection and then weekly thereafter. 24 hours prior to T cell infusion, groups a and d received preconditioning cyclophosphamide 100 mg/kg.

Duration/treatment phase of the combination treatment study is very short, this and the manipulation of tumor and immune compartment with CPA might have made it less likely to identify the full potential of the combination therapy. Therapy for a period of at least 35-42 days (5-6 weeks instead of 3) might have shown more of the potential of respective treatments. Combination therapy resulting in better but not good outcome is not sufficient with regard to clinical application. The authors clearly address this in the discussion. They may hypothesise why it is that no better results are achieved? Other reports combine IL12 immune cytokines with irradiation, and or additional cytokines (IL-2) to support T cell responses established by a reverted stroma. Also, CI might be mentioned as an additional therapeutic add on.

Response: We apologize for the ambiguity raised due to our error in defining the treatment groups receiving CPA preconditioning. As described in the previous response, we have corrected this in the methods section of the revised manuscript and included a detailed schematic of the study design to provide additional clarity. **Accepted.**

Based on the reviewer's comments, we have added additional data to the revised manuscript detailing stromal reversion associated with CBD-IL-12 therapy including in combination with STEAP1 CAR T cell therapy. Importantly, we observe a loss of stromal reversion when tumors relapse/progress after combined STEAP1 CAR T cell and CBD-IL-12

therapy. Overall, we present a realistic perspective that combination therapy may improve outcomes but does not result in cures within our tumor model. As mentioned previously, additional optimization of dose or administration schedule of CBD-IL-12 may be necessary to achieve maximal therapeutic benefit. **or to use as mentioned before, other tumor lines** We also agree that the use of additional cytokines including IL-2, radiotherapy, or immune checkpoint inhibitors may be strategies to deepen antitumor responses as mentioned now in **lines 760-765 do not exist in the discussion section.** of the discussion.

Is the methodology sound? Does the work meet the expected standards in your field?

Comment: Methodology is state of the art and comprehensive. Except for CPA preconditioning (see above).

Response: We have addressed this important comment and oversight related to the description of CPA preconditioning in treatment conditions involving CBD-IL-12 treatment in the previous responses.

Is there enough detail provided in the methods for the work to be reproduced?

Comment: Yes. Taken together: The questions raised in the comments should be answered. However, this work is of paramount interest to a broad readership including clinicians, translational researchers, immunologists.

Response: We thank the reviewer for this very positive assessment of our work.

Reviewer 4 Comments (with expertise in CAR-T):

The authors present a comprehensive data set and very compelling story of targeting STEAP1 in prostate cancer with CAR T cells. The manuscript is very well written, the methods are elaborate and even though prostate cancer is not “cured” (even in combination of STEAP1 CAR T cells with the CBD-IL12), this likely provides a realistic perspective also for clinical translation.

Major comment 1: STEAP1 expression: The authors state that up to three cores from the same patient, but obtained from different tumor sites (primary vs. metastatic) were analysed. Can the authors add data on the homogeneity/heterogeneity of STEAP1 expression in these different tumor sites in each patient and comment on how this may impact efficacy?

Response: We appreciate the reviewer’s suggestion to add data on the homogeneity/heterogeneity of STEAP1 expression in different tumor sites in each patient. We would first clarify that the tissue microarray analyzed is composed of tissues from metastatic sites and does not include the primary site (prostate). In the revised manuscript, we provide new analysis (**Fig. 1e**) showing inter-patient and intra-patient heterogeneity in STEAP1 and PSMA expression using hypergeometric, Simpson, and Shannon diversity scores. These results show that about 70% of patients had STEAP1 expression (H-score >30) across all metastatic sites whereas approximately 30% of patients showed heterogeneous expression. On the contrary, 23% of patients showed no PSMA expression in their metastatic tissues, 32% demonstrated heterogeneous PSMA expression, and 45% showed PSMA expression across all metastatic sites. These results indicate broader expression of STEAP1 compared to PSMA in end-stage metastatic castration-resistant prostate cancer. STEAP1 heterogeneity is likely to be a major mechanism of resistance to STEAP1 CAR T cell therapy and thus we believe that combination therapies (i.e., delivery of CBD-IL-12) may be necessary to overcome this issue.

Major comment 2: STEAP1 vs. PSMA expression: The authors show loss of STEAP1 expression in their in vivo models. At the time of relapse with (potential neuroendocrine

trans-differentiation) and STEAP1 loss – is PSMA expression retained? Any data on dual antigen targeting of STEAP1 together with PSMA?

Response: We appreciate these important questions raised by the reviewer. We have shown that 22Rv1 tumors with STEAP1 antigen loss after STEAP1 CAR T cell therapy retain the expression of PSMA (**Supplementary Fig. 7d**). In addition, we did not observe any changes in tumor cell morphology or expression of SYP and AR (**Supplementary Fig. 7d**) to indicate neuroendocrine transdifferentiation. Due to the findings of retained PSMA expression after STEAP1 antigen loss and the common co-expression of STEAP1 and PSMA in lethal metastatic castration-resistant prostate cancer (**Fig. 1b**), we are actively developing and optimizing dual STEAP1 and PSMA CAR T cell therapy strategies for prostate cancer. However, this work is premature for inclusion here and we intend to submit this work as a future manuscript.

Major comment 3: CAR design: The authors use a modified IgG4 spacer that due to the elimination of Fc motifs may be immunogenic in humans. Also in mice, this spacer is very likely immunogenic. Please include any data on immunogenicity or immune responses that have been observed, at least in the immunocompetent mouse models.

Response: Thank you for this excellent suggestion to assess the immunogenicity of the modified IgG4 spacer in our CAR design. We assayed for anti-human IgG and IgM antibodies produced at day 8 compared to day 0 in the hSTEAP1-KI immunocompetent mice model harbouring RM9-hSTEAP1 tumors treated with untransduced T cells and STEAP1-mBB ζ CAR T cells. We observed no anti-human IgG or IgM antibodies (**Supplementary Fig. 14c**) above the minimum detection limit of the commercial Mouse Anti-Human Antibody ELISA kit.

Major comment 4: Toxicology testing: Can the authors present data on STEAP CAR T cell transfer into non-tumor bearing mice (that have the human STEAP1 KI)? Since all the mice in the current experiment die from progressive tumor, the claim of “no deaths” from targeting STEAP1 is hard to hold up.

Response: We conducted this experiment at the request of the reviewer to provide additional toxicology testing. Untransduced T cells and STEAP1-mBB ζ CAR T cells were introduced into hSTEAP1-KI mice bearing RM9-hSTEAP1 tumors or no tumors. The results are provided in **Supplementary Fig. 14a, b** where we appreciated no deaths, loss of body weight, or evidence of gross toxicity in non-tumor bearing mice beyond one month after treatment.

Major comment 5: CBD-IL12: Any data on the effect of CBD-IL-12 alone in the immunocompetent mouse models? Since there is systemic administration and biodistribution – any toxicity? Any data on innate immune cell activation and migration to the tumor site(s)? Any data to demonstrate that CBD-IL 12-activated innate immune cells can eliminate STEAP1-positive and negative tumor cells?

Response: Our scientific collaborator and co-senior author Dr. Jun Ishihara at Imperial College London has previously published on the effect of CBD-IL-12 in enhancing tumor inflammation and driving complete responses in breast and melanoma mouse models (*Mansurov A, Ishihara J, et al. Nat Biomed Eng. 2020.*). In this publication, the biodistribution of CBD-12 (relative to IL-12) was characterized and showed increased localization of CBD-IL-12 to tumor stroma which was associated with diminished levels of systemic IFN- γ . Further, CBD-IL-12 was shown to activate innate and adaptive immunity with increased intratumoral enrichment of cytotoxic CD8⁺ T cells and MHC-II⁺CD80⁺ macrophages as well as a reduced number of Tregs in a pulmonary metastatic model of B16F10 melanoma.

We have shown that CBD-IL-12 demonstrates antitumor effects in two independent mouse prostate cancer models, RM9 and Myc-CaP (**Fig. 6b** and **Supplementary Fig. 16a**). Further, single-cell RNA-seq analysis of RM9 tumors treated with CBD-IL-12 demonstrated enrichment of innate and adaptive immune cells indicative of an inflamed tumor microenvironment (**Fig. 6c, d**) and consistent with the prior publication (*Mansurov A, Ishihara J, et al. Nat Biomed Eng. 2020.*). Our data also indicated that the addition of CBD-IL-12 to STEAP1-mBB ζ CAR T cell therapy in the disseminated RM9-hSTEAP1 mouse model (where antigen loss is the evident mechanism of resistance to STEAP1-mBB ζ CAR T cell therapy) leads to enhanced survival. We have also provided additional data as per the response to Reviewer 3, Comment 2 on the effects on the tumor immune microenvironment. Further, we have demonstrated that the repertoire of T cell receptors (TCRs) associated with intratumoral T cells is enhanced with CBD-IL-12 therapy (**Fig. 8h**). Taken together, these data indicate that CBD-IL-12 can broaden the antitumor immune responses through effects on both innate and adaptive immunity to combat both STEAP1-positive and -negative disease.

Reviewer #4 (Remarks to the Author):

The authors have adequately addressed my comments and concerns.

Response Letter-2

Reviewer #2 (Remarks to the Author):

The manuscript has been revised according to my suggestions. The manuscript can now be published. There are only two new comments from my side on minor issues:

Minor Comment 1: Lines 93-104: References should be added.

Response: The relevant references have been added in the revised manuscript.

New comment: References should be added not only to the text in lines 97-102, but also to the text in lines 102-108.

Response: We appreciate the reviewer's assessment of our revised manuscript. As per the suggestion, the relevant references have been added to the manuscript.

Minor Comment 2: *Suppl. Fig. 15: The bars showing INF- γ release in the "untransduced T cells + CBD-IL-12 group" (day 0 vs day 8) suggest a significant difference. If so, the p value should be added.*

Response: Statistical analysis of IFN- γ release showed no significant difference between untransduced T cells + CBD-IL-12 treatment (day 0 vs day 8). This is likely due to the large standard deviation between the mouse samples.

New comment: Fig. 7e (formerly Suppl. Fig15): It cannot be understood that the green bars (INF γ release of untransduced T cells + CBD-IL-12, day 0 vs. day 8), should not be significantly different, as no large standard deviations can be seen. However, since the authors confirm that there is no significance, this is to be accepted.

Response: We appreciate reviewer's concern for INF γ release in the untransduced T cells + CBD-IL-12 (day 0 vs. day 8) in Figure 7e and the absence of statistical significance. We would like to bring to reviewers' attention that the Y axis of the graph represents a logarithmic scale. This statistical analysis is not significant by ANOVA.

Reviewer #3 (Remarks to the Author):

Comment 1: *Response to comment to reviewer3 regarding tumor antigen loss is not given as indicated by the authors in lines 738-745 and 752 – 760. Discussion section already ends line 744.*

Moreover the response of the authors where they describe how they correct false description of preconditioning they mention that they now include in the manuscript that "additional cytokines, radiotherapy, CI may be strategies to deepen anti-tumor responses" in lines 760-765. There is no line 760-765, in line 745 begins already Method section.

Response: We apologize for the confusion. The line numbers provided referred to those when the manuscript was viewed in track-changes mode.

Comment 2: *The already originally submitted staining of CD3 is not of the highest quality. Low magnification, low contrast, positive cells not clearly distinguishable, can only be approximated.*

The now subsequently submitted multiparametric phenotype analysis is based on flow and is thus statistical, why do the authors not provide a multiparametric (classical) histological analysis of the cells in the TME? FC analysis is very questionable, especially DCs and

macrophages are very difficult to "isolate". This method is never quantitative, not reproducible, and different for each tumour as it depends on the connective tissue portion of the tumour. Also, the phenotype changes if any form of digestion is used. Ultrahigh-content imaging Macsima or PhenoCycler (formerly Codex) allows deep spatial phenotyping with target cells in the relevant context and in the respective cell-typical quantity ratios. This and conventional histology are much more informative than phenotyping cells that have been manipulated and taken out of their context. Stroma reversion is not convincingly demonstrated.

Response: We believe the multiparametric flow cytometry data provided in the revised manuscript is corroborated by the findings of the single-cell RNA-seq analysis of CBD-IL-12 treated RM9 tumors. We agree that future studies should involve ultrahigh content spatial phenotyping and we intend to study in-depth stromal reversion with multiplex immunofluorescence-based panels. However, this will require significant optimization and the use of multiparametric flow cytometry to analyze components of the tumor immune microenvironment is considered a standard in the field.

Comment 3: *25µg/g body weight is high dose therapy. Perhaps it makes sense to take a tumour cell line next time that does not close the time window before successful stroma reversion.*

Response: We apologize and would like to clarify that mice were administered a fixed dose of 25 ug of CBD-IL-12 as mentioned in the Methods section. We agree with the reviewer's comment that future studies utilizing different, perhaps less aggressive syngeneic prostate cancer cell lines should be used to test STEAP1 CAR T cell and CBD-IL-12 therapy.

Reviewer #4 (Remarks to the Author):

The authors have adequately addressed my comments and concerns.

Response: We thank the reviewer for their time/effort.